# Neural Jump Ordinary Differential Equations: Consistent Continuous-Time Prediction and Filtering

**Calypso Herrera**    **Florian Krach**    **Josef Teichmann**
Department of Mathematics, ETH Zurich, Switzerland
`{firstname.lastname}@math.ethz.ch`

## Abstract

Combinations of neural ODEs with recurrent neural networks (RNN), like GRU-ODE-Bayes or ODE-RNN are well suited to model irregularly observed time series. While those models outperform existing discrete-time approaches, no theoretical guarantees for their predictive capabilities are available. Assuming that the irregularly-sampled time series data originates from a continuous stochastic process, the $L^2$-optimal online prediction is the conditional expectation given the currently available information. We introduce the Neural Jump ODE (NJ-ODE) that provides a data-driven approach to learn, continuously in time, the conditional expectation of a stochastic process. Our approach models the conditional expectation between two observations with a neural ODE and jumps whenever a new observation is made. We define a novel training framework, which allows us to prove theoretical guarantees for the first time. In particular, we show that the output of our model converges to the $L^2$-optimal prediction. This can be interpreted as solution to a special filtering problem. We provide experiments showing that the theoretical results also hold empirically. Moreover, we experimentally show that our model outperforms the baselines in more complex learning tasks and give comparisons on real-world datasets.

## 1 Introduction

Stochastic processes are widely used in many fields to model time series that exhibit a random behaviour. In this work, we focus on processes that can be expressed as solutions of stochastic differential equations (SDE) of the form

$$dX_t = \mu(t, X_t)dt + \sigma(t, X_t)dW_t \, ,$$

with certain assumptions on the drift $\mu$ and the diffusion $\sigma$. With respect to the $L^2$-norm, the best prediction of a future value of the process is provided by the conditional expectation given the current value. If the drift and diffusion are known or a good estimation is available, the conditional expectation can be approximated by a Monte Carlo (MC) simulation. However, since $\mu$ and $\sigma$ are usually unknown, this approach strongly depends on the assumptions made on their parametric form. A more flexible approach is given by *neural SDEs*, where the drift $\mu$ and diffusion $\sigma$ are modelled by neural networks (Tzen & Raginsky, 2019; Li et al., 2020; Jia & Benson, 2019). Nevertheless, modelling the diffusion can be avoided if one is only interested in forecasting the behaviour instead of sampling new paths.

An alternative widely used approach is to use Recurrent Neural Networks (RNN), where a neural network dynamically updates a latent variable with the observations of a discrete input time-series. RNNs are successfully applied to tasks for which time-series are regularly sampled, as for example speech or text recognition. However, often observations are irregularly observed in time. The standard approach of dividing the time-line into equally-sized intervals and imputing or aggregating observations might lead to a significant loss of information (Rubanova et al., 2019). Frameworks that overcome this issue are the GRU-ODE-Bayes (Brouwer et al., 2019) and the ODE-RNN (Rubanova et al., 2019), which combine a RNN with a *neural ODE* (Chen et al., 2018). In standard RNNs, the

hidden state is updated at each observation and constant in between. Conversely, in the GRU-ODE-Bayes and ODE-RNN framework, a neural ODE is trained to model the continuous evolution of the hidden state of the RNN between two observations. While GRU-ODE-Bayes and ODE-RNN both provide convincing empirical results, they lack thorough theoretical guarantees.

**Contribution.** In this paper, we introduce a mathematical framework to precisely describe the problem statement of online prediction and filtering of a stochastic process with temporal irregular observations. Based on this rigorous mathematical description, we introduce the Neural Jump ODE (NJ-ODE). The model architecture is very similar to the one of GRU-ODE-Bayes and ODE-RNN, however we introduce a novel training framework, which in contrast to them allows us to prove convergence guarantees for the first time. Moreover, we demonstrate empirically the capabilities of our model.

**Precise problem formulation.** We emphasize that a precise definition of all ingredients is needed, to be able to show theoretical convergence guarantees, which is the main purpose of this work. Since the objects of interest are stochastic processes, we use tools from probability theory and stochastic calculus. To make the paper more readable and comprehensible also for readers without background in these fields, the precise formulations and demonstrations of all claims are given in the appendix, while the main part of the paper focuses on giving well understandable heuristics.

## 2  PROBLEM STATEMENT

The problem we consider in this work, is the online forecasting of temporal data. We assume that we make observations of a Markovian stochastic process described by the stochastic differential equation (SDE)

$$dX_t = \mu(t, X_t)dt + \sigma(t, X_t)dW_t,  \tag{1}$$

at irregularly-sampled time points. Between those observation times, we want to predict the stochastic process, based only on the observations that we made previously in time, excluding the possibility to interpolate observations. Due to the Markov property, only the last observation is needed for an optimal prediction. Hence, after each observation we extrapolate the current observation into the future until we make the next observation. The time at which the next observation will be made is random and assumed to be independent of the stochastic process itself.

More precisely, we suppose to have a training set of $N$ independent realisations of the $\mathbb{R}^{d_X}$-dimensional stochastic process $X$ defined in (1). Each realisation $j$ is observed at $n_j$ random observation times $t_1^{(j)}, \ldots, t_{n_j}^{(j)} \in [0, T]$ with values $x_1^{(j)}, \ldots, x_{n_j}^{(j)} \in \mathbb{R}^{d_X}$. We assume that all coordinates of the vector $x_i^{(j)}$ are observed. We are interested in forecasting how a new independent realization evolves in time, such that our predictions of $X$ minimize the expected squared distance ($L^2$-metric) to the true unknown path. The optimal prediction, i.e. the $L^2$-minimizer, is the conditional expectation. Given that the value of the new realization at time $t$ is $x_t$, we are therefore interested in estimating the function

$$f(x_t, t, s) := E[X_{t+s}|X_t = x_t], \ s \geq 0,  \tag{2}$$

which is the $L^2$-optimal prediction until the next observation is made. To learn an approximation $\hat{f}$ of $f$ we make use of the $N$ realisations of the training set. After training, $\hat{f}$ is applied to the new realization. Hence, this can be interpreted as a special type of filtering problem. The following example illustrates the considered problem.

**Example.** A complicated to measure vital parameter of patients in a hospital is measured multiple times during the first 48 hours of their stay. For each patient, this happens at different times depending on the resources, hence the observation dates are irregular and exhibit some randomness. Patient 1 has $n_1 = 4$ measurements at hours $(t_1^{(1)}, t_2^{(1)}, t_3^{(1)}, t_4^{(1)}) = (1, 14, 27, 34)$ where the values $(x_1^{(1)}, x_2^{(1)}, x_3^{(1)}, x_4^{(1)}) = (0.74, 0.65, 0.78, 0.81)$ are measured. Patient 2 only has $n_2 = 2$ measurements at hours $(t_1^{(2)}, t_2^{(2)}) = (3, 28)$ where the values $(x_1^{(2)}, x_2^{(2)}) = (0.56, 0.63)$ are measured. Similarly, the $j$-th patient has $n_j$ measurements at times $(t_1^{(j)}, \ldots, t_{n_j}^{(j)})$ and has the measured values $(x_1^{(j)}, \ldots, x_{n_j}^{(j)})$. Based on this data, we want to forecast the vital parameter of new patients coming to the hospital. In particular, for a patient with measured values $x_1$ at time $t_1$, we want to predict what

his values will likely be at any time $t_1 + s > t_1$. Importantly, we do not only focus on predicting the value at some $t_2 > t_1$, but we want to know the entire evolution of the value.

## 3 BACKGROUND

**Recurrent Neural Network.** The input to a RNN is a discrete time series of observations $\{x_1, \cdots, x_n\}$. At each observation time $t_{i+1}$, a neural network, the RNNCell, updates the latent variable $h$ using the previous latent variable $h_i$ and the input $x_{i+1}$ as

$$h_{i+1} := \text{RNNCell}(h_i, x_{i+1}).$$

**Neural Ordinary Differential Equation.** Neural ODEs (Chen et al., 2018) are a family of continuous-time models defining a latent variable $h_t := h(t)$ to be the solution to an ODE initial-value problem

$$h_t := h_0 + \int_{t_0}^t f(h_s, s, \theta) ds, \quad t \geq t_0, \tag{3}$$

where $f(\cdot, \cdot, \theta) = f_\theta$ is a neural network with weights $\theta$. Therefore, the latent variables can be updated continuously by solving this ODE (3). We can emphasize the dependence of $h_t$ on a numerical ODE solver by rewriting (3) as

$$h_t := \text{ODESolve}(f_\theta, h_0, (t_0, t)). \tag{4}$$

**ODE-RNN.** ODE-RNN (Rubanova et al., 2019) is a mixture of a RNN and a neural ODE. In contrast to a standard RNN, we are not only interested in an output at the observation times $t_i$, but also in between those times. In particular, we want to have an output stream that is generated continuously in time. This is achieved by using a neural ODE to model the latent dynamics between two observation times, i.e. for $t_{i-1} < t < t_i$ the latent variable is defined as in (3) and (4), with $h_0$ and $t_0$ replaced by $h_{i-1}$ and $t_{i-1}$. At the next observation time $t_i$, the latent variable is updated by a RNN with the new observation $x_i$. Fixing $h_0$, the entire latent process can be computed by iteratively solving an ODE followed by applying a RNN. Rubanova et al. (2019) write this as

$$\begin{cases} h_i' & := & \text{ODESolve}(f_\theta, h_{i-1}, (t_{i-1}, t_i)) \\ h_i & := & \text{RNNCell}(h_i', x_i). \end{cases} \tag{5}$$

**GRU-ODE-Bayes.** The model architecture describing the latent variable in GRU-ODE-Bayes (Brouwer et al., 2019) is defined as a special case of the ODE-RNN architecture. In particular, a gated recurrent unit (GRU) is used for the RNN-cell and a continuous version of the GRU for the neural ODE $f_\theta$. Therefore, we focus on explaining the difference between our model architecture and the ODE-RNN architecture, in the following section.

## 4 PROPOSED METHOD – NEURAL JUMP ODE

**Markovian paths.** Our assumptions on the stochastic process $X$ imply that it is a Markov process. In particular, the optimal prediction of a future state of $X$ only depends on the current state rather than on the full history. Hence, the previous values do not provide any additional useful information for the prediction.

**JumpNN instead of RNN.** Using a neural ODE between two observation has the advantage that it allows to continuously model the hidden state between two observations. But since the underlying process is Markov, there is no need to use a RNN-cell to model the updates of the hidden state at each new observation. Instead, whenever a new observation is made, the new hidden state can solely be defined from this observation. Therefore, we replace the RNN-cell used in ODE-RNN by a standard neural network mapping the observation to the hidden state. We call this the jumpNN which can be interpreted as an encoder map. Compared to ODE-RNN, this architecture is easier to train.

**Last observation and time increment as additional inputs for the neural ODE.** The neural network $f_\theta$ used in the neural ODE takes two arguments as inputs, the hidden state $h_t$ and the current time $t$. However, our theoretical problem analysis suggests, that instead of $t$ the last observation time $t_{i-1}$ and the time increment $t - t_{i-1}$ should be used. Additionally the last observation $x_{i-1}$ should also be part of the input.

**NJ-ODE.** Combining the ODE-RNN architecture (5) with the previous considerations, we introduce the modified architecture of Neural Jump ODE (NJ-ODE)

$$
\begin{cases}
h_i' & := \quad \text{ODESolve}(f_\theta, (h_{i-1}, x_{i-1}, t_{i-1}, t - t_{i-1}), (t_{i-1}, t_i)) \\
h_i & := \quad \text{jumpNN}(x_i).
\end{cases}
\tag{6}
$$

An implementable version of this method is presented in the Algorithm 1. A neural ODE $f_\theta$ transforms the hidden state between observations, and the hidden state jumps according to jumpNN when a new observation is available. The outputNN, a standard neural network, maps any hidden state $h_t$ to the output $y_t$. To implement the continuous-in-time ODE evaluation, a discretization scheme is provided by the inner loop. In the training process, the weights of all three neural networks, jumpNN, the neural ODE $f_\theta$ and outputNN are optimized.

---

**Algorithm 1** The NJ-ODE. A small step size $\Delta t$ is fixed and we denote $t_{n+1} := T$.

---

  **Input:** Data points with timestamps $\{(x_i, t_i)\}_{i=0\ldots n}$,
  **for** $i = 0$ **to** $n$ **do**
    $h_{t_i} = \text{jumpNN}(x_i)$                     ▷ Update hidden state given next observation $x_i$
    $y_{t_i} = \text{outputNN}(h_{t_i})$                              ▷ compute output
    $s \leftarrow t_i$
    **while** $s + \Delta t \leq t_{i+1}$ **do**
      $h_{s+\Delta t} = \text{ODESolve}(f_\theta, h_s, x_i, t_i, s - t_i, (s, s + \Delta t))$      ▷ get next hidden state
      $y_{s+\Delta t} = \text{outputNN}(h_{s+\Delta t})$                   ▷ compute output
      $s \leftarrow s + \Delta t$
    **end while**
  **end for**

---

**Objective function.** Our goal is to train the NJ-ODE model such that its output approximates the conditional expectation (2), which is the optimal prediction of the target process $X$ with respect to the $L^2$-norm. Therefore, we define a new objective function, with which we can prove convergence. Let $y_{i-}$ denote the output of the NJ-ODE at $t_i$ before the jump and $y_i$ the output at $t_i$ after the jump. Note that the outputs depend on parameters $\theta$ and the previously observed $x_i$ which are inputs to the model. Then the objective function is defined as

$$
\hat{\Phi}_N(\theta) := \underbrace{\frac{1}{N} \sum_{j=1}^{N}}_{\text{paths}} \underbrace{\frac{1}{n^j} \sum_{i=1}^{n^j}}_{\text{dates}} \Big( \underbrace{|x_i^{(j)} - y_i^{(j)}|}_{\substack{\text{jump part} \\ \text{at observations}}} + \underbrace{|y_i^{(j)} - y_{i-}^{(j)}|}_{\substack{\text{continuous part} \\ \text{between two observations}}} \Big)^2.
\tag{7}
$$

We give an intuitive explanation for this definition. The "jump part" of the loss function forces the jumpNN to produce good updates based on new observations, while the other part forces the jump size to be small in (the empirical) $L^2$-norm. Since the conditional expectation minimizes the jump size with respect to the $L^2$-norm, this forces the neural ODE $f_\theta$ to continuously transform the hidden state such that the output approximates the conditional expectation. Moreover, both parts of the loss function force the outputNN to reasonably transform the hidden state $h_t$ to the output $y_t$.

## 5   MAIN RESULT – THEORETICAL CONVERGENCE GUARANTEE

In the following we informally state our main result. To formally state and prove the theorem, precise definitions of all ingredients are needed. This analysis is provided in the appendix where the following results is stated in Theorem E.2 and Theorem E.13.

**Theorem 5.1** (informal). *We assume that for each number of paths $N$ and for every size of the neural networks $M$, their weights are chosen optimally, as to minimize $\hat{\Phi}_N(\theta)$. Then, if $N$ and $M$ tend to infinity, the output of NJ-ODE converges in mean ($L^1$-convergence) to the conditional exception of the stochastic process $X$ given the current information.*

An intuitive explanation for this theorem was given with the definition of the objective function. In this result, the focus lies on the convergence analysis under the assumption that optimal weights are found. In the Appendix we discuss why this assumption is not restrictive.

## 6 EXPERIMENTS

For further details and results for all experiments see Appendix F.

### 6.1 TRAINING ON SYNTHETIC DATASETS

**Evaluation metric.** For synthetic datasets where an analytic formula for the conditional expectation exists, we can evaluate the distance of the model output to the target process (2). We use a sampling time grid with equidistant step size $\Delta_t := \frac{T}{K}$, $K \in \mathbb{N}$, on $[0, T]$. On this grid, we compare for path $j$ at time $t$, the true conditional expectation $\hat{x}_t^{(j)}$ with the predicted conditional expectation (the model output) $y_t^{(j)}$. For $N_2$ test samples, the evaluation metric is defined as

$$\text{eval}(\hat{x}, y) := \frac{1}{N_2} \sum_{j=1}^{N_2} \frac{1}{K+1} \sum_{i=0}^{K} \left( \hat{x}_{i\Delta_t}^{(j)} - y_{i\Delta_t}^{(j)} \right)^2. \tag{8}$$

**Black-Scholes, Ornstein-Uhlenbeck and Heston.** We test our algorithm on three scalar stochastic models, Black-Scholes, Ornstein-Uhlenbeck and Heston, with fixed parameters. For each model, we generate a dataset by sampling $N = 20'000$ paths on the time interval $[0, 1]$ using the Euler-scheme with 100 time steps. Independently for each path, on average $10\%$ of the grid points are randomly chosen as observation times. The NJ-ODE is trained on $80\%$ of the data. On the remaining $20\%$ the model is tested, by comparing the loss function (31) computed with the NJ-ODE to the loss function computed with the true conditional expectation (Figure 2). During training, the relative difference becomes very small, hence the true conditional expectation is nearly replicated.

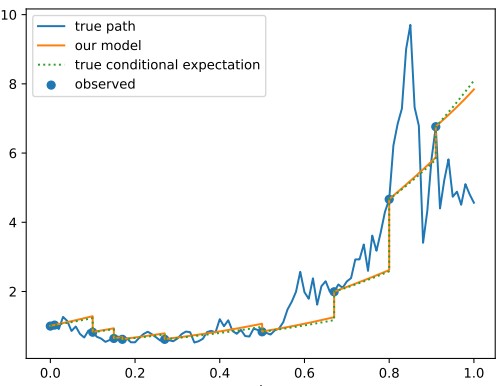

Figure 1: Predicted and true conditional expectation on a test sample of the Heston dataset.

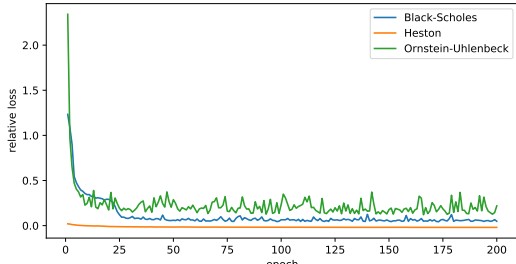

Figure 2: The relative difference between the loss function computed with the predicted conditional expectation and the loss function computed with the true conditional expectation is shown with respect to the number of epochs.

### 6.2 FURTHER SYNTHETIC DATASETS

**Heston model without Feller condition.** We also train our model on a Heston dataset for which the Feller condition is not satisfied. As explained by Andersen (2007); Jean-François et al. (2015) this is a more delicate situation. We see that our model is very robust, since even in this critical case it learns to replicate the true conditional expectation process. In the Heston model, the variance of the stochastic process $X_t$ is a stochastic process, $v_t$. Here, we train our model to predict both processes at the same time. The training on this 2-dimensional dataset is successful as can be seen in Figure 3. The minimal evaluation metric after 200 epochs is 0.0983.

**Dataset with changing regime.** In this experiment we test how well our model can deal with stochastic processes, that undergo an (abrupt) change of regime at a certain point in time. Many real world time series might exhibit such a change of regime. Some examples are listed below.

- Longitudinal patient health recordings might experience changes depending on seasonal or longer-term influences, as for example due to the seasonal flue or currently the Covid-19 pandemic.
- In many regions climate data has strong seasonal dependencies, that can lead to relatively abrupt changes as for example when the weather changes from dry to rain season.

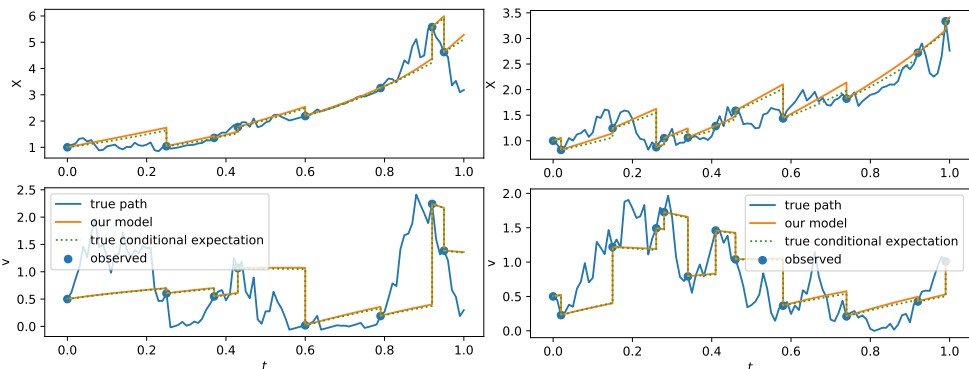

Figure 3: Heston model without Feller condition. In both plots, the upper sub-plot corresponds to the 1-dimensional path of $X_t$ and the lower sub-plot corresponds to the 1-dimensional path of $v_t$.

- A stock market that suddenly changes from a bullish to a bearish market, for example due to a macro-economic event. An example for this would be the start of the Covid-19 crisis in the first quarter of 2020.

We test a change of regime by combining two synthetic datasets. On the first half of the time interval $[0, 0.5]$ we use the Ornstein-Uhlenbeck and on the second half $[0.5, 1]$ the Black-Scholes model. In Figure 4 we see that our model correctly learns the change of regime. The minimal evaluation metric after 200 epochs is $0.0463$.

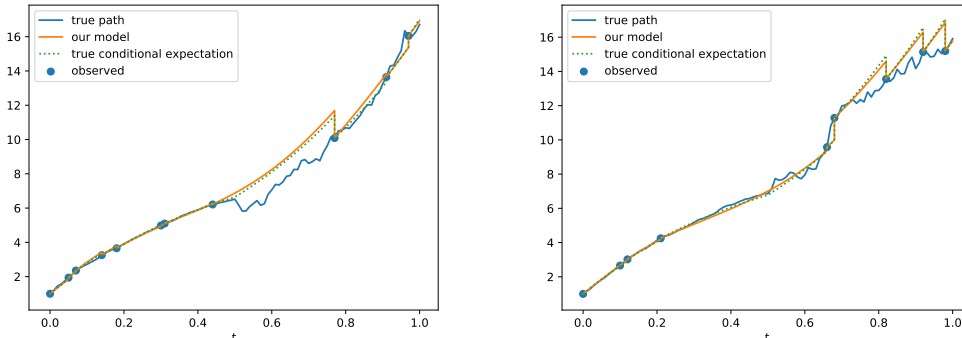

Figure 4: Our model evaluated o a stochastic dataset that follows an Ornstein-Uhlenbeck SDE on the time interval $[0, 0.5]$ and an Black-Scholes model on the time interval $(0.5, 1]$. We see that our model correctly learns the change of regime.

**Dataset with explicit time dependence.** Many real world datasets, have an explicit time dependence, i.e. the drift and diffusion of (9) explicitly depend on $t$. Examples are all datasets that have a certain periodicity, as for example weather data (seasonal and daily periodicity), intraday periodicity of stock prices (Andersen & Bollerslev, 1997) or prices of certain seasonal goods. We incorporate an explicit time dependence into the Black-Scholes dataset, by replacing the drift constant $\mu$ with the time dependent constant $\frac{\alpha}{2}(\sin(\beta t) + 1)$, for $\alpha, \beta > 0$. In Figure 5 we see that the model learns to adapt to the time-dependent coefficients. The minimal evaluation metric after 100 epochs is $0.0215$ ($\beta = 2\pi$) and $0.02805$ ($\beta = 4\pi$) respectively.

## 6.3 EMPIRICAL CONVERGENCE STUDY

We confirm the theoretical results of Theorem 5.1 by an empirical convergence study for growing numbers of training samples $N_1$ and network size $M$, where the performance is measured by the evaluation metric (8). For each combination of the number of training samples $N_1$ and the neural network size $M$, the NJ-ODE is trained 5 times on the Black-Scholes, Ornstein-Uhlenbeck and

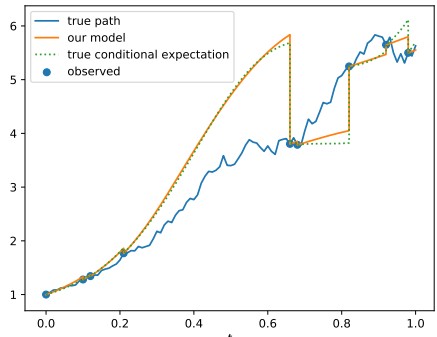 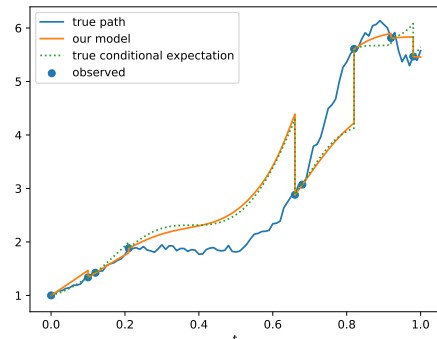

Figure 5: NJ-ODE evaluated on the time-dependent Black-Scholes dataset with $\beta = 2\pi$ (left) and $\beta = 4\pi$ (right).

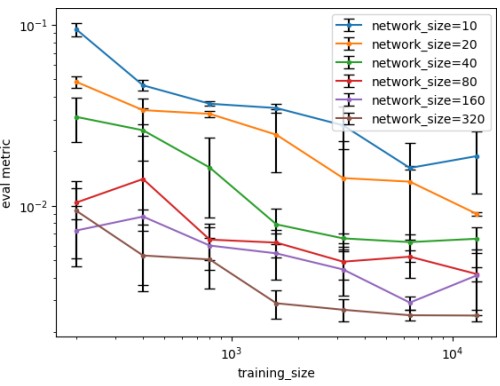 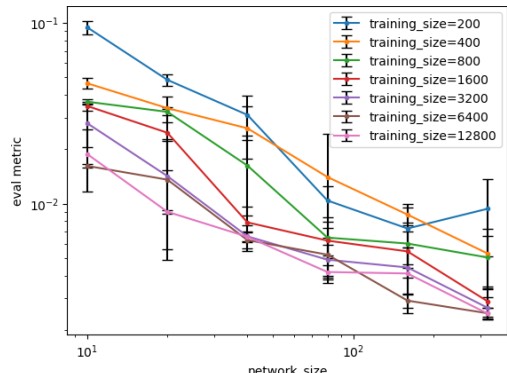

Figure 6: Black-Scholes dataset. Mean $\pm$ standard deviation (black bars) of the evaluation metric for varying training samples $N_1$ and network size $M$.

Heston datasets. The means of the evaluation metric over the 5 trials with their standard deviations are shown in Figure 6 for Black-Scholes. Already 200 training samples lead to a small evaluation metric if the network size is big enough, suggesting that the Monte Carlo approximation of the loss function is good, already with a few samples.

### 6.4 COMPARISON TO GRU-ODE-BAYES ON SYNTHETIC DATA

On the Black-Scholes, Ornstein-Uhlenbeck and Heston datasets, we compare our model to GRU-ODE-Bayes (Brouwer et al., 2019), which is, to the best of our knowledge, the neural network based method with the most similar task to ours. Results of our comparison are shown in Table 6.4. On the Black-Scholes and Ornstein-Uhlenbeck dataset, our model performs similarly to GRU-ODE-Bayes. However, on the more complicated Heston dataset, the training of GRU-ODE-Bayes is unstable and does not converge. On the other hand, our model converges during training. Although the value of the evaluation metric is much higher than for Black-Scholes and Ornstein-Uhlebeck, the resulting model output is still meaningful, which is not the case for GRU-ODE-Bayes. Hence we conclude, that GRU-ODE-Bayes cannot be applied reasonably for the Heston dataset, while our method works as the theoretical results suggest.

### 6.5 REAL WORLD DATASETS WITH INCOMPLETE OBSERVATIONS

**Self-imputation for incomplete observations.** Until now we assumed that at each observation time all coordinates of the stochastic process $X$ are observed. However, in many real world applications, the observations are incomplete, i.e. only for some of the coordinates an observation is available. To deal with such incomplete observations, we propose the following self-imputation method. Whenever

Table 1: The minimal, last and average value of the evaluation metric throughout the 100 epochs of training are shown for GRU-ODE-Bayes and our method, together with the number of trainable parameters.

| | Black-Scholes | | Ornstein-Uhlenbeck | | Heston | |
|---|---|---|---|---|---|---|
| | GRU | ours | GRU | ours | GRU | ours |
| min | $7 \cdot 10^{-4}$ | $7 \cdot 10^{-4}$ | $3 \cdot 10^{-5}$ | $5 \cdot 10^{-4}$ | 4.24 | 1.33 |
| last | $3 \cdot 10^{-3}$ | $8 \cdot 10^{-3}$ | $1 \cdot 10^{-4}$ | $6 \cdot 10^{-4}$ | 30.9 | 1.35 |
| average | $3 \cdot 10^{-3}$ | $1 \cdot 10^{-2}$ | $7 \cdot 10^{-4}$ | $8 \cdot 10^{-4}$ | 56.2 | 1.65 |
| params | $112'602$ | $10'071$ | $112'602$ | $10'071$ | $112'602$ | $10'071$ |

an observation $x_i \in \mathbb{R}^{d_X}$ is made the mask $m_i \in \{0,1\}^{d_X}$ tells whether the $k$-th coordinate of the vector $x_i$ is observed ($m_i^k = 1$) or not ($m_i^k = 0$). Then we use the prediction $y_{i-}$ of our model to generate the imputed observation vector

$$\tilde{x}_i := m_t \odot x_i + (\mathbf{1}_{d_X} - m_i) \odot y_{i-},$$

where $\odot$ is the element-wise multiplication (Hadamar product) and $\mathbf{1}_{d_X} \in \mathbb{R}^{d_X}$ is the one-vector. Instead of $x_i$ we use $(\tilde{x}_i, m_i)$ as an input for the jump part jumpNN. The intuition behind this definition is the following. In the one dimensional case, if we do not make an observation, but input $y_{t-}$ as if it was an observation, we expect that this does not change the output $y_s$ for $s \geq t$. From this point of view, $y_{t-}$ does not provide any additional information for the model. Similarly, we expect that imputing $y_{t-}$ for unobserved coordinates does not provide any information about this coordinate to the model. However, since the model might learn to transfer the information about an observed coordinate to an unobserved one, we extend the input to also include the information which coordinates were observed. For the ODE part we use $y_i$, the prediction after processing the input $(\tilde{x}_i, m_i)$, as input instead of $x_i$. Here, the intuition is that if the model learns how to best use the incomplete observation, i.e. if the jump is good, then this is the best approximation for $x_i$. Our objective function is adjusted by multiplying each term in the sum with the mask.

**Climate forecast.** We compare our model to GRU-ODE-Bayes on the USHCN daily dataset (Menne et al., 2016), using the same experimental setting as was used by Brouwer et al. (2019). We train a small (S) and a large (L) version of NJ-ODE with different total number of parameters. The validation set was used for early stopping after the first 100 epochs, where we trained for a total of 200 epochs. We see in Table 6.5 that our small version performs slightly worse than GRU-ODE-Bayes while our large version slightly outperforms it.

Table 2: Mean and standard deviation of MSE on the test sets of USHCN. Result of baselines were reported by Brouwer et al. (2019). Where known, the number of trainable parameters is reported.

| | USHCN – MSE | # params |
|---|---|---|
| neural ODE-VAE | $0.96 \pm 0.11$ | - |
| neural ODE-VAE-MASK | $0.83 \pm 0.10$ | - |
| sequential VAE | $0.83 \pm 0.07$ | - |
| GRU-SIMPLE | $0.75 \pm 0.12$ | - |
| GRU-D | $0.53 \pm 0.06$ | - |
| T-LSTM | $0.59 \pm 0.11$ | - |
| GRU-ODE-Bayes | $0.43 \pm 0.07$ | $42'640$ |
| NJ-ODE (S) | $0.45 \pm 0.06$ | $10'925$ |
| NJ-ODE (L) | $\mathbf{0.40 \pm 0.07}$ | $571'305$ |

**Physionet.** We compare our model to the latent ODE on their extrapolation task on the PhysioNet Challenge 2012 dataset (Goldberger et al., 2000), using the same experimental setting as was used by Rubanova et al. (2019). The mean and standard deviation over 5 runs starting at different random initializations is reported for our model. We see in Table 6.5 that our model outperforms the latent ODE models although having only about a seventh of the trainable weights.

Table 3: Mean and standard deviation of MSE on the test set of physionet. Result of baselines were reported by Rubanova et al. (2019). Where known, the number of trainable parameters is reported.

| | Physionet – MSE ($\times 10^{-3}$) | # params |
|---|---|---|
| RNN-VAE | $3.055 \pm 0.145$ | - |
| Latent ODE (RNN enc.) | $3.162 \pm 0.052$ | - |
| Latent ODE (ODE enc) | $2.231 \pm 0.029$ | $163'972$ |
| Latent ODE + Poisson | $2.208 \pm 0.050$ | $181'723$ |
| NJ-ODE | $\mathbf{1.945 \pm 0.007}$ | $24'423$ |

## 7 RELATED WORK

**Stochastic filtering theory.** Our main theorem is similar to the theoretical results of *neural filtering* (Lo, 2009), which is a neural network approach to stochastic filtering theory in discrete time. In stochastic filtering, potentially incomplete and noisy observations of $X$ are available continuously in time. This observation process $Y$ is usually described by the dynamics $dY_t = h(X_t)dt + dB_t$, where $h$ is a measurable function and $B$ is a Brownian motion. Stochastic filtering then estimates the conditional law of $X_t$ given the noisy observations $(Y_s)_{0 \leq s \leq t}$ (Bain & Crisan, 2008, Def. 3.2) and therefore can provide conditional expectations. Comparably, we directly compute the conditional expectation given the last observation. Similar to our assumptions, in the neural filtering approach of GRU-ODE-Bayes (Brouwer et al., 2019) and (Ryder et al., 2018) observations are only available at irregular discrete time points. They approximate the conditional law of $X$ given the last observation by a Gaussian distribution and learn its mean and variance parameters. In particular, the conditional expectation is then given by this mean parameter. In contrast, we do not make normality assumptions about the conditional distribution and we theoretically prove convergence to the true conditional expectation.

**Neural ODEs with jumps.** Except for GRU-ODE-Bayes (Brouwer et al., 2019) and ODE-RNN (Rubanova et al., 2019) another work studying a neural ODE with jumps is *Neural Jump SDE* (NJSDE) (Jia & Benson, 2019). Similar to the NJ-ODE framework (29), the latent process of NJSDE is described by a neural ODE with jumps at random times. This model is used to describe hybrid systems which evolve continuously in time but may also be interrupted by stochastic events. In contrast to that, we model the conditional expectation of a continuous stochastic process.

## 8 CONCLUSION

We presented the Neural Jump ODE, a data-driven framework for modelling the conditional expectation of a stochastic process given the previous observations. We introduced a rigorous mathematical description of our model and more generally for the class of neural ODE based models. Moreover, for the first time we provided theoretical guarantees for a model falling in this category. We evaluated our model empirically on six synthetic and two real world datasets. In comparison to the baselines GRU-ODE-Bayes and latent ODE, we achieved better results especially on complex datasets.

## ACKNOWLEDGEMENT

The authors thank Andrew Allan, Robert A. Crowell, Anastasis Kratsios and Pierre Ruyssen for helpful discussions, providing references and insights. Moreover, the authors thank the reviewers for their thoughtful feedback that contributed to significantly improve the paper. The authors gratefully acknowledge financial support coming from the Swiss National Science Foundation (SNF) under grant 179114.

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

APPENDIX

## A  SETUP

### A.1  STOCHASTIC PROCESS $X$

In this section we rigorously describe the process $X$ and give the assumptions which are needed in order to derive the convergence results. Let $d_X, d_W \in \mathbb{N}$ and $T > 0$ be the fixed time horizon. Consider a filtered probability space $(\Omega, \mathcal{F}, \mathbb{F} := \{\mathcal{F}_t\}_{0 \le t \le T}, \mathbb{P})$, on which an adapted $d_W$-dimensional Brownian motion $\{W_t\}_{t \in [0,T]}$ is defined. We define the stochastic process[1] $X := (X_t)_{t \in [0,T]}$ as the solution of the stochastic differential equation (SDE)

$$dX_t = \mu(t, X_t)dt + \sigma(t, X_t)dW_t \,, \tag{9}$$

for all $0 \le t \le T$, where $X_0 = x \in \mathbb{R}^{d_X}$ is the starting point and the measurable functions $\mu : [0,T] \times \mathbb{R}^{d_X} \to \mathbb{R}^{d_X}$ and $\sigma : [0,T] \times \mathbb{R}^{d_X} \to \mathbb{R}^{d_X \times d_W}$ are the *drift* and the *diffusion* respectively. We impose the following assumptions:

- **$X$ is continuous and square integrable**, i.e. for $\mathbb{P}$-a.e. $\omega \in \Omega$ the map $t \mapsto X_t(\omega)$ is continuous and $\mathbb{E}[X_t^2] < \infty$ for every $t \in [0,T]$.

- **$\mu$ and $\sigma$ are both globally Lipschitz continuous in their second component**, i.e. for $\varphi \in \{\mu, \sigma\}$ there exists a constant $\tilde{M} > 0$ such that for all $t \in [0,T]$

$$|\varphi(t,x) - \varphi(t,y)|_2 \le \tilde{M}|x - y|_2 \quad \text{and} \quad |\varphi(t,x)|_2 \le (1 + |x|_2)\tilde{M}. \tag{10}$$

  In particular, their growth is at most linear in the second component.

- **$\mu$ is bounded and continuous in its first component ($t$) uniformly in its second component ($x$)**, i.e. for every $t \in [0,T]$ and $\varepsilon > 0$ there exists a $\delta > 0$ such that for all $s \in [0,T]$ with $|t - s| < \delta$ and all $x \in \mathbb{R}^{d_X}$ we have $|\mu(t,x) - \mu(s,x)| < \epsilon$.

- **$\sigma$ is càdlàg** (right-continuous with existing left-limit) **in the first component and $L^2$ integrable with respect to $W$, $\sigma \in L^2(W)$**, i.e.

$$\mathbb{E}\left[ \sum_{i=1}^{d_X} \sum_{j=1}^{d_W} \int_0^T \sup_x \sigma_{i,j}(x,t)^2 d[W^j, W^j]_t \right] = \int_0^T |\sup_x \sigma(x,t)|_F^2 \, dt < \infty, \tag{11}$$

  where $|\cdot|_F$ denotes the Frobenius matrix norm. This is in particular implied if $\sigma$ is bounded.

### A.2  RANDOM OBSERVATION DATES

In this section we describe how we model the observation dates. We want to treat irregularly observed time-series, i.e. without a fixed number of observations and not necessarily equally distributed. Therefore, we suppose that we have a possibly random number of $n$ observations and that those observations are made at random times $t_1, \ldots, t_n$. For simplicity we make the assumption that the observation dates are independent of the stochastic process itself. In future work, this will be generalized to observation dates correlated with $X$, that could be modelled by a point process on the same probability space. In contrast, here we hard-code the independence assumption by considering a second probability space $(\tilde{\Omega}, \tilde{\mathcal{F}}, \tilde{\mathbb{P}})$, on which the random observation times of the stochastic process $X$ are defined. More precisely, we assume that:

- $n : \tilde{\Omega} \to \mathbb{N}_{\ge 0}$ is a random variable with $\mathbb{E}_{\tilde{\mathbb{P}}}[n] < \infty$, the random number of observations,

- $t_i : \tilde{\Omega} \to [0,T]$ for $0 \le i \le n$ are *sorted* [2] random variables, the random observation times.

We denote the joint pushforward measure of $n$ and $\{t_i\}_{0 \le i \le n}$ as $\tilde{\mathbb{P}}_t := (n, t_0, \ldots, t_n) \# \tilde{\mathbb{P}}$. The random variable $n$ can but does not have to be unbounded. If it is bounded, we define $K :=$

---

[1]A stochastic process is a collection of random variables $X_t : \Omega \to \mathbb{R}^{d_X}, \omega \mapsto X_t(\omega)$ for $0 \le t \le T$.

[2]For all $\tilde{\omega} \in \tilde{\Omega}, 0 = t_0 < t_1(\tilde{\omega}) < \cdots < t_{n(\tilde{\omega})}(\tilde{\omega}) \le T$.

$\max\left\{k \in \mathbb{N} \mid \tilde{\mathbb{P}}(n \geq k) > 0\right\}$ to be the maximal value of $n$, which otherwise is infinity. We use the notation $\mathcal{B}([0,T])$ for the Borel $\sigma$-algebra of the set $[0,T]$. Then we define for each $1 \leq k \leq K$

$$\lambda_k : \mathcal{B}([0,T]) \to [0,1], \quad B \mapsto \lambda_k(B) := \frac{\tilde{\mathbb{P}}(n \geq k, (t_k-) \in B)}{\tilde{\mathbb{P}}(n \geq k)},$$

which is a probability measure on the time interval as shown in the lemma below. Here, $(t_k-)$ means the left-point of $t_k$, for example, if $t_0 = 0, t_1 = 1$ we have $t_0 \in [0,1), t_1 \notin [0,1)$ but $t_0- \notin [0,1), t_1- \in [0,1)$.

**Lemma A.1.** *For $1 \leq k < K + 1$ the map $\lambda_k$ defines a probability measure.*

*Proof.* First we see that $\lambda_k([0,T]) = 1$ since $t_k$ maps to $[0,T]$. Furthermore, for any disjoint sets $B_i \in \mathcal{B}([0,T])$, $i \in \mathbb{N}$, we have $\{n \geq k, t_k- \in \dot{\cup}_{i\geq1}B_i\} = \dot{\cup}_{i\geq1}\{n \geq k, t_k- \in B_i\}$ and these sets are $\tilde{\mathcal{F}}$-measurable, since they are defined through pre-images of random variables. Therefore, the additivity of $\tilde{\mathbb{P}}$ implies that $\lambda_k(\dot{\cup}_{i\geq1}B_i) = \sum_{i\geq1}\lambda_k(B_i)$. □

Moreover, we define $\tau$ as the time of the last observation before a certain time $t$,

$$\tau : [0,T] \times \tilde{\Omega} \to [0,T], \quad (t,\tilde{\omega}) \mapsto \tau(t,\tilde{\omega}) := \max\{t_i(\tilde{\omega})|0 \leq i \leq n(\tilde{\omega}), t_i(\tilde{\omega}) \leq t\}.$$

**Example A.2.** *We give two examples how the random observation dates could be defined.*

- *Let $(N_t)_{t\in[0,T]}$ be a homogeneous Poisson point process with rate $r > 0$. Hence, $N(0) = 0$, $t \mapsto N(t)$ is constant except for jumps of size 1 at discrete random times and for any fixed $t \in [0,T]$, $N(t)$ is Poisson distributed, i.e.*

$$\tilde{\mathbb{P}}[N(t) = k] = \frac{(rt)^k}{k!}e^{-rt} \text{ for } k \in \mathbb{N}.$$

  *Then the number of observations is defined as $n := N(T)$ and satisfies $\mathbb{E}_{\tilde{\mathbb{P}}}[n] = rT$. The observation dates $t_1, \ldots, t_n$ are defined as the discontinuity times of $N$, i.e. the times where $N$ increases by 1. In particular, for $0 \leq k \leq n$, $t_k := \inf\{t|N(t) = k\} \leq T$. Therefore, for $0 \leq a \leq b \leq T$ we can rewrite*

$$\lambda_k((a,b)) = \frac{\tilde{\mathbb{P}}(N(a)<k,N(b)\geq k)}{\tilde{\mathbb{P}}(N(T)\geq k)}.$$

- *We define $n$ by one of the following options:*

  - *as a constant in $\mathbb{N}_{>0}$.*
  - *as a Binomial random variable, $n \sim \mathrm{Binom}(p, n_{\max})$, for $p \in (0,1)$, $n_max \in \mathbb{N}_{>0}$.*
  - *as a Geometric random variable, $n \sim \mathrm{Geom}(p)$, for $p \in (0,1)$.*
  - *as a Poisson random variable, $n \sim \mathrm{Poi}(r)$, for $r > 0$.*

  *and $t_1, \ldots, t_n$ are defined by choosing $n$ uniform random variables on $[0,T]$ and sorting them.*

### A.3 INFORMATION $\sigma$-ALGEBRA

In this section, we define a mathematical tool, the $\sigma$-algebra, that is essential to the description of the conditional expectation. This object describes which information is available at a certain time $t$. In the following, we leave away $\tilde{\omega} \in \tilde{\Omega}$ whenever the meaning is clear. We define the filtration of the currently available information $\mathbb{A} := (\mathcal{A}_t)_{t\in[0,T]}$ by

$$\mathcal{A}_t := \boldsymbol{\sigma}(X_{t_i}|t_i \leq t),$$

where $t_i$ are the observation times and $\boldsymbol{\sigma}(\cdot)$ denotes the generated $\sigma$-algebra. By the definition of $\tau$ we have $\mathcal{A}_t = \mathcal{A}_{\tau(t)}$ for all $t \in [0,T]$. $(\Omega \times \tilde{\Omega}, \mathcal{F} \otimes \tilde{\mathcal{F}}, \mathbb{F} \otimes \tilde{\mathcal{F}}, \mathbb{P} \times \tilde{\mathbb{P}})$ is the filtered product probability space which, intuitively speaking, combines the randomness of the stochastic process with the randomness of the observations. Here, $\mathbb{F} \otimes \tilde{\mathcal{F}}$ consists of the tensor-product $\sigma$-algebras $(\mathcal{F} \otimes \tilde{\mathcal{F}})_t := \mathcal{F}_t \otimes \tilde{\mathcal{F}}$ for $t \in [0,T]$. As explained in the remark below, $\mathcal{A}_t$ can be identified with a sub-$\sigma$-algebra of $(\mathcal{F} \otimes \tilde{\mathcal{F}})_t$.

**Remark A.3.** *The $\sigma$-fields $\mathcal{A}_t$ depend on $\tilde{\omega} \in \tilde{\Omega}$ as well. If we look at the product probability space $(\Omega \times \tilde{\Omega}, \mathcal{F} \otimes \tilde{\mathcal{F}}, \mathbb{F} \otimes \tilde{\mathcal{F}}, \mathbb{P} \times \tilde{\mathbb{P}})$, where $\mathbb{F} \otimes \tilde{\mathcal{F}}$ consists of the tensor-product $\sigma$-algebras $(\mathcal{F} \otimes \tilde{\mathcal{F}})_t := \mathcal{F}_t \otimes \tilde{\mathcal{F}}$ for $t \in [0, T]$, then*

$$\boldsymbol{\sigma}(X_{t_i} | t_i \leq t)$$
$$:= \boldsymbol{\sigma} \left( \left\{ \left\{ (\omega, \tilde{\omega}) \in \Omega \times \tilde{\Omega} \big| X_{t_i(\tilde{\omega})}(\omega) \in A, n(\tilde{\omega}) \geq i, t_i(\tilde{\omega}) \leq t \right\} \Big| A \in \mathcal{B}(\mathbb{R}^{d_X}), i \in \mathbb{N} \right\} \right)$$

*is a well defined sub-$\sigma$-algebra of $(\mathcal{F} \otimes \tilde{\mathcal{F}})_t$. Furthermore, we can recover the $\tilde{\omega}$-wise defined version of $\mathcal{A}_t$ by intersecting each set in it with $\Omega \times \{\tilde{\omega}\}$ and subsequently projecting the intersection on its first component. We use the notation $\tilde{\mathcal{A}}_t := \tilde{\mathcal{A}}_t(\tilde{\omega}) = \boldsymbol{\sigma}(X_{t_i(\tilde{\omega})} | t_i(\tilde{\omega}) \leq t)$ to distinguish this $\tilde{\omega}$-wise definition from the definition as sub-$\sigma$-algebra of the product space given above. However, Lemma B.3 implies that for our considerations, both versions of this $\sigma$-algebra have the same effect, therefore we will simply write $\mathcal{A}_t$ for both versions, by abuse of notation.*

# B  OPTIMAL APPROXIMATION $\hat{X}$ OF THE STOCHASTIC PROCESS $X$

We are interested in the "best" approximation (or prediction) $\hat{X}_t$ of the process $X$ that one can make at any time $t \in [0, T]$, given the currently available information $\mathcal{A}_t$. For us "best" refers to the $L^2(\Omega \times \tilde{\Omega}, \mathbb{F} \otimes \tilde{\mathcal{F}}, \mathbb{P} \times \tilde{\mathbb{P}})$-minimizer, therefore, this approximation is given by the conditional expectation. Indeed, if we define $\Delta := \{(t, r) \in [0, T]^2 | t + r \leq T\}$, and the function

$$\tilde{\mu} : \Delta \times \mathbb{R}^{d_X} \to \mathbb{R}^{d_X}, \quad ((t, r), \xi) \mapsto \mathbb{E}\left[\mu(t + r, X_{t+r}) | X_t = \xi\right],$$

this is proven in the following proposition.

**Proposition B.1.** *The optimal (i.e. $L^2$-norm minimizing) $\mathbb{A}$-adapted process in $L^2(\Omega \times \tilde{\Omega}, \mathbb{F} \otimes \tilde{\mathcal{F}}, \mathbb{P} \times \tilde{\mathbb{P}})$ approximating $(X_t)_{t \in [0,T]}$ is given by[3] $\hat{X} := (\hat{X}_t)_{t \in [0,T]}$ with $\hat{X}_t := \mathbb{E}_{\mathbb{P} \times \tilde{\mathbb{P}}}[X_t | \mathcal{A}_t]$. Moreover, this process is unique up to $(\mathbb{P} \times \tilde{\mathbb{P}})$-null-sets. In addition we have ($\tilde{\omega}$-wise for $\tilde{\omega} \in \tilde{\Omega}$) that*

$$\hat{X}_t = X_{\tau(t)} + \int_{\tau(t)}^t \tilde{\mu}\left(\tau(t), s - \tau(t), X_{\tau(t)}\right) ds, \tag{12}$$

*implying that $\hat{X}$ is càdlàg.*

The first part of the result follows from the elementary Proposition below (which is proven for example in (Durrett, 2010, Thm. 5.1.8) for $\mathbb{R}$-valued random variables and can easily be extended to $\mathbb{R}^d$-valued random variables when using the 2-norm).

**Proposition B.2.** *Given a probability space $(\Omega, \mathcal{F}, \mathbb{P})$ and a sub-$\sigma$-algebra $\mathcal{A} \subset \mathcal{F}$, the orthogonal projection of $X \in L^2(\Omega, \mathcal{F}, \mathbb{P})$ on $L^2(\Omega, \mathcal{A}, \mathbb{P})$ is given by $\hat{X} := \mathbb{E}[X|\mathcal{A}]$. In particular, for every $Z \in L^2(\Omega, \mathcal{A}, \mathbb{P})$ with $\mathbb{P}(Z \neq \hat{X}) > 0$ we have*

$$E[|X - Z|_2^2] = \mathbb{E}[|X - \hat{X}|_2^2] + \mathbb{E}[|Z - \hat{X}|_2^2] > \mathbb{E}[|X - \hat{X}|_2^2].$$

*Proof Proposition B.1.* In our case this means, that the optimal $\mathbb{A}$-adapted process in $L^2(\Omega \times \tilde{\Omega}, \mathbb{F} \otimes \tilde{\mathcal{F}}, \mathbb{P} \times \tilde{\mathbb{P}})$ approximating $X$ is given by $(\hat{X}_t)_{t \in [0,T]}$ with $\hat{X}_t := \mathbb{E}_{\mathbb{P} \times \tilde{\mathbb{P}}}[X_t | \mathcal{A}_t]$. Here, $\mathcal{A}_t$ is meant as a sub-$\sigma$-algebra of $\mathcal{F}_t \times \tilde{\mathcal{F}}$. This process is unique up to $(\mathbb{P} \times \tilde{\mathbb{P}})$-null-sets. Moreover, the following lemma shows that it coincides $\tilde{\omega}$-wise with $\mathbb{E}_{\mathbb{P}}[X_t | \tilde{\mathcal{A}}_t](\tilde{\omega})$, where $\tilde{\mathcal{A}}_t = \tilde{\mathcal{A}}_t(\tilde{\omega})$ is defined in Remark A.3.

**Lemma B.3.** *For $\tilde{\mathbb{P}}$-almost-every $\tilde{\omega} \in \tilde{\Omega}$ we have $\mathbb{E}_{\mathbb{P} \times \tilde{\mathbb{P}}}[X_t | \mathcal{A}_t](\tilde{\omega}) = \mathbb{E}_{\mathbb{P}}[X_t | \tilde{\mathcal{A}}_t](\tilde{\omega})$ $\mathbb{P}$-almost-surely.*

---

[3]While we give a pointwise definition, (Cohen & Elliott, 2015, Theorem 7.6.5) allows to define $\hat{X}$ directly as the optional projection. By (Cohen & Elliott, 2015, Remark 7.2.2) this implies that the process $\hat{X}$ is progressively measurable, in particular, jointly measurable in $t$ and $\omega \times \tilde{\omega}$. However, as we show below, even from the pointwise definition, it follows that $\hat{X}$ is càdlàg, hence optional (Cohen & Elliott, 2015, Theorem 7.2.7).

*Proof.* Otherwise we have by Fubini's theorem, Proposition B.2 and an argument similar to the one in Lemma E.4

$$\mathbb{E}_{\mathbb{P}\times\tilde{\mathbb{P}}}\left[\left|X_t - \mathbb{E}_{\mathbb{P}\times\tilde{\mathbb{P}}}[X_t|\mathcal{A}_t]\right|_2^2\right] = \mathbb{E}_{\tilde{\mathbb{P}}}\left[\mathbb{E}_{\mathbb{P}}\left[\left|X_t - \mathbb{E}_{\mathbb{P}\times\tilde{\mathbb{P}}}[X_t|\mathcal{A}_t]\right|_2^2\right]\right]$$
$$> \mathbb{E}_{\tilde{\mathbb{P}}}\left[\mathbb{E}_{\mathbb{P}}\left[\left|X_t - \mathbb{E}_{\mathbb{P}}[X_t|\tilde{\mathcal{A}}_t]\right|_2^2\right]\right],$$

which is a contradiction to Proposition B.2. □

This proves the first part of Proposition B.1. The second part of this Proposition, i.e. (12), should be understood $\tilde{\omega}$-wise, for $\tilde{\omega} \in \tilde{\Omega}$. This is justified by Lemma B.3 and derived below. In particular, for the remainder of this section, all statements are meant $\tilde{\omega}$-wise.

With the assumption that $\mu$ and $\sigma$ are Lipschitz, (Protter, 2005, Thm. 7, Chap. V) implies that a unique solution of (9) exists. Furthermore, this solution is a Markov process as soon as the starting point $x$ is fixed (Protter, 2005, Thm. 32, Chap. V). Hence, one can define a transition function $P$ (compare (Protter, 2005, Chap. V.6)) such that for all $s < t$ and $\phi$ bounded and measurable,

$$P_{s,t}(X_s, \phi) := \mathbb{E}_{\mathbb{P}}[\phi(X_t)|\boldsymbol{\sigma}(X_s)] = \mathbb{E}_{\mathbb{P}}[\phi(X_t)|\mathcal{F}_s].$$

We have that $X_{\tau(s)}$ is $\mathcal{A}_{\tau(s)}$-measurable and therefore, since $\mathcal{A}_s = \mathcal{A}_{\tau(s)} \subset \mathcal{F}_{\tau(s)}$,

$$P_{\tau(s),t}(X_{\tau(s)}, \phi) = \mathbb{E}_{\mathbb{P}}[\phi(X_t)|\mathcal{A}_s]. \tag{13}$$

By our additional assumption on $\sigma$ it follows from (Protter, 2005, Lem. before Thm. 28, Chap. IV) that

$$M_t := \int_0^t \sigma(s, X_s)dW_s, \quad 0 \le t \le T,$$

is a square integrable martingale, since the Browian motion $W$ is square integrable. In particular, for $0 \le s \le t \le T$ we have $\mathbb{E}_{\mathbb{P}}[\int_s^t \sigma(X_r)dW_r|\mathcal{F}_s] = \mathbb{E}_{\mathbb{P}}[M_t - M_s|\mathcal{F}_s] = 0$. Moreover, the same is true when conditioning on $\mathcal{A}_s$ [4].

Using the martingale property of $M$, we have ($\tilde{\omega}$-wise) for every $t \in [0, T]$

$$\hat{X}_t = \mathbb{E}_{\mathbb{P}}[(X_t - X_{\tau(t)}) + X_{\tau(t)}|\mathcal{A}_{\tau(t)}]$$
$$= X_{\tau(t)} + \mathbb{E}_{\mathbb{P}}\left[\int_{\tau(t)}^t \mu(r, X_r)dr\Big|\mathcal{A}_{\tau(t)}\right] + \mathbb{E}_{\mathbb{P}}\left[\int_{\tau(t)}^t \sigma(r, X_r)dW_r\Big|\mathcal{A}_{\tau(t)}\right] \tag{14}$$
$$= X_{\tau(t)} + \int_{\tau(t)}^t \mathbb{E}_{\mathbb{P}}\left[\mu(r, X_r)|\mathcal{A}_{\tau(t)}\right]dr,$$

where we used Fubini's Theorem (for conditional expectations) in the last step. This is justified because $\mathbb{E}_{\mathbb{P}}[\int_0^T |\mu(r, X_r)|_2 dr] < \infty$ follows from $\mu$ being bounded. Let us define $\Delta := \{(t, r) \in [0, T]^2 | t + r \le T\}$ and the function

$$\tilde{\mu} : \Delta \times \mathbb{R}^{d_X} \to \mathbb{R}^{d_X}, \quad ((t, r), \xi) \mapsto P_{t,t+r}(X_t, \mu)\big|_{X_t = \xi} = \mathbb{E}_{\mathbb{P}}\left[\mu(t + r, X_{t+r})|X_t = \xi\right],$$

then we can use (13) to rewrite (14) as

$$\hat{X}_t = X_{\tau(t)} + \int_{\tau(t)}^t \tilde{\mu}\left(\tau(t), s - \tau(t), X_{\tau(t)}\right)ds. \tag{15}$$

This proves the second part of Proposition B.1. □

**Proposition B.4.** *The function $\tilde{\mu}$ is (jointly) continuous.*

---

[4]To see this, we choose a localizing sequence $(\tau_n)_{n\in\mathbb{N}}$ such that $M^{\tau_n}$ is bounded by $n$ (works since $M$ is continuous). Then the Markov property implies that $\mathbb{E}[M_t^{\tau_n} - M_s^{\tau_n}|\mathcal{A}_s] = \mathbb{E}[M_t^{\tau_n} - M_s^{\tau_n}|\mathcal{F}_s] = 0$. Since $M_t^{\tau_n} \xrightarrow{n\to\infty} M_t$ $\mathbb{P}$-a.s. and since this sequence is dominated by the integrable random variable $1 + \sup_{r \le t}|M_r|_1^2$ (by Doob's inequality and square integrability of $M$), dominated convergence implies that $\mathbb{E}[M_t - M_s|\mathcal{A}_s] = 0$.

*Proof of Proposition B.4.* For any fixed $s \in [0, T], x \in \mathbb{R}^{d_X}$, we define

$$\zeta_{s,\cdot}(x) : [0, T] \times \Omega \to \mathbb{R}^{d_X}, (t, \omega) \mapsto \zeta_{s,t}(x)(\omega)$$

to be the solution of the SDE

$$\zeta_{s,t}(\xi) = \xi + \int_s^t \mu(r, \zeta_{s,r}) dr + \int_s^t \sigma(r, \zeta_{s,r}) dW_r.$$

This solution exists and is unique by (Protter, 2005, Chap. V, Thm. 7), therefore we have that $X_t = \zeta_{0,t}(x)$ $\mathbb{P}$-almost surely. Furthermore, (Gubinelli, 2016, Thm. 4) implies that for $s \le t$ we have $X_t = \zeta_{s,t}(\zeta_{0,s}(x))$. Hence, for $t = s + r$, we have the identity $\tilde{\mu}(s, r, \xi) = \mathbb{E}[\mu(t, \zeta_{s,t}(\xi))]$.

Furthermore, by (Gubinelli, 2016, Thm. 8) we have for any $\xi, \xi' \in \mathbb{R}^{d_X}$ and $(s, r), (s', r') \in \Delta$ with $t := s + r, t' := s' + r' \in [0, T]$ that there exists some constant $C$ such that

$$\mathbb{E}\left[|\zeta_{s,t}(\xi) - \zeta_{s',t'}(\xi')|_2^2\right] \le C\left[|\xi - \xi'|_2^2 + (1 + |\xi|_2 + |\xi'|_2)^2 (|t - t'| + |s - s'|)\right]. \tag{16}$$

Therefore, we have that

$$\begin{aligned}
|\tilde{\mu}(s, r, \xi) - \tilde{\mu}(s', r', \xi')|_2 &= |\mathbb{E}[\mu(t, \zeta_{s,t}(\xi))] - \mathbb{E}[\mu(t', \zeta_{s',t'}(\xi'))]|_2 \\
&\le |\mathbb{E}[\mu(t, \zeta_{s,t}(\xi))] - \mathbb{E}[\mu(t', \zeta_{s,t}(\xi))]|_2 \\
&\quad + |\mathbb{E}[\mu(t', \zeta_{s,t}(\xi))] - \mathbb{E}[\mu(t', \zeta_{s',t'}(\xi'))]|_2 \\
&\le \mathbb{E}\left[|\mu(t, \zeta_{s,t}(\xi)) - \mu(t', \zeta_{s,t}(\xi))|_2^2\right]^{1/2} \\
&\quad + \mathbb{E}\left[|\mu(t', \zeta_{s,t}(\xi)) - \mu(t', \zeta_{s',t'}(\xi'))|_2^2\right]^{1/2} \\
&\le \mathbb{E}\left[|\mu(t, \zeta_{s,t}(\xi)) - \mu(t', \zeta_{s,t}(\xi))|_2^2\right]^{1/2} \\
&\quad + \tilde{M}\,\mathbb{E}\left[|\zeta_{s,t}(\xi) - \zeta_{s',t'}(\xi')|_2^2\right]^{1/2},
\end{aligned} \tag{17}$$

where we used Jensen's inequality in the second last and (10) in the last step. Hence, for $(s', r', \xi') \to (s, r, \xi)$ we have that the first term of (17) goes to zero due to continuity of $\mu$ in its first component uniformly in the second component. Moreover, the second term of (17) converges to zero by (16). Together, this proves continuity of $\tilde{\mu}$. $\square$

## C RECALL: RNN, NEURAL ODE AND ODE-RNN - EQUIVALENT WAYS OF WRITING

We describe our model as the solution of the SDE (29), which is a compact way of writing. In the following, we first shortly recall recurrent neural networks (RNN) and the neural ODE and then recall the ODE-RNN. Furthermore, we give a step-by-step explanation, how the way ODE-RNN was formalized, can equivalently be written in terms of an SDE similar to (29). Finally we give the alternative way of writing our model and a short comparison to ODE-RNN.

**Recurrent Neural Network.** The input to a RNN is a discrete time series of observations $\{x_{t_1}, \cdots, x_{t_n}\}$. At each time $t_{i+1}$, the latent variable $h$ is updated using the previous latent variable $h_{t_i}$ and the input $x_{t_{i+1}}$ as

$$h_{t_{i+1}} := \text{RNNCell}(h_{t_i}, x_{t_{i+1}}), \tag{18}$$

where RNNCell is a neural network.

**Neural Ordinary Differential Equation.** Neural ODEs (Chen et al., 2018) are a family of continuous-time models defining a latent variable $h_t := h(t)$ to be the solution to an ODE initial-value problem (IVP):

$$h_t := h_{t_0} + \int_{t_0}^t f(h_s, s, \theta) ds, \quad t \ge t_0, \tag{19}$$

where $f(\cdot, \cdot, \theta) = f_\theta$ is a neural network with weights $\theta$. Therefore, the latent variables can be updated continuously by solving this ODE (19). We can emphasize the dependence of $h_t$ on a numerical ODE solver by rewriting (19) as

$$h_t := \text{ODESolve}(f_\theta, h_{t_0}, (t_0, t)). \tag{20}$$

**ODE-RNN.** ODE-RNN (Rubanova et al., 2019) is a mixture of a RNN and a neural ODE. In contrast to a standard RNN, we are not only interested in an output at the observation times $t_i$, but also in between those times. In particular, we want to have an output stream that is generated continuously in time. This is achieved by using a neural ODE to model the latent dynamics between two observation times, i.e. for $t_i < t < t_{i+1}$ the latent variable is defined as in (19) and (20), with $h_{t_0}$ and $t_0$ replaced by $h_{t_i}$ and $t_i$. At the next observation time $t_{i+1}$, the latent variable is then updated by a RNN. Rubanova et al. (2019) write this as

$$\begin{cases} h'_{t_{i+1}} & := & \text{ODESolve}(f_\theta, h_{t_i}, (t_i, t_{i+1})) \\ h_{t_{i+1}} & := & \text{RNNCell}(h'_{t_{i+1}}, x_{t_{i+1}}). \end{cases} \tag{21}$$

Therefore, fixing $h_{t_0} := h_0$, the entire latent process can be computed by iteratively solving an ODE followed by applying a RNN.

**GRU-ODE-Bayes.** The model architecture describing the latent variable in GRU-ODE-Bayes (Brouwer et al., 2019) is defined as the ODE-RNN but with the special choice of a gated recurrent unit (GRU) for the RNN-cell and the neural network $f_\theta$ also being derived from a GRU.

**ODE-RNN as càdlàg process.** Thinking about the process $h := (h_t)_{t \geq t_0}$ defined in (21) as a (stochastic) process in time, it is defined to evolve continuously for $t_i \leq t < t_{i+1}$ according to the ODE dynamics $f_\theta$ and jumps at time $t_{i+1}$ according to the RNN cell. In particular, it is defined to be a *càdlàg*[5] process, for which $h_{t_{i+1}-}$ is the standard notation for the left limit, i.e. the last point before the jump at time $t_{i+1}$. According to this notation we have $h_{t_{i+1}-} = h'_{t_{i+1}}$, hence, we can rewrite (21) as

$$\begin{cases} h_{t_{i+1}-} & := & \text{ODESolve}(f_\theta, h_{t_i}, (t_i, t_{i+1})) \\ h_{t_{i+1}} & := & \text{RNNCell}(h_{t_{i+1}-}, x_{t_{i+1}}). \end{cases} \tag{22}$$

## C.1 RESIDUAL ODE-RNN AS A SPECIAL CASE OF CONTROLLED ODE

**Residual ODE-RNN.** We replace the standard RNN cell by a *residual RNN cell* (rRNN), as it was described e.g. in Yue et al. (2018). In particular, instead of applying the RNN cell such that $h_{t_i} = \text{RNNCell}(h_{t_i-}, x_i)$ we use a residual RNN cell to have $h_{t_i} = h_{t_i-} + \text{rRNNCell}(h_{t_i-}, x_i)$. The residual RNN is as expressive as the standard RNN and was empirically shown to perform very similarly or even better than the standard framework (Yue et al., 2018). This way, the residual RNN cell models exactly the jump of the latent variable (i.e. the differences) that occurs at the time $t_{i+1}$ when taking into account the next observation $x_{t_{i+1}}$. Therefore, we can rewrite the ODE-RNN (22) as

$$\begin{cases} h_{t_{i+1}-} & := & \text{ODESolve}(f_\theta, h_{t_i}, (t_i, t_{i+1})) \\ h_{t_{i+1}} & := & h_{t_{i+1}-} + \text{rRNNCell}(h_{t_{i+1}-}, x_i). \end{cases} \tag{23}$$

**Controlled Ordinary Differential Equation.** We briefly recall the definition of *controlled ODEs* as it was given in (Herrera et al., 2020, Section 4.1) and used in (Cuchiero et al., 2019; Herrera et al., 2020) to describe neural networks. We fix $\ell, d \in \mathbb{N}$ and define for $1 \leq i \leq d$ the vector fields

$$V_i : \Theta \times \mathbb{R}_{\geq 0} \times \mathbb{R}^\ell \to \mathbb{R}^\ell, (\theta, t, x) \mapsto V_i^\theta(t, x),$$

which are càglàd[6] in $t$ and Lipschitz continuous in $x$. Furthermore, we define the scalar càdlàg *control* functions

$$u_i : \mathbb{R}_{\geq 0} \to \mathbb{R}, t \mapsto u_i(t),$$

which have finite variation and satisfy $u_i(0) = 0$. Then we define the process $Z := (Z_t)_{t \geq 0}$ as the solution of the controlled ODE

$$dZ_t = \sum_{i=1}^d V_i^\theta(t, Z_{t-}) du_i(t), \ Z_0 = z, \tag{24}$$

where $z \in \mathbb{R}^\ell$ is some starting point. (24) is written in Itô's differential notation for (stochastic) integrals. The solution of (24) exists and is unique under much more general assumptions than we made here (Protter, 2005, Chap. V, Thm. 7).

---

[5]i.e. right-continuous with existing left limits, also denoted as *RCLL*
[6]i.e. left continuous with existing right limits

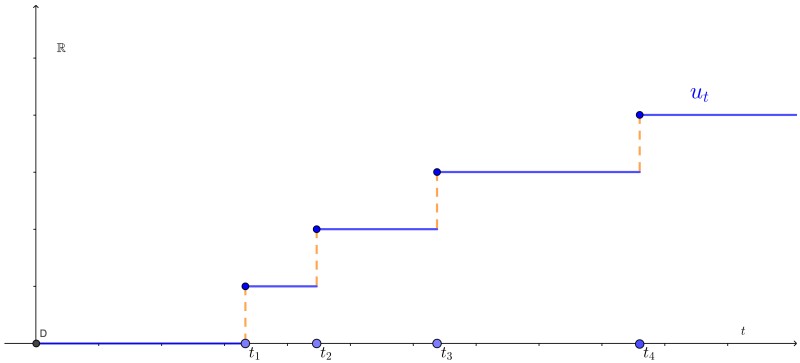

Figure 7: Schematic representation of the stochastic control $u_t$ (25).

**Special Case: ODE-RNN.** To write the ODE-RNN as controlled ODE, we set $d = 2$ and define $V_1^\theta := f_\theta$, $V_2 := \text{rRNNCell}$ and $u_1(t) := t$. As pointed out by Herrera et al. (2020), one can take the $u_i$ to be semimartingales instead of deterministic functions. In line with this, we define $u_2 := u$ as the pure jump stochastic control process

$$u : \tilde{\Omega} \times [0, T] \to \mathbb{R}, (\tilde{\omega}, t) \mapsto u_t(\tilde{\omega}) := \sum_{i=1}^{n(\tilde{\omega})} \mathbf{1}_{[t_i(\tilde{\omega}), \infty)}(t). \tag{25}$$

We note that $u$ is an adapted process starting at 0 with finite variation on the product probability space $(\Omega \times \tilde{\Omega}, \mathcal{F} \otimes \tilde{\mathcal{F}}, \mathbb{F} \otimes \tilde{\mathcal{F}}, \mathbb{P} \times \tilde{\mathbb{P}})$, since the total variation of $u$ up to time $T$ is $n$ and $\mathbb{E}_{\mathbb{P} \times \tilde{\mathbb{P}}}[n] < \infty$. The following result shows that the residual ODE-RNN can compactly be described as a controlled ODE.

**Proposition C.1.** *Using the vector fields and controls defined above, the latent variable process $h = (h_t)_{t \geq t_0}$ of the residual ODE-RNN can equivalent be written as the solution of the controlled ODE*

$$dh_t = f_\theta(h_{t-}, t-)dt + \text{rRNNCell}(h_{t-}, x_t)du(t), \quad h_{t_0} = h_0. \tag{26}$$

*Proof of Proposition C.1.* By (Protter, 2005, Chap. II, Thm. 17), the stochastic integral is indistinguishable from the path-wise Lebesgue-Stieltjes integral if the integrator is of finite variation. Hence, we can assume that some $\tilde{\omega} \in \tilde{\Omega}$ is fixed and that the following expressions are evaluated at this $\tilde{\omega}$ whenever applicable. First note, that $u$ is constant except at the $t_i$ where it increases by 1 (cf. Figure 7). In particular, the Lebesgue-Stieltjes integral of some càdlàg function $g$ with respect to $u$ is a sum, i.e. $\int_0^t g(s-)du_s = \sum_{t_i \leq t} g(t_i-)\Delta u_{t_i}$, where $\Delta u_{t_i} = u_{t_i} - u_{t_i-} = 1$. Therefore, integrating (26) from $t_0$ to $t$ we get

$$h_t = h_{t_0} + \int_{t_0}^t f_\theta(h_{s-}, s-)ds + \int_{t_0}^t \text{rRNNCell}(h_{s-}, x_s)du(s)$$

$$= h_{t_0} + \int_{t_0}^t f_\theta(h_{s-}, s-)ds + \sum_{t_i \leq t} \text{rRNNCell}(h_{t_i-}, x_{t_i}).$$

In particular, since the first integral is continuous in $t$, we have for every $1 \leq k \leq n$

$$h_{t_k} = h_{t_0} + \int_{t_0}^{t_k-} f_\theta(h_{s-}, s-)ds + \sum_{t_i < t_k} \text{rRNNCell}(h_{t_i-}, x_{t_i}) + \text{rRNNCell}(h_{t_k-}, x_{t_k})$$

$$= h_{t_k-} + \text{rRNNCell}(h_{t_k-}, x_{t_k}), \tag{27}$$

and for $t_k < t < t_{k+1}$

$$h_t = h_{t_0} + \int_{t_0}^{t_k} f_\theta(h_{s-}, s-)ds + \sum_{t_i \leq t_k} \text{rRNNCell}(h_{t_i-}, x_{t_i}) + \int_{t_k}^t f_\theta(h_{s-}, s-)ds$$

$$= h_{t_k} + \int_{t_k}^t f_\theta(h_{s-}, s-)ds. \tag{28}$$

Together, (27) and (28) prove that (26) is equivalent to (23). We also emphasize that $x_t$ is used as input only for $t = t_i$, $1 \leq i \leq n$. $\qquad\qquad\qquad\qquad\qquad\qquad\qquad\qquad\qquad\qquad\qquad\square$

## D  NEURAL JUMP ODE

We propose a model framework whose architecture can be interpreted as a simplification of the ODE-RNN (Rubanova et al., 2019) and the GRU-ODE-Bayes (Brouwer et al., 2019) model architecture.

### D.1  THE MODEL FRAMEWORK: NEURAL ODE BETWEEN JUMPS

We define $\mathcal{X} \subset \mathbb{R}^{d_X}$ and $\mathcal{H} \subset \mathbb{R}^{d_H}$ to be the observation and latent space for $d_X, d_H \in \mathbb{N}$. Moreover, we define the following feed-forward neural networks with sigmoid activation functions:

- $f_{\theta_1} : \mathbb{R}^{d_H} \times \mathbb{R}^{d_X} \times [0, T] \times [0, T] \to \mathbb{R}^{d_H}$ modelling the ODE dynamics,
- $\rho_{\theta_2} : \mathbb{R}^{d_X} \to \mathbb{R}^{d_H}$ modelling the jumps when new observations are made, and
- $g_{\theta_3} : \mathbb{R}^{d_H} \to \mathbb{R}^{d_Y}$ the readout map, mapping into the target space $\mathcal{Y} \subset \mathbb{R}^{d_Y}$ for $d_Y \in \mathbb{N}$.

The trainable parameters of the neural networks are $\theta := (\theta_1, \theta_2, \theta_3) \in \Theta := \cup_{M \geq 1} \Theta_M$. Here, for every $M \in \mathbb{N}$, $\Theta_M$ is defined to be the set of all parameters such that $f_{\theta_1}$ and $\rho_{\theta_2}$ have hidden dimension $M$. We define the pure jump stochastic process (cf. Appendix C)

$$u : \tilde{\Omega} \times [0, T] \to \mathbb{R}, (\tilde{\omega}, t) \mapsto u_t(\tilde{\omega}) := \sum_{i=1}^{n(\tilde{\omega})} \mathbf{1}_{[t_i(\tilde{\omega}), \infty)}(t).$$

Then the *Neural Jump ODE* (NJ-ODE) is defined by the latent process $H := (H_t)_{t \in [0,T]}$ and the output process $Y := (Y_t)_{t \in [0,T]}$ defined as solutions of the SDE system (cf. Appendix C)

$$\begin{aligned}
H_0 &= \rho_{\theta_2}(X_0), \\
dH_t &= f_{\theta_1}\left(H_{t-}, X_{\tau(t)}, \tau(t), t - \tau(t)\right) dt + \left(\rho_{\theta_2}(X_t) - H_{t-}\right) du_t, \\
Y_t &= g_{\theta_3}(H_t).
\end{aligned} \tag{29}$$

Note that, only the values of $X$ at the times $t_i, 0 \leq i \leq n$ are used as inputs. We will use the notation $H_t^\theta(X)$ and $Y_t^\theta(X)$ to emphasize the dependence of the latent process $H$ and the output process $Y$ on the model parameters $\theta$ and the input $X$.

**Existence and uniqueness.** We note that $u$ is an adapted process starting at 0 with finite variation on the product probability space $(\Omega \times \tilde{\Omega}, \mathcal{F} \otimes \tilde{\mathcal{F}}, \mathbb{F} \otimes \tilde{\mathcal{F}}, \mathbb{P} \times \tilde{\mathbb{P}})$, since the total variation of $u$ up to time $T$ is $n$ and $\mathbb{E}_{\mathbb{P} \times \tilde{\mathbb{P}}}[n] < \infty$. Hence, since $f_{\theta_1}$ and $\rho_{\theta_2}$ are Lipschitz continuous (as composition of Lipschitz continuous functions) a unique càdlàg solution $H^\theta$ exists, once an initial value is fixed (Protter, 2005, Thm. 7, Chap. V). Moreover, the resulting process $(Y_t^\theta)_{t \in [0,T]}$ is also càdlàg and $\mathbb{A}$-adapted.

**Equivalent way of writing NJ-ODE.** By applying the steps that were explained in Section C backwards to (29) (but without introducing the residual version of the RNN), it is easy to see that our model can equivalently be written (similar to (22)) as

$$\begin{cases}
h_{t_{i+1}-} &:= \text{ODESolve}(f_{\theta_1}, (h_t, x_{t_i}, t_i, t - t_i), (t_i, t_{i+1})) \\
h_{t_{i+1}} &:= \rho_{\theta_2}(x_{t_{i+1}}).
\end{cases} \tag{30}$$

In particular, the main difference to (22) is that we use a modified input for $f_{\theta_1}$ and that the neural network performing the jumps does not take $h_{t_{i+1}-}$ as an input, i.e. it is not a RNN.

### D.2  OBJECTIVE FUNCTION

We introduce a loss function, such that the output $Y^\theta$ of the NJ-ODE can be trained to approximate $\hat{X}$, i.e. to model the best *online* prediction of the stochastic process $X$. We define the theoretical loss function and its Monte Carlo and Ergodic approximation.

**Objective function.** Let us define $\mathbb{D}$ to be the set of all $\mathbb{R}^{d_X}$-valued $\mathbb{A}$-adapted processes on the probability space $(\Omega \times \tilde{\Omega}, \mathcal{F} \otimes \tilde{\mathcal{F}}, \mathbb{F} \otimes \tilde{\mathcal{F}}, \mathbb{P} \times \tilde{\mathbb{P}})$. Then we define our objective functions

$$\Psi : \mathbb{D} \to \mathbb{R}, \qquad Z \mapsto \Psi(Z) := \mathbb{E}_{\mathbb{P} \times \tilde{\mathbb{P}}} \left[ \frac{1}{n} \sum_{i=1}^{n} \left( |X_{t_i} - Z_{t_i}|_2 + |Z_{t_i} - Z_{t_i-}|_2 \right)^2 \right], \qquad (31)$$

$$\Phi : \Theta \to \mathbb{R}, \qquad \theta \mapsto \Phi(\theta) := \Psi(Y^\theta(X)), \qquad (32)$$

where $\Phi$ will be our (theoretical) loss function. Remark that from the definition of $Y^\theta$, it directly follows that it is an element of $\mathbb{D}$, hence $\Phi$ is well-defined.

**Monte Carlo approximation of the objective function.** Let us assume, that we observe $N \in \mathbb{N}$ independent realisations of the path $X$ at times $(\tilde{t}_1^{(j)}, \cdots, \tilde{t}_{n^j}^{(j)})$, $1 \leq j \leq N$, which are themselves independent realisations of the random vector $(n, t_1, \cdots, t_n)$. In particular, let us assume that $X^{(j)} \sim X$ and $(n^j, t_1^{(j)}, \cdots, t_{n^j}^{(j)}) \sim \mathbb{P}_t$ are i.i.d. random processes (respectively variables) for $1 \leq j \leq N$ and that our training data is one realisation of them. Then

$$\hat{\Phi}_N(\theta) := \frac{1}{N} \sum_{j=1}^{N} \frac{1}{n^j} \sum_{i=1}^{n^j} \left( \left| X_{t_i^{(j)}}^{(j)} - Y_{t_i^{(j)}}^\theta(X^{(j)}) \right|_2 + \left| Y_{t_i^{(j)}}^\theta(X^{(j)}) - Y_{t_i^{(j)}-}^\theta(X^{(j)}) \right|_2 \right)^2, \qquad (33)$$

converges $(\mathbb{P} \times \tilde{\mathbb{P}})$-a.s. to $\Phi(\theta)$ as $N \to \infty$, by the law of large numbers (cf. Theorem E.13).

**Ergodic approximation of the objective function.** If we only observe one realization of the path $X$ at times $(\tilde{t}_1, \cdots, \tilde{t}_N)$, we can still approximate the objective function by assuming that $\mu$ and $\sigma$ are time-independent and that the stochastic process $X$ is ergodic in the following sense. We fix $n = 1$ and assume that the time increments $\Delta \tilde{t}_j := \tilde{t}_j - \tilde{t}_{j-1}$ are i.i.d. realizations of the probability distribution $\lambda_1$. Furthermore, we consider each observation $X_{\tilde{t}_j}$ as one sample with initial condition $X_{\tilde{t}_{j-1}}$ for which $Y^{\theta,j}$ is the realization of $Y^\theta$. Then we approximate the objective function by

$$\hat{\Phi}_N(\theta) := \frac{1}{N} \sum_{j=1}^{N} \left( \left| X_{\tilde{t}_j} - Y_{\Delta \tilde{t}_j}^\theta(X) \right|_2 + \left| Y_{\Delta \tilde{t}_j}^\theta(X) - Y_{\Delta \tilde{t}_j-}^\theta(X) \right|_2 \right)^2, \qquad (34)$$

which is assumed to converge by the ergodicity assumption for $N \to \infty$ to

$$\mathbb{E}_{\mathbb{P} \times \tilde{\mathbb{P}}} \left[ \left( |X_{t_1} - Y_{t_1}^\theta|_2 + |Y_{t_1}^\theta - Y_{t_1-}^\theta|_2 \right)^2 \right]. \qquad (35)$$

Instead of setting the random variable $n = 1$, one could similarly fix the time horizon $T$, and take for each sample all subsequent observations that lie in the time interval $[t_{\text{start}}, t_{\text{start}} + T]$, where $t_{\text{start}}$ is the date of the first observation of this sample. The next sample would then start with the first observation after $t_{\text{start}} + T$.

# E    THEORETICAL CONVERGENCE RESULTS

Our main results show, that the output of NJ-ODE converges to the conditional expectation when the size of the neural networks and the number of samples go to infinity. For each network size (and number of samples) we assume to have the weights minimizing the (Monte Carlo approximation of the) loss function. In particular, we do not consider the problem of finding those optimal weights, and therefore also do not analyse backpropagation through our model. In a similar setting, backpropagation was studied by Jia & Benson (2019).

For completeness we recall the definition of $L^p$-convergence.

**Definition E.1.** *Let $1 \leq p < \infty$. Let $(\Omega, \mathcal{F}, \mathbb{P})$ be a probability space. Then a sequence of random variables $(X_n)_{n \in \mathbb{N}}$ converges to a random variable $X$ in $L^p(\Omega, \mathcal{F}, \mathbb{P})$ (or simply $L^p$), if*

$$\mathbb{E}[|X_n - X|^p] \xrightarrow{n \to \infty} 0.$$

*We use the notation*

$$X_n \xrightarrow{L^p} X.$$

*$L^1$-convergence is also denoted* convergence in mean.

### E.1 Convergence with respect to theoretical objective function

**Theorem E.2.** *Let $\theta_M^{\min} \in \Theta_M^{\min} := \mathrm{argmin}_{\theta \in \Theta_M}\{\Phi(\theta)\}$ for every $M \in \mathbb{N}$. Then, for $M \to \infty$, the value of the loss function $\Phi$ (32) converges to the minimal value of $\Psi$ (31) which is uniquely achieved by $\hat{X}$, i.e.*

$$\Phi(\theta_M^{\min}) \xrightarrow{M \to \infty} \min_{Z \in \mathbb{D}} \Psi(Z) = \Psi(\hat{X}).$$

*Furthermore, for every $1 \leq k \leq K$ we have that $Y^{\theta_M^{\min}}$ converges to $\hat{X}$ as random variable in $L^1(\Omega \times [0, T], \mathbb{P} \times \lambda_k)$. In particular, the limit process $Y := \lim_{M \to \infty} Y^{\theta_M^{\min}}$ equals $\hat{X}$ ($\mathbb{P} \times \lambda_k$)- almost surely as a random variable on $\Omega \times [0, T]$.*

The idea of the proof is to split up the target jump process into its continuous-in-time parts and into the jumps. Both parts are continuous functions of their inputs and can therefore be approximated by neural networks (Hornik et al., 1989). More precisely, by Proposition B.1 and B.4 the continuous-in-time part can be written as an integral over time of a function which is jointly continuous in its inputs. This jointly continuous function is approximated by a neural network, which is itself integrated over time – that is the neural ODE part of the NJ-ODE. The remainder of the proof shows $L^1$-convergence of the output of NJ-ODE with optimal weights (with respect to the loss function) to the conditional expectation process.

For completeness we restate the straight forward generalization of the universal approximation result (Hornik et al., 1989, Thm. 2.4) to multidimensional output.

**Theorem E.3** (Hornik). *Let $r, d \in \mathbb{N}$ and $\sigma$ be a sigmoid function. Let $\mathcal{N}\mathcal{N}_{r,d}^{\sigma}$ be the set of all 1-hidden layer neural networks mapping $\mathbb{R}^r$ to $\mathbb{R}^d$. Then for every compact subset $K \subset \mathbb{R}^r$, every $\epsilon > 0$ and every $f \in C(\mathbb{R}^r, \mathbb{R}^d)$ there exists a neural network $g \in \mathcal{N}\mathcal{N}_{r,d}^{\sigma}$ such that $\sup_{x \in K}|f(x) - g(x)|_2 < \epsilon$.*

The following Lemmas are used in the Proof of Theorem E.2.

**Lemma E.4.** *Let $1 \leq k \leq K$ and let $Z \in \mathbb{D}$ be a process such that $(\mathbb{P} \times \lambda_k)[\hat{X} \neq Z] > 0$. Then there exists an $\varepsilon > 0$ such that $\tilde{B} := \{t \in [0, T] \mid \mathbb{E}_{\mathbb{P}}[|X_t - Z_{t-}|_2^2] \geq \varepsilon + \mathbb{E}_{\mathbb{P}}[|X_t - \hat{X}_{t-}|_2^2]\}$ satisfies $\lambda_k(\tilde{B}) > 0$.*

*Proof.* First remark that since $X$ is continuous, we have $X_t = X_{t-}$. Let us define $C := \{(\omega, t) \in \Omega \times [0, T] \mid \hat{X}_{t-}(\omega) \neq Z_{t-}(\omega)\}$ and for each $t \in [0, T]$ let $C_t := \{\omega \in \Omega \mid (\omega, t) \in C\}$. Then we have for $B := \{t \in [0, T] \mid \mathbb{P}(C_t) > 0\}$ that $\lambda_k(B) > 0$, since otherwise by Fubini's theorem

$$0 < (\mathbb{P} \times \lambda_k)[C] = \int_{[0,T]} \mathbb{P}(C_t) d\lambda_k(t) = 0,$$

which is a contradiction. Now Proposition B.2 yields that for each $t \in B$ there exists some $\varepsilon_t > 0$ such that $\mathbb{E}_{\mathbb{P}}[|X_{t-} - Z_{t-}|_2^2] \geq \varepsilon_t + \mathbb{E}_{\mathbb{P}}[|X_{t-} - \hat{X}_{t-}|_2^2]$. This implies the claim. Indeed, assume no such $\varepsilon > 0$ exists, then we have for each $n \in \mathbb{N}$ that $\lambda_k\left(\{t \in [0, T] \mid \mathbb{E}_{\mathbb{P}}[|X_{t-} - Z_{t-}|_2^2] \geq \frac{1}{n} + \mathbb{E}_{\mathbb{P}}[|X_{t-} - \hat{X}_{t-}|_2^2]\}\right) = 0$. Therefore,

$$\lambda_k\left(\{t \in [0, T] \mid \mathbb{E}_{\mathbb{P}}[|X_{t-} - Z_{t-}|_2^2] > \mathbb{E}_{\mathbb{P}}[|X_{t-} - \hat{X}_{t-}|_2^2]\}\right)$$
$$\leq \sum_{n \in \mathbb{N}} \lambda_k\left(\{t \in [0, T] \mid \mathbb{E}_{\mathbb{P}}[|X_{t-} - Z_{t-}|_2^2] \geq \frac{1}{n} + \mathbb{E}_{\mathbb{P}}[|X_{t-} - \hat{X}_{t-}|_2^2]\}\right) = 0,$$

which is a contradiction to $\lambda_k(B) > 0$. □

**Lemma E.5.** *For any $\mathbb{A}$-adapted process $Z$ it holds that*

$$\mathbb{E}_{\mathbb{P} \times \tilde{\mathbb{P}}}\left[\frac{1}{n}\sum_{i=1}^{n}|X_{t_i} - Z_{t_i-}|_2^2\right] = \mathbb{E}_{\mathbb{P} \times \tilde{\mathbb{P}}}\left[\frac{1}{n}\sum_{i=1}^{n}\left|X_{t_i} - \hat{X}_{t_i-}\right|_2^2\right] + \mathbb{E}_{\mathbb{P} \times \tilde{\mathbb{P}}}\left[\frac{1}{n}\sum_{i=1}^{n}\left|\hat{X}_{t_i-} - Z_{t_i-}\right|_2^2\right].$$

*Proof.* First recall that by continuity $X_{t_i} = X_{t_i-}$. Then the statement is a consequence of Proposition B.2, Lemma B.3 and Fubini's theorem, which imply

$$
\begin{aligned}
\mathbb{E}_{\mathbb{P}\times\tilde{\mathbb{P}}}\left[\frac{1}{n}\sum_{i=1}^{n}|X_{t_i-}-Z_{t_i-}|_2^2\right] &= \mathbb{E}_{\tilde{\mathbb{P}}}\left[\frac{1}{n}\sum_{i=1}^{n}\mathbb{E}_{\mathbb{P}}\left[|X_{t_i-}-Z_{t_i-}|_2^2\right]\right]\\
&= \mathbb{E}_{\tilde{\mathbb{P}}}\left[\frac{1}{n}\sum_{i=1}^{n}\left(\mathbb{E}_{\mathbb{P}}\left[\left|X_{t_i-}-\hat{X}_{t_i-}\right|_2^2\right]+\mathbb{E}_{\mathbb{P}}\left[\left|\hat{X}_{t_i-}-Z_{t_i-}\right|_2^2\right]\right)\right]\\
&= \mathbb{E}_{\mathbb{P}\times\tilde{\mathbb{P}}}\left[\frac{1}{n}\sum_{i=1}^{n}\left|X_{t_i-}-\hat{X}_{t_i-}\right|_2^2\right]+\mathbb{E}_{\mathbb{P}\times\tilde{\mathbb{P}}}\left[\frac{1}{n}\sum_{i=1}^{n}\left|\hat{X}_{t_i-}-Z_{t_i-}\right|_2^2\right].
\end{aligned}
$$

$\square$

**Lemma E.6.** *Let $1 \le p < \infty$. Let $(Z_n)_{n\in\mathbb{N}}$ be a sequence of random variables, and $Z$ and $\tilde{Z}$ random variables defined on a common probability space such that $Z_n \xrightarrow{L^p} Z$ and $Z_n \xrightarrow{L^p} \tilde{Z}$. Then $Z = \tilde{Z}$ almost surely.*

*Proof.* With the triangle inequality it follows that $\mathbb{E}[|Z-\tilde{Z}|^p]^{1/p} = 0$, which implies the claim. $\square$

**Lemma E.7.** *The random variable $S_T := \sup_{0\le t\le T}|X_t|_1$ is square integrable and bounded in probability, i.e. for any $\varepsilon > 0$ exist some $K > 0$ such that $\mathbb{P}[S_T > K] \le \varepsilon$.*

*Proof.* From the proof of Proposition B.1 we know that

$$
M_t := \int_0^t \sigma(s, X_s)dW_s, \quad 0 \le t \le T,
$$

is a square integrable martingale and that $\mathbb{E}_{\mathbb{P}}[\sup_{0\le t\le T}|M_t|_1^j] \le c\,\mathbb{E}_{\mathbb{P}}[|M_T|_1^j] < \infty$, for $j = 1, 2$ and some constant $c > 0$ by Doob's inequality. Moreover, $\mu$ is bounded, say by $B$, hence

$$
\begin{aligned}
|X_t|_1 &= \left|x + \int_0^t \mu(r, X_r)dr + \int_0^t \sigma(r, X_r)dW_r\right|_1\\
&\le |x|_1 + \int_0^t |\mu(r, X_r)|_1\,dr + |M_t|_1\\
&\le |x|_1 + B\,T + |M_t|_1.
\end{aligned}
$$

Therefore, $\mathbb{E}_{\mathbb{P}}[S_T] \le |x|_1 + B\,T + c\,\mathbb{E}_{\mathbb{P}}[|M_T|_1] < \infty$ and similar $\mathbb{E}_{\mathbb{P}}[S_T^2] < \infty$, which implies the claim. Indeed, if for a fixed $\varepsilon > 0$ no such $K$ exists, then $\mathbb{P}[S_T = \infty] \ge \varepsilon$, which is a contradiction to integrability. $\square$

*Proof of Theorem E.2.* We start by showing that $\hat{X} \in \mathbb{D}$ is the unique minimizer of $\Psi$ up to $(\mathbb{P} \times \lambda_k)$-null-sets for any $k \le K$. First, we recall that for every $t_i$ we have $\hat{X}_{t_i} = X_{t_i}$ and by continuity of $X$ that $X_{t_i} = X_{t_i-}$. Therefore,

$$
\begin{aligned}
\Psi(\hat{X}) &= \mathbb{E}_{\mathbb{P}\times\tilde{\mathbb{P}}}\left[\frac{1}{n}\sum_{i=1}^{n}\left|X_{t_i}-\hat{X}_{t_i-}\right|_2^2\right]\\
&= \mathbb{E}_{\tilde{\mathbb{P}}}\left[\frac{1}{n}\sum_{i=1}^{n}\mathbb{E}_{\mathbb{P}}\left[\left|X_{t_i}-\hat{X}_{t_i-}\right|_2^2\right]\right]\\
&= \min_{Z\in\mathbb{D}}\mathbb{E}_{\mathbb{P}\times\tilde{\mathbb{P}}}\left[\frac{1}{n}\sum_{i=1}^{n}|X_{t_i}-Z_{t_i-}|_2^2\right]\\
&\le \min_{Z\in\mathbb{D}}\mathbb{E}_{\mathbb{P}\times\tilde{\mathbb{P}}}\left[\frac{1}{n}\sum_{i=1}^{n}(|X_{t_i}-Z_{t_i}|_2+|Z_{t_i}-Z_{t_i-}|_2)^2\right]\\
&= \min_{Z\in\mathbb{D}}\Psi(Z),
\end{aligned}
$$

where we used Fubini's theorem for the second line, Proposition B.2 for the third line and the triangle inequality for the fourth line. Hence, $\hat{X}$ is a minimizer of $\Psi$. To see that it is unique $(\mathbb{P} \times \lambda_k)$-a.s., let $Z \in \mathbb{D}$ be a process such that $(\mathbb{P} \times \lambda_k)[\hat{X} \neq Z] > 0$. By Lemma E.4, this implies that there exists an $\varepsilon > 0$ such that $B := \{t \in [0,T] \,|\, \mathbb{E}_{\mathbb{P}}[|X_{t-} - Z_{t-}|_2^2] \geq \epsilon + \mathbb{E}_{\mathbb{P}}[|X_{t-} - \hat{X}_{t-}|_2^2]\}$ satisfies $\lambda_k(B) > 0$.

Now recall that by definition of $\lambda_k$ we have $\lambda_k(B) = \tilde{\mathbb{P}}(\dot{\cup}_{j \geq k}\{n = j, t_k- \in B\})/\tilde{\mathbb{P}}(n \geq k) > 0$. This implies that there exists $j \in \mathbb{N}_{\geq k}$ such that $\tilde{\mathbb{P}}(n = j, t_k- \in B) > 0$. Therefore,

$$\mathbb{E}_{\tilde{\mathbb{P}}}\left[\frac{1}{n}\sum_{i=1}^{n}\mathbf{1}_{\{t_i-\in B\}}\right] \geq \mathbb{E}_{\tilde{\mathbb{P}}}\left[\mathbf{1}_{\{n=j\}}\frac{1}{n}\sum_{i=1}^{n}\mathbf{1}_{\{t_i-\in B\}}\right]$$
$$\geq \mathbb{E}_{\tilde{\mathbb{P}}}\left[\mathbf{1}_{\{n=j\}}\tfrac{1}{j}\mathbf{1}_{\{t_k-\in B\}}\right]$$
$$= \tfrac{1}{j}\tilde{\mathbb{P}}(n = j, t_k- \in B) > 0.$$

This inequality implies now that $Z$ is not a minimizer of $\Psi$, because

$$\Psi(Z) = \mathbb{E}_{\mathbb{P}\times\tilde{\mathbb{P}}}\left[\frac{1}{n}\sum_{i=1}^{n}(|X_{t_i} - Z_{t_i}|_2 + |Z_{t_i} - Z_{t_i-}|_2)^2\right]$$
$$\geq \mathbb{E}_{\tilde{\mathbb{P}}}\left[\frac{1}{n}\sum_{i=1}^{n}\mathbb{E}_{\mathbb{P}}\left[|X_{t_i} - Z_{t_i-}|_2^2\right]\right]$$
$$= \mathbb{E}_{\tilde{\mathbb{P}}}\left[\frac{1}{n}\sum_{i=1}^{n}\left(\mathbf{1}_{\{t_i-\in B\}} + \mathbf{1}_{\{t_i-\in B^C\}}\right)\mathbb{E}_{\mathbb{P}}\left[|X_{t_i} - Z_{t_i-}|_2^2\right]\right]$$
$$\geq \mathbb{E}_{\tilde{\mathbb{P}}}\left[\frac{1}{n}\sum_{i=1}^{n}\left(\varepsilon\mathbf{1}_{\{t_i-\in B\}} + \mathbb{E}_{\mathbb{P}}\left[\left|X_{t_i} - \hat{X}_{t_i-}\right|_2^2\right]\right)\right]$$
$$= \varepsilon\mathbb{E}_{\tilde{\mathbb{P}}}\left[\frac{1}{n}\sum_{i=1}^{n}\mathbf{1}_{\{t_i-\in B\}}\right] + \min_{Z\in\mathbb{D}}\Psi(Z)$$
$$> \min_{Z\in\mathbb{D}}\Psi(Z).$$

Next we show that (29) can approximate $\hat{X}$ arbitrarily well. Since the dimension $d_H$ can be chosen freely, let us fix it to $d_H := d_X$. Furthermore, let us fix $\theta_2^*$ and $\theta_3^*$ such that $\rho_{\theta_2^*} = g_{\theta_3^*} = \text{id}$. From Theorem E.3 it follows that for any $\varepsilon > 0$ there exist $M \in \mathbb{N}$ and $\theta_1^*$ with $(\theta_1^*, \theta_2^*, \theta_3^*) \in \Theta_M$ such that

$$\sup_{(u,v,t,r)\in[-M,M]^{d_H\times d_X}\times\Delta}\left|f_{\theta_1^*}(u,v,t,r) - \tilde{\mu}(t,r,v)\right|_2 \leq \varepsilon, \tag{36}$$

where we used that $\tilde{\mu}$ is continuous by Proposition B.4. Since $\mu$ is bounded, also $\tilde{\mu}$ is bounded, say by $B - 1 > 0$. On $[-M,M]^{d_X}$ we approximate $\tilde{\mu}$ by the neural network $f_{\theta_1^*}$ and outside we continuously extend $f_{\theta_1^*}$ such that it is bounded by $B$. By abuse of notation we call this $f_{\theta_1^*}$ again and use it as our neural network. Hence, $\left|f_{\theta_1^*} - \tilde{\mu}\right|_2 \leq 2B$. Using this, we can bound the distance between $Y_t^{\theta_M^*}$ and $\hat{X}$. In particular, if $t \in \{t_1, \cdots, t_n\}$, we have

$$\left|Y_t^{\theta_M^*} - \hat{X}_t\right|_2 = \left|(H_{t-} + (\rho_{\theta_M^*}(X_t) - H_{t-})) - X_t\right|_2 = 0,$$

and if $t$ not in $\{t_1, \cdots, t_n\}$, then (12), (36) and the previous bound yield for $S_T := \sup_{0\leq t\leq T}|X_t|_1$

$$\left|Y_t^{\theta_M^*} - \hat{X}_t\right|_1 \leq \left|Y_{\tau(t)}^{\theta_M^*} - \hat{X}_{\tau(t)}\right|_1$$
$$+ \int_{\tau(t)}^{t}\left|f_{\theta_1^*}\left(H_{s-}, X_{\tau(s)}, \tau(s), s - \tau(s)\right) - \tilde{\mu}\left(\tau(s), s - \tau(s), X_{\tau(s)}\right)\right|_1 ds$$
$$\leq \varepsilon T\mathbf{1}_{\{S_T\leq M\}} + 2BT\mathbf{1}_{\{S_T>M\}} \leq \varepsilon T + 2BT\mathbf{1}_{\{S_T>M\}}.$$

Here we used that $S_T \leq M$ implies that (36) can be used for all $\tau(t) \leq s \leq t$. Moreover, by equivalence of the 1- and 2-norm, there exists a constant $c > 0$ such that

$$\left| Y_t^{\theta_M^*} - \hat{X}_t \right|_2 \leq c \varepsilon T + 2cBT\mathbf{1}_{\{S_T > M\}}.$$

With Lemma E.7 we know that $\mathbb{E}_\mathbb{P}[\mathbf{1}_{\{S_T > M\}}] = \mathbb{P}[S_T > M] =: \epsilon_M \xrightarrow{M \to \infty} 0$. Now we can show the convergence of $\Phi(\theta_M^{\min})$ using these two bounds and $X_{t_i} = \hat{X}_{t_i}$. Indeed,

$$\min_{Z \in \mathbb{D}} \Psi(Z) \leq \Phi(\theta_M^{\min}) \leq \Phi(\theta_M^*)$$

$$= \mathbb{E}_{\mathbb{P} \times \tilde{\mathbb{P}}} \left[ \frac{1}{n} \sum_{i=1}^n \left( \left| X_{t_i} - Y_{t_i}^{\theta_M^*} \right|_2 + \left| Y_{t_i}^{\theta_M^*} - Y_{t_i-}^{\theta_M^*} \right|_2 \right)^2 \right]$$

$$\leq \mathbb{E}_{\mathbb{P} \times \tilde{\mathbb{P}}} \left[ \frac{1}{n} \sum_{i=1}^n \left( \left| \hat{X}_{t_i} - Y_{t_i}^{\theta_M^*} \right|_2 + \left| Y_{t_i}^{\theta_M^*} - \hat{X}_{t_i} \right|_2 + \left| \hat{X}_{t_i} - \hat{X}_{t_i-} \right|_2 + \left| \hat{X}_{t_i-} - Y_{t_i-}^{\theta_M^*} \right|_2 \right)^2 \right]$$

$$\leq \mathbb{E}_{\mathbb{P} \times \tilde{\mathbb{P}}} \left[ \frac{1}{n} \sum_{i=1}^n \left( \left| X_{t_i} - \hat{X}_{t_i-} \right|_2 + c(\varepsilon T + 2BT\mathbf{1}_{\{S_T > M\}}) \right)^2 \right]$$

$$\leq \mathbb{E}_{\tilde{\mathbb{P}}} \left[ \frac{1}{n} \sum_{i=1}^n \mathbb{E}_\mathbb{P} \left[ \left( \left| X_{t_i} - \hat{X}_{t_i-} \right|_2 + c(\varepsilon T + 2BT\mathbf{1}_{\{S_T > M\}}) \right)^2 \right] \right]$$

$$\leq \mathbb{E}_{\tilde{\mathbb{P}}} \left[ \frac{1}{n} \sum_{i=1}^n \left( \mathbb{E}_\mathbb{P} \left[ \left| X_{t_i} - \hat{X}_{t_i-} \right|_2^2 \right]^{1/2} + \mathbb{E}_\mathbb{P} \left[ (c \varepsilon T + 2cBT\mathbf{1}_{\{S_T > M\}})^2 \right]^{1/2} \right)^2 \right],$$

where we used the triangle-inequality for the $L^2$-norm in the last step. We can bound

$$\mathbb{E}_\mathbb{P} \left[ (c \varepsilon T + 2cBT\mathbf{1}_{\{S_T > M\}})^2 \right] \leq 3(c \varepsilon T)^2 + 3(2cBT)^2 \epsilon_M =: c_M^2,$$

which is a constant converge to 0 as $M \to \infty$. Using that for $a \in \mathbb{R}$, we have $a \leq a^2 + 1$, we get

$$\min_{Z \in \mathbb{D}} \Psi(Z) \leq \Phi(\theta_M^{\min}) \leq \Phi(\theta_M^*)$$

$$\leq \mathbb{E}_{\tilde{\mathbb{P}}} \left[ \frac{1}{n} \sum_{i=1}^n \left( \mathbb{E}_\mathbb{P} \left[ \left| X_{t_i} - \hat{X}_{t_i-} \right|_2^2 \right]^{1/2} + c(M) \right)^2 \right]$$

$$\leq \mathbb{E}_{\tilde{\mathbb{P}}} \left[ \frac{1}{n} \sum_{i=1}^n \left( \mathbb{E}_\mathbb{P} \left[ \left| X_{t_i} - \hat{X}_{t_i-} \right|_2^2 \right] + 2c_M \mathbb{E}_\mathbb{P} \left[ \left| X_{t_i} - \hat{X}_{t_i-} \right|_2^2 \right]^{1/2} + c_M^2 \right) \right]$$

$$\leq (1 + 2c_M)\mathbb{E}_{\mathbb{P} \times \tilde{\mathbb{P}}} \left[ \frac{1}{n} \sum_{i=1}^n \left| X_{t_i} - \hat{X}_{t_i-} \right|_2^2 \right] + 2c_M + c_M^2$$

$$\leq (1 + 2c_M) \min_{Z \in \mathbb{D}} \Psi(Z) + 2c_M + c_M^2 \xrightarrow{M \to \infty} \min_{Z \in \mathbb{D}} \Psi(Z),$$

where we used $\Psi(\hat{X}) = \min_{Z \in \mathbb{D}} \Psi(Z)$ and that $c_M$ converges to 0.

In the last step we show that the limits $\lim_{M \to \infty} Y^{\theta_M^{\min}}$ and $\lim_{M \to \infty} Y^{\theta_M^*}$ exist as limits in the Banach space $\mathbb{L} := L^1(\Omega \times [0, T], \mathbb{F} \otimes \mathcal{B}([0, T]), \mathbb{P} \times \lambda_k)$, for every $k \leq K$, and that they are both equal to $\hat{X}$. Let us fix $k \leq K$. First we note that for every $B \in \mathcal{B}([0, T])$ we have

$$\mathbb{E}_{\lambda_k}[\mathbf{1}_B] = \lambda_k(B) = \frac{\tilde{\mathbb{P}}(n \geq k, t_k - \in B)}{\tilde{\mathbb{P}}(n \geq k)} = \frac{\mathbb{E}_{\tilde{\mathbb{P}}}[\mathbf{1}_{\{n \geq k\}} \mathbf{1}_{\{t_k - \in B\}}]}{\tilde{\mathbb{P}}(n \geq k)}.$$

Using "measure theoretic induction" (Durrett, 2010, Case 1-4 of Proof of Thm. 1.6.9) this yields for $c := (\mathbb{P}(n \geq k))^{-1}$ and a $\mathcal{B}([0, T])$-measurable function $Z : [0, T] \to \mathbb{R}, t \mapsto Z_t := Z(t)$ that

$$\mathbb{E}_{\lambda_k}[Z] = c\, \mathbb{E}_{\tilde{\mathbb{P}}}[\mathbf{1}_{\{n \geq k\}} Z_{t_k-}]. \tag{37}$$

Moreover, the triangle inequality and Lemma E.5 yield

$$
\Phi(\theta_M^*) - \Psi(\hat{X}) \geq \mathbb{E}_{\mathbb{P} \times \tilde{\mathbb{P}}} \left[ \frac{1}{n} \sum_{i=1}^{n} \left| X_{t_i} - Y_{t_i-}^{\theta_M^*} \right|_2^2 \right] - \Psi(\hat{X})
$$
$$
= \mathbb{E}_{\mathbb{P} \times \tilde{\mathbb{P}}} \left[ \frac{1}{n} \sum_{i=1}^{n} \left| \hat{X}_{t_i-} - Y_{t_i-}^{\theta_M^*} \right|_2^2 \right].
$$
(38)

For any $\mathbb{R}^{d_X}$-valued $Z \in \mathbb{L}$ the Hölder inequality, together with the fact that $n \geq 1$, yields

$$
\mathbb{E}_{\mathbb{P} \times \tilde{\mathbb{P}}} \left[ |Z|_2 \right] = \mathbb{E}_{\mathbb{P} \times \tilde{\mathbb{P}}} \left[ \frac{\sqrt{n}}{\sqrt{n}} |Z|_2 \right] \leq \mathbb{E}_{\mathbb{P} \times \tilde{\mathbb{P}}} [n]^{1/2} \, \mathbb{E}_{\mathbb{P} \times \tilde{\mathbb{P}}} \left[ \frac{1}{n} |Z|_2^2 \right]^{1/2}.
$$
(39)

Together, this implies that $\lim_{M \to \infty} Y^{\theta_M^*} = \hat{X}$ as a $\mathbb{L}$-limit. Indeed, with $\tilde{c} := \mathbb{E}_{\mathbb{P} \times \tilde{\mathbb{P}}} [n]^{1/2} < \infty$ we have

$$
\mathbb{E}_{\mathbb{P} \times \lambda_k} \left[ \left| \hat{X} - Y^{\theta_M^*} \right|_2 \right] = c \, \mathbb{E}_{\mathbb{P} \times \tilde{\mathbb{P}}} \left[ \mathbf{1}_{\{n \geq k\}} \left| \hat{X}_{t_k-} - Y_{t_k-}^{\theta_M^*} \right|_2 \right]
$$
$$
\leq c \, \tilde{c} \, \mathbb{E}_{\mathbb{P} \times \tilde{\mathbb{P}}} \left[ \mathbf{1}_{\{n \geq k\}} \frac{1}{n} \left| \hat{X}_{t_k-} - Y_{t_k-}^{\theta_M^*} \right|_2^2 \right]^{1/2}
$$
$$
\leq c \, \tilde{c} \, \mathbb{E}_{\mathbb{P} \times \tilde{\mathbb{P}}} \left[ \mathbf{1}_{\{n \geq k\}} \frac{1}{n} \sum_{i=1}^{n} \left| \hat{X}_{t_i-} - Y_{t_i-}^{\theta_M^*} \right|_2^2 \right]^{1/2}
$$
$$
\leq c \, \tilde{c} \, \mathbb{E}_{\mathbb{P} \times \tilde{\mathbb{P}}} \left[ \frac{1}{n} \sum_{i=1}^{n} \left| \hat{X}_{t_i-} - Y_{t_i-}^{\theta_M^*} \right|_2^2 \right]^{1/2}
$$
$$
\leq c \, \tilde{c} \left( \Phi(\theta_M^*) - \Psi(\hat{X}) \right)^{1/2} \xrightarrow{M \to \infty} 0,
$$

where we used first (37) and (39) followed by two simply upper bounds and (38) in the last step. The same argument can be applied to show that $\lim_{M \to \infty} Y^{\theta_M^{\min}} = \hat{X}$ as a $\mathbb{L}$-limit. In particular, this proves that the limit $Y := \lim_{M \to \infty} Y^{\theta_M^{\min}}$ exists as $\mathbb{L}$-limit and by Lemma E.6 it equals $\hat{X}$ $(\mathbb{P} \times \lambda_k)$-almost surely, for any $k \leq K$. $\qquad \square$

**Remark E.8.** *This result can be extended to any other neural network architecture for which a universal approximation theorem equivalent to (Hornik et al., 1989, Thm. 2.4) exists. Moreover, the stochastic process $X$ defined in (9) can be chosen more general, in particular, the diffusion part $\int \sigma dW$ can be replaced by any martingale, as long as the resulting process still is a Markov process and $\tilde{\mu}$ stays continuous.*

**Remark E.9.** *If we used the modified loss function $\tilde{\Psi}$ which is identical to $\Psi$ except that we drop the factor $\frac{1}{n}$, everything would work similarly and we could show $L^2$-convergence instead of $L^1$-convergence of $Y^{\theta_M^{\min}}$ to $\hat{X}$. However, we remark that there might exists $Z \in \mathbb{D}$, such that $\Psi(Z) < \infty$ while $\tilde{\Psi}(Z) = \infty$. In particular, if some moment of $n$ does not exist, such a process can be constructed.*

**Remark E.10.** *It follows directly from Theorem E.2 that $\lim_{M \to \infty} Y^{\theta_M^{\min}} = \hat{X}$ as random variables on $\Omega \times [0, T]$ except on sets which are null sets with respect to every product measure $\mathbb{P} \times \lambda_k$ for $1 \leq k \leq K$.*

**Remark E.11.** *The result of Theorem E.2 does not imply that $\hat{X}$ and $\lim_{M \to \infty} Y^{\theta_M^{\min}}$ are modifications or indistinguishable. For example, if $B \subset [0, T]$ is a subset such that no left-point of the observation times $(t_k-)$ lies in $B$ with probability greater 0, i.e. $\lambda_k(B) = 0$ for $1 \leq k \leq K$, then Theorem E.2 does not tell us how close $(\lim_{M \to \infty} Y^{\theta_M^{\min}})_t$ is to $\hat{X}_t$ for $t \in B$. In particular, it does not tell us whether they are equal $\mathbb{P}$-almost surely. Furthermore, such a set $B$ always exists, since there has to exists $t \in [0, T]$ such that $B := \{t\}$ has measure 0 for all $k$.*

In the following corollary we show, that Theorem E.2 can be extended to show convergence to the conditional expectation of $\varphi(X)$, for some function $\varphi \in C^{2,b}(\mathbb{R}^{d_X}, \mathbb{R})$, i.e. a function that is twice continuously differentiable with bounded derivatives.

**Corollary E.12.** *Let $\varphi \in C^{2,b}(\mathbb{R}^{d_X}, \mathbb{R})$, then the statement of Theorem E.2 holds equivalently, when replacing $X$ in the loss functions $\Psi$ and $\Phi$ by $\Gamma = \varphi(X)$ and $\hat{X}$ by the conditional expectation $\hat{\Gamma}$, where $\hat{\Gamma}_t := \mathbb{E}_{\mathbb{P} \times \tilde{\mathbb{P}}}[\varphi(X_t)|\mathcal{A}_t]$.*

Corollary E.12 combined with the monotone convergence theorem for conditional expectation theoretically enables us to make statements about the conditional law and conditional moments of $X$ under some a priori integrability assumptions.

*Proof of Corollary E.12.* We first remark that Proposition B.2 and Lemma E.4, E.5, B.3 hold similarly for the conditional expectation $\hat{\Gamma}$. Hence, the same argument as in the proof of Theorem E.2 implies that $\hat{\Gamma}$ is the unique $\mathbb{A}$-adapted minimizer of $\Psi$.

For simplicity of the notation we assume that $d_X = d_Y = 1$, i.e. that the process $X$ and the Brownian motion $W$ are 1-dimensional. However, the following works as well in the general case, where the correlations of the Brownian motion components have to be taken into account. By Itô's Formula (Protter, 2005, Chap. II, Thm. 32), $\Gamma = \varphi(X)$ is the solution of the SDE

$$
\begin{aligned}
d\varphi(X)_t &= \varphi'(X_t)dX_t + \tfrac{1}{2}\varphi''(X_t)d[X,X]_t \\
&= \varphi'(X_t)\mu(t,X_t)dt + \varphi'(X_t)\sigma(t,X_t)dW_t + \tfrac{1}{2}\varphi''(X_t)\sigma(t,X_t)^2 dt \\
&= \alpha(t,X_t)dt + \beta(t,X_t)dW_t,
\end{aligned}
$$

for $\alpha(t,X_t) := \varphi'(X_t)\mu(t,X_t) + \tfrac{1}{2}\varphi''(X_t)\sigma(t,X_t)^2$ and $\beta(t,X_t) := \varphi'(X_t)\sigma(t,X_t)$. Defining $\tilde{\alpha}$ similar to $\tilde{\mu}$ as

$$
\tilde{\alpha} : \Delta \times \mathbb{R}^{d_X} \to \mathbb{R}^{d_X}, \quad (t,r,\xi) \mapsto P_{t,t+r}(X_t,\alpha)\big|_{X_t=\xi} = \mathbb{E}_{\mathbb{P}}\left[\alpha(t+r,X_{t+r})|X_t=\xi\right],
$$

one can use the boundedness of of $\varphi'$ and $\varphi''$ to show that it is continuous and that

$$
\hat{\Gamma}_t = \mathbb{E}[\varphi(X)_t|\mathcal{A}_t] = \Gamma_{\tau(t)} + \int_{\tau(t)}^t \tilde{\alpha}\left(\tau(t), s - \tau(t), X_{\tau(t)}\right) ds.
$$

In particular, Proposition B.1 and B.4 hold equivalently for $\Gamma$.
Similar to (36), the neural network parameters can be chosen such that

$$
\sup_{(u,v,t,r)\in[-M,M]^{d_H \times d_X} \times \Delta} \left|f_{\theta_1^*}(u,v,t,r) - \tilde{\alpha}(t,r,v)\right|_2 \leq \varepsilon, \tag{40}
$$

which then implies the statement of the Corollary similar as in the proof of Theorem E.2. $\qquad\square$

### E.2 CONVERGENCE OF THE MONTE CARLO APPROXIMATION

In the following, we assume the size of the neural network $M$ is fixed and we study the convergence with respect to the number of samples $N$. Moreover, we show that both types of convergence can be combined. To do so, we define $\tilde{\Theta}_M := \{\theta \in \Theta_M \mid |\theta|_2 \leq M\}$, which is a compact subspace of $\Theta_M$. It is straight forward to see, that $\Theta_M$ in Theorem E.2 can be replaced by $\tilde{\Theta}_M$. Indeed, if the needed neural network weights for an $\varepsilon$-approximation have too large norm, then one can increase $M$ until it is sufficiently big. The following convergence analysis is based on (Lapeyre & Lelong, 2019, Chapter 4.3).

**Theorem E.13.** *Let $\theta_{M,N}^{\min} \in \Theta_{M,N}^{\min} := \arg\inf_{\theta \in \tilde{\Theta}_M}\{\hat{\Phi}_N(\theta)\}$ for every $M, N \in \mathbb{N}$. Then, for every $M \in \mathbb{N}$, $(\mathbb{P} \times \hat{\mathbb{P}})$-a.s.*

$$
\hat{\Phi}_N \xrightarrow{N \to \infty} \Phi \quad \text{uniformly on } \tilde{\Theta}_M.
$$

*Moreover, for every $M \in \mathbb{N}$, $(\mathbb{P} \times \tilde{\mathbb{P}})$-a.s.*

$$
\Phi(\theta_{M,N}^{\min}) \xrightarrow{N \to \infty} \Phi(\theta_M^{\min}) \quad \text{and} \quad \hat{\Phi}_N(\theta_{M,N}^{\min}) \xrightarrow{N \to \infty} \Phi(\theta_M^{\min}).
$$

*In particular, one can define an increasing sequence $(N_M)_{M \in \mathbb{N}}$ in $\mathbb{N}$ such that for every $1 \leq k \leq K$ we have that $Y^{\theta_{M,N_M}^{\min}}$ converges to $\hat{X}$ for $M \to \infty$ as random variable in $L^1(\Omega \times [0,T], \mathbb{P} \times \lambda_k)$. In particular, the limit process $Y := \lim_{M \to \infty} Y^{\theta_{M,N_M}^{\min}}$ equals $\hat{X}$ $(\mathbb{P} \times \lambda_k)$-almost surely as a random variable on $\Omega \times [0,T]$.*

The following Monte Carlo convergence analysis is based on (Lapeyre & Lelong, 2019, Section 4.3). In comparison to them, we do not need the additional assumptions that were essential in (Lapeyre & Lelong, 2019, Section 4.3), i.e. that all minimizing neural network weights generate the same neural network output. This assumption is not needed, because we do not aim to show that $Y^{\theta_{M,N}^{\min}}$ converges to $Y^{\theta_M^{\min}}$.

We define the separable Banach space $\mathcal{S} := \{x = (x_i)_{\in \mathbb{N}} \in \ell^1(\mathbb{R}^{d_X}) \mid \|x\|_{\ell^1} < \infty\}$ with the norm $\|x\|_{\ell^1} := \sum_{i \in \mathbb{N}} |x_i|_2$.

### E.2.1 Convergence of Optimization Problems

Consider a sequence of real valued functions $(f_n)_n$ defined on a compact set $K \subset \mathbb{R}^d$. Define, $v_n = \inf_{x \in K} f_n(x)$ and let $x_n$ be a sequence of minimizers $f_n(x_n) = \inf_{x \in K} f_n(x)$.

From (Rubinstein & Shapiro, 1993, Theorem A1 and discussion thereafter) we have the following Lemma.

**Lemma E.14.** *Assume that the sequence $(f_n)_n$ converges uniformly on $K$ to a continuous function $f$. Let $v^* = \inf_{x \in K} f(x)$ and $\mathcal{S}^* = \{x \in K : f(x) = v^*\}$. Then $v_n \to v^*$ and $d(x_n, \mathcal{S}^*) \to 0$ a.s.*

The following lemma is a consequence of (Ledoux & Talagrand, 1991, Corollary 7.10) and (Rubinstein & Shapiro, 1993, Lemma A1).

**Lemma E.15.** *Let $(\xi_i)_{i \geq 1}$ be a sequence of i.i.d random variables with values in $\mathcal{S}$ and $h : \mathbb{R}^d \times \mathcal{S} \to \mathbb{R}$ be a measurable function. Assume that a.s., the function $\theta \in \mathbb{R}^d \mapsto h(\theta, \xi_1)$ is continuous and for all $C > 0$, $\mathbb{E}(\sup_{|\theta|_2 \leq C} |h(\theta, \xi_1)|) < +\infty$. Then, a.s. $\theta \in \mathbb{R}^d \mapsto \frac{1}{N} \sum_{i=1}^N h(\theta, \xi_i)$ converges locally uniformly to the continuous function $\theta \in \mathbb{R}^d \mapsto \mathbb{E}(h(\theta, \xi_1))$,*

$$\lim_{N \to \infty} \sup_{|\theta|_2 \leq C} \left| \frac{1}{N} \sum_{i=1}^N h(\theta, \xi_i) - \mathbb{E}(h(\theta, \xi_1)) \right| = 0 \qquad a.s.$$

### E.2.2 Strong law of large numbers

Let us define

$$F(x, y, z) := |x - y|_2 + |y - z|_2$$

and $\xi_j := (X_{t_1^j}^j, \ldots, X_{t_{n^j}^j}^j, 0, \ldots)$, where $X_{t_i^j}^j$ are random variables describing the realizations of the training data, as defined in Section D.2. By this definition we have $n^j := n^j(\xi_j) := \max_{i \in \mathbb{N}} \{\xi_{j,i} \neq 0\}$ $\mathbb{P}$-almost-surely and we know that $\xi_j$ are i.i.d. random variables taking values in $\mathcal{S}$. Furthermore, let us write $Y_t^\theta(\xi)$ to make the dependence of $Y$ on the input and the weight $\theta$ explicit. Then we define

$$h(\theta, \xi_j) := \frac{1}{n^j} \sum_{i=1}^{n^j} F\left(X_{t_i^j}, Y_{t_i^j}^\theta(\xi_j), Y_{t_i^j-}^\theta(\xi_j)\right)^2.$$

**Lemma E.16.** *The following properties are satisfied.*

($\mathcal{P}_1$) *There exists $\kappa > 0$ such that for all $S \in \mathcal{S}$ and $\theta \in \tilde{\Theta}_M$ we have $|Y_t^\theta(S)|_2 \leq \kappa (1 + |X_{\tau(t)}|_2)$ for all $t \in [0, T]$.*

($\mathcal{P}_2$) *Almost-surely the random function $\theta \in \tilde{\Theta}_M \mapsto Y_t^\theta$ is uniformly continuous for every $t \in [0, T]$.*

($\mathcal{P}_3$) *We have $\mathbb{E}_{\mathbb{P} \times \tilde{\mathbb{P}}} \left[ \frac{1}{n} \sum_{i=1}^n |X_{t_i}|_2^2 \right] < \infty$ and $\mathbb{E}_{\mathbb{P} \times \tilde{\mathbb{P}}} \left[ \frac{1}{n} \sum_{i=1}^n |X_{t_i-}|_2^2 \right] < \infty$.*

*Proof.* By definition of the neural networks with sigmoid activation functions (in particular having bounded outputs), all neural network outputs are bounded in terms of the norm of the network weights, which is assumed to be bounded, not depending on the norm of the input. Since after a jump at $\tau(t)$, $Y$ has the value $X_{\tau(t)}$, we can find $\kappa$ depending on $T$, such that the claimed bound is satisfied for all $t$, proving ($\mathcal{P}_1$).

Since the activation functions are continuous, also the neural networks are continuous with respect to their weights $\theta$, which implies that also $\theta \in \tilde{\Theta}_M \mapsto Y_t^\theta$ is continuous. Since $\tilde{\Theta}_M$ is compact, this automatically yields uniform continuity and therefore finishes the proof of $(\mathcal{P}_2)$.

$(\mathcal{P}_3)$ follows directly from the stronger result in Lemma E.7. $\qquad\square$

*Proof of Theorem E.13.* We apply Lemma E.15 to the sequence of $i.i.d$ random function $h(\theta, \xi_j)$. From $(\mathcal{P}_1)$ of Lemma E.16 we have that

$$
\begin{aligned}
F(X_{t_i^j}, Y_{t_i^j}, Y_{t_i^j-})^2 &= \left( \left| X_{t_i^j} - Y_{t_i^j}^\theta \right|_2 + \left| Y_{t_i^j}^\theta - Y_{t_i^j-}^\theta \right|_2 \right)^2 \\
&\leq 4 \left( |X_{t_i^j}|_2^2 + |Y_{t_i^j}^\theta|_2^2 + |Y_{t_i^j}^\theta|_2^2 + |Y_{t_i^j-}^\theta|_2^2 \right) \\
&\leq 4 \left( 3|X_{t_i^j}|_2^2 + \kappa(1 + |X_{t_{i-1}^j}|_2^2) \right).
\end{aligned}
$$

Hence, we obtain that

$$
h(\theta, \xi_j) = \frac{1}{n^j} \sum_{i=1}^{n^j} F(X_{t_i^j}, Y_{t_i^j}, Y_{t_i^j-})^2 \leq \frac{12 + 4\kappa}{n^j} \sum_{i=1}^{n^j} \left| X_{t_i^j} \right|_2^2 + 4\kappa + |x|_2^2,
$$

implying that

$$
\mathbb{E}_{\mathbb{P} \times \tilde{\mathbb{P}}} \left[ \sup_{\theta \in \tilde{\Theta}_M} h(\theta, \xi_j) \right] \leq (12 + 4\kappa) \mathbb{E}_{\mathbb{P} \times \tilde{\mathbb{P}}} \left[ \frac{1}{n} \sum_{i=1}^n |X_{t_i}|_2^2 \right] + 4\kappa + |x|_2^2 < \infty, \qquad (41)
$$

using $(\mathcal{P}_3)$ of Lemma E.16. By $(\mathcal{P}_2)$ of Lemma E.16, the function $\theta \mapsto h(\theta)$ is continuous. Therefore, we can apply Lemma E.15, yielding that almost-surely for $N \to \infty$ the function

$$
\theta \mapsto \frac{1}{N} \sum_{j=1}^N h(\theta, \xi_j) = \hat{\Phi}_N(\theta) \qquad (42)
$$

converges uniformly on $\tilde{\Theta}_M$ to

$$
\theta \mapsto \mathbb{E}_{\mathbb{P} \times \tilde{\mathbb{P}}}[h(\theta, \xi_1)] = \Phi(\theta). \qquad (43)
$$

We deduce from Lemma E.14 that $d(\theta_{M,N}^{\min}, \Theta_M^{\min}) \to 0$ a.s. when $N \to \infty$. Then there exists a sequence $(\hat{\theta}_{M,N}^{\min})_{N \in \mathbb{N}}$ in $\Theta_M^{\min}$ such that $|\theta_{M,N}^{\min} - \hat{\theta}_{M,N}^{\min}|_2 \to 0$ a.s. for $N \to \infty$. The uniform continuity of the random functions $\theta \mapsto Y_t^\theta$ on $\tilde{\Theta}_M$ implies that $|Y_t^{\theta_{M,N}^{\min}} - Y_t^{\hat{\theta}_{M,N}^{\min}}|_2 \to 0$ a.s. when $N \to \infty$ for all $t \in [0, T]$. By continuity of $F$ this yields $|h(\theta_{M,N}^{\min}, \xi_1) - h(\hat{\theta}_{M,N}^{\min}, \xi_1)|_2 \to 0$ a.s. as $N \to \infty$. With (41) we can apply dominated convergence which yields

$$
\lim_{N \to \infty} \mathbb{E}_{\mathbb{P} \times \tilde{\mathbb{P}}} \left[ |h(\theta_{M,N}^{\min}, \xi_1) - h(\hat{\theta}_{M,N}^{\min}, \xi_1)|_2 \right] = 0.
$$

Since for every integrable random variable $Z$ we have $0 \leq |\mathbb{E}[Z]|_2 \leq \mathbb{E}[|Z|_2]$ and since $\hat{\theta}_{M,N}^{\min} \in \Theta_M^{\min}$ we can deduce

$$
\lim_{N \to \infty} \Phi(\theta_{M,N}^{\min}) = \lim_{N \to \infty} \mathbb{E}_{\mathbb{P} \times \tilde{\mathbb{P}}} \left[ h(\theta_{M,N}^{\min}, \xi_1) \right] = \lim_{N \to \infty} \mathbb{E}_{\mathbb{P} \times \tilde{\mathbb{P}}} \left[ h(\hat{\theta}_{M,N}^{\min}, \xi_1) \right] = \Phi(\theta_M^{\min}). \qquad (44)
$$

Now by triangle inequality,

$$
|\hat{\Phi}_N(\theta_{M,N}^{\min}) - \Phi(\theta_M^{\min})| \leq |\hat{\Phi}_N(\theta_{M,N}^{\min}) - \Phi(\theta_{M,N}^{\min})| + |\Phi(\theta_{M,N}^{\min}) - \Phi(\theta_M^{\min})|. \qquad (45)
$$

(42), (43) and (44) imply that both terms on the right hand side converge to 0 when $N \to \infty$, which finishes the proof of the first part of the Theorem.

We define $N_0 := 0$ and for every $M \in \mathbb{N}$

$$
N_M := \min \left\{ N \in \mathbb{N} \mid N > N_{M-1}, |\Phi(\theta_{M,N}^{\min}) - \Phi(\theta_M^{\min})| \leq \tfrac{1}{M} + |\Phi(\theta_M^{\min}) - \Psi(\hat{X})| \right\},
$$

which is possibly due to (44). Then Theorem E.2 implies that

$$|\Phi(\theta^{\min}_{M,N_M}) - \Psi(\hat{X})| \leq \tfrac{1}{M} + 2|\Phi(\theta^{\min}_M) - \Psi(\hat{X})| \xrightarrow{M \to \infty} 0.$$

Therefore, we can apply the same arguments as in the proof of Theorem E.2 (starting from (38)) to show that

$$\mathbb{E}_{\mathbb{P} \times \lambda_k} \left[ \left| \hat{X} - Y^{\theta^{\min}_{M,N_M}} \right|_2 \right] \leq c\, \tilde{c} \left( \Phi(\theta^{\min}_{M,N_M}) - \Psi(\hat{X}) \right)^{1/2} \xrightarrow{M \to \infty} 0,$$

for every $1 \leq k \leq K$. $\qquad\square$

**Corollary E.17.** *In the setting of Theorem E.13, we also have that $(\mathbb{P} \times \tilde{\mathbb{P}})$-a.s.*

$$\Phi(\theta^{\min}_{M,N_M}) \xrightarrow{M \to \infty} \Psi(\hat{X}) \quad and \quad \hat{\Phi}_{\tilde{N}_M}(\theta^{\min}_{M,\tilde{N}_M}) \xrightarrow{M \to \infty} \Psi(\hat{X}),$$

*where $(\tilde{N}_M)_{M \in \mathbb{N}}$ is another increasing sequence in $\mathbb{N}$.*

*Proof.* The first convergence result was already shown in the proof of Theorem E.13 and the second one can be shown similarly, when defining $\tilde{N}_M$ by $\tilde{N}_0 := 0$ and for every $M \in \mathbb{N}$

$$\tilde{N}_M := \min \left\{ N \in \mathbb{N} \mid N > \tilde{N}_{M-1}, |\hat{\Phi}_N(\theta^{\min}_{M,N}) - \Phi(\theta^{\min}_M)| \leq \tfrac{1}{M} + |\Phi(\theta^{\min}_M) - \Psi(\hat{X})| \right\},$$

which is possibly due to (45). $\qquad\square$

### E.3 DISCUSSION ABOUT OPTIMAL WEIGHTS

In Theorem E.2 and E.13, the focus lies on the convergence analysis under the assumption that optimal weights are found. Below we discuss, why this assumption is not restrictive in theory.

**Global versus local optima.** The assumption that the optimal weights are found, is typical for a convergence analysis of a neural network based algorithm, since the objective function is highly complex and non-convex with respect to the weights. In particular, it is well known that the standard choice of (stochastic) gradient descent optimization methods do in general only find local and not global minima. Since the difference between any local minimum and the global minimum can not generally be bounded, it is unrealistic to hope for a theoretical proof of convergence with respect to such optimisation schemes. On the other hand, global optimization methods as for example simulated annealing provably convergence (in probability) to a global optimum (Locatelli, 2000; Lecchini-Visintini et al., 2008). Hence, combining those with our result, convergence in probability of our model output to the conditional expectation can be established, without the assumption that the optimal weights are found. However, these global optimization schemes come at the cost of much slower training compared to (stochastic) gradient descent methods when applied in practice. Moreover, several works have focused on showing that most local optima of neural networks are nearly global, see for example (Feizi et al., 2017) and the related work therein. Hence, using (stochastic) gradient descent optimization methods likely yield nearly globally optimal weights much more efficiently. In our case, this is also supported by our empirical convergence studies in Section 6.3.

# F  EXPERIMENTAL DETAILS

All implementations were done using PyTorch. The code is available at https://github.com/HerreraKrachTeichmann/NJODE.

## F.1  IMPLEMENTATION DETAILS

**Dataset.** For each of the SDE models (Black-Scholes, Ornstein-Uhlenbeck, Heston) a dataset was generated by sampling $N = 20'000$ paths of the SDE using the Euler-scheme. We used an equidistant time grid of mesh $0.01$ between time $0$ and $T = 1$. Independently for each path, observation times were sampled from $\mathbb{P}_t$, by using each of the grid points with probability $0.1$ as an observation time. In particular, $n \sim \text{Bin}(100, 0.1)$, $t_0 = 0$ and the observation times $\{t_i\}_{1 \leq i \leq n}$ were chosen uniformly on the time grid. Hence, $10\%$ of the grid points were used on average. This way, $n$ and $t_i$ are defined as a discretized version of those given in Example A.2 where $n$ is binomially distributed and $t_i$ chosen uniformly on $[0, T]$. For each of these datasets, the samples were used in a $80\%/20\%$ split for training and testing. The SDEs of the dataset models and the chosen parameters are described below.

- **Black-Scholes:**
  - SDE: $dX_t = \mu X_t dt + \sigma X_t dW_t$, where $W$ is a 1-dimensional Brownian motion
  - conditional expectation: $E(X_{t+s}|X_t) = X_t e^{\mu s}$
  - used parameters: $\mu = 2$, $\sigma = 0.3$, $X_0 = 1$

- **Ornstein-Uhlenbeck:**
  - SDE: $dX_t = -k(X_t - m)dt + \sigma dW_t$, where $W$ is a 1-dimensional Brownian motion
  - conditional expectation: $E(X_{t+s}|X_t) = X_t e^{-ks} + m\left(1 - e^{-ks}\right)$
  - used parameters: $k = 2, m = 4, \sigma = 0.3, X_0 = 1$

- **Heston:**
  - SDE: for $W$ and $Z$ 1-dimensional Brownian motions

$$dX_t = \mu X_t dt + \sqrt{v_t} X_t dW_t$$
$$dv_t = -k(v_t - m)dt + \sigma \sqrt{v_t} dZ_t$$

  - conditional expectation[7]: $E(X_{t+s}|X_t) = X_t e^{\mu s}$
  - used parameters: $\mu = 2, \sigma = 0.3, X_0 = 1$ $k = 2, m = 4, v_0 = 4, \rho = \text{Corr}(W, Z) = 0.5$

**Architecture.** In our experiments we choose the dimension of the latent variable to be $d_H = 10$. For $f_{\theta_1}, \tilde{g}_{\theta_3}$ and $\tilde{\rho}_{\theta_2}$ we use 2-hidden-layer feed-forward neural networks, with 50 nodes in each hidden layer and $\tanh$ activation functions. Then the neural networks $g_{\theta_3}$ and $\rho_{\theta_2}$ are defined as residual versions of $\tilde{g}_{\theta_3}$ and $\tilde{\rho}_{\theta_2}$, by adding a residual shortcut between the input and the output of the neural networks. Dropout was applied after each non-linearity with a rate of $0.1$. To scale the possibly unbounded inputs of the neural networks to a bounded hypercube, we applied $\tanh$ component-wise to $x$ and $h$ in every neural network. This was done, because neural networks sometimes become unstable when their inputs become large. To solve the ODE of the neural ODE part, the simple Euler-method was used.

**Training.** The neural networks were trained using the Adam optimizer (Kingma & Ba, 2014) with a learning rate of $0.001$ and weight decay $0.0005$ for 200 epochs using a batch size of 200. A random initialization was used and no hyper-parameter optimization was needed.

**Further training results.** Further training results on test samples are shown in Figure 8, 9, 10.

## F.2  EXPERIMENTS ON OTHER DATASETS

We test our framework on additional synthetic datasets.

---

[7]see (Rujivan & Zhu, 2012, Equation 2.9)

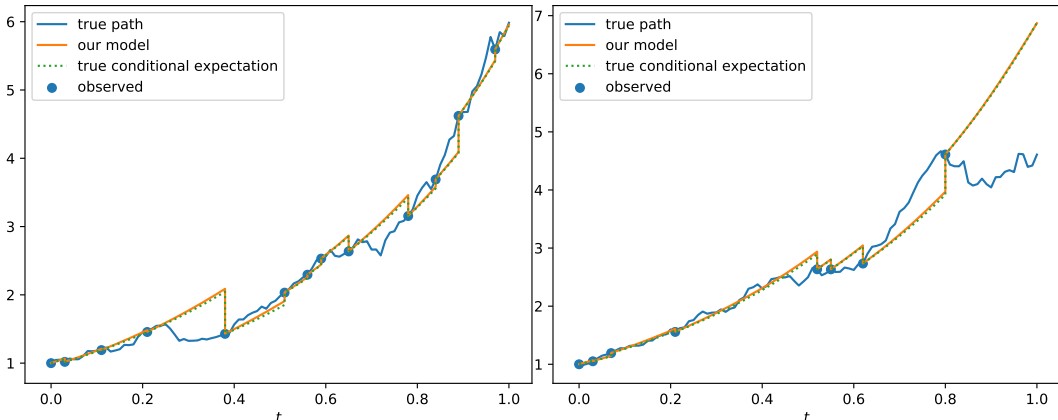

Figure 8: Black-Scholes

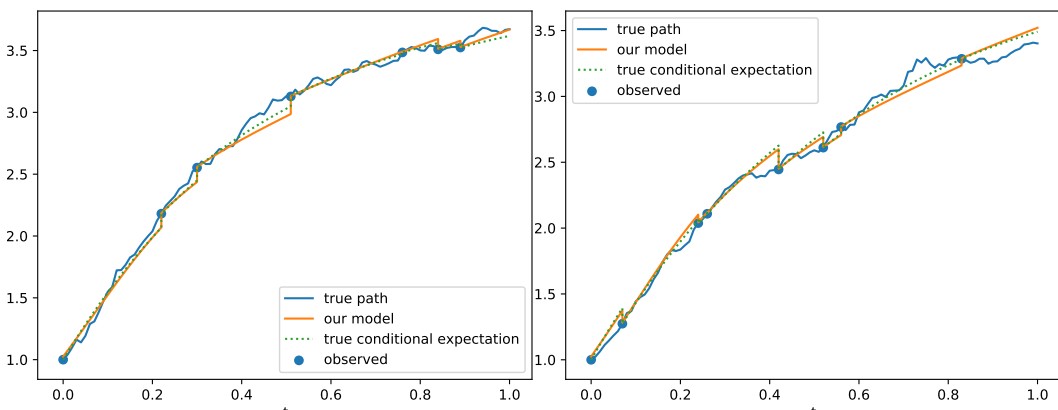

Figure 9: Ornstein-Uhlenbeck

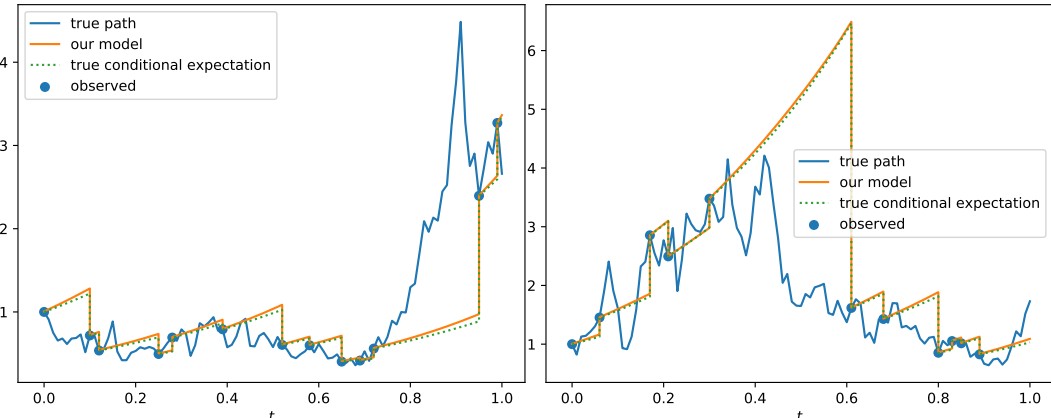

Figure 10: Heston

### F.2.1 HESTON MODEL WITHOUT FELLER CONDITION

If the Feller condition

$$2\,k\,m \geq \sigma^2$$

is satisfied in the Heston model, it is known that the variance process $v_t$ is always strictly bigger than 0 and that the Euler-scheme works well to sample from it. However, if the Feller condition is not satisfied, the variance process can touch 0, where the process is reflected deterministically. Then the Euler-scheme can not longer be used to sample from the model. Moreover, this situation is generally considered as more delicate. Close to 0 the distribution of $v_t$ behaves differently than further away of 0 (Andersen, 2007, Section 3). As explained by Andersen (2007), there are differently well performing sampling schemes for a Heston model where the Feller condition is not satisfied. We use the simplest to implement, which is a slight extension of the Euler scheme, where values of $v_t$ below 0 are replaced by 0 (Andersen, 2007, Section 2.3). Although there is empirical evidence, that the resulting sampling distribution of $v_t$ close to 0 is not correctly replicating the true distribution (Jean-François et al., 2015, Figure 2,3), this method already produces sufficiently good sample paths.

**Dataset.** Heston model, sampled as in Section F.1 with the extension described above. Used parameters: $\mu = 2$, $\sigma = 3$, $X_0 = 1$ $k = 2$, $m = 1$, $v_0 = 0.5$, $\rho = \text{Corr}(W, Z) = 0.5$. Hence the Feller condition and also the weaker condition $4\,k\,m \geq \sigma^2$ discussed in (Jean-François et al., 2015, Section 3.2), are both not satisfied. We produce two datasets, a 1-dimensional one similar to before, where only $X$ is stored and a 2-dimensional one, where both $X$ and $v$ are stored, hence also $v$ is a target for prediction. Note that $v$ has the same conditional expectation as the Ornstein-Uhlenbeck SDE. In the 2-dimensional dataset, $X$ and $v$ are always observed at the same time.

**Architecture & Training.** Same as in Section F.1, but with batch size 100.

**Results.** The model learns to replicate the true conditional expectation process, which is analytic, hence not effected by the sampling scheme. In particular, we see that our model is very robust, since even in the delicate case where the Feller condition is not satisfied and a sampling scheme is used, that does not perfectly replicate the true distribution, our model still works well. Due to the very similar results, in Figure 11 we only show plots on test samples of the 2-dimensional dataset, where $X$ and $v$ are predicted.

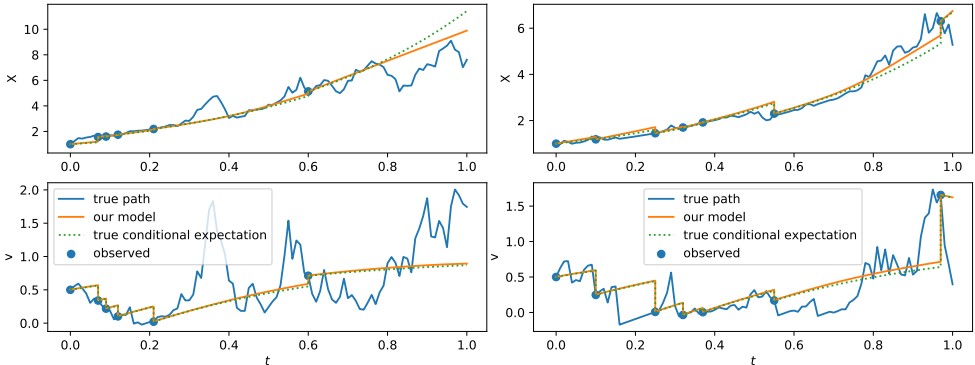

Figure 11: Heston model without Feller condition. In both plots, the upper sub-plot corresponds to the 1-dimensional path of $X_t$ and the lower sub-plot corresponds to the 1-dimensional path of $v_t$.

### F.2.2 DATASET WITH CHANGING REGIME

**Dataset.** To be able to evaluate the performance of our model, we test a change of regime by combining two synthetic datasets. On the first half of the time interval $[0, 0.5]$ we use the Ornstein-Uhlenbeck and on the second half $[0.5, 1]$ the Black-Scholes model. The Black-Scholes process takes as starting point, the last point of the Ornstein-Uhlenbeck process. We use the same hyper-parameters for the dataset generation as in Section F.1, except that we set the parameter $m = 10$ in the Ornstein-Uhlenbeck model to make the two parts act on similar scales.

**Architecture & Training.** Same as in Section F.1, but with batch size 100. Moreover, we used 100 neurons in each hidden layer, to account for the more complicated setting, where also a time dependence has to be learnt.

**Results.** In the plots on test samples of the dataset shown in Figure 12 we see that our model correctly learns the change of regime.

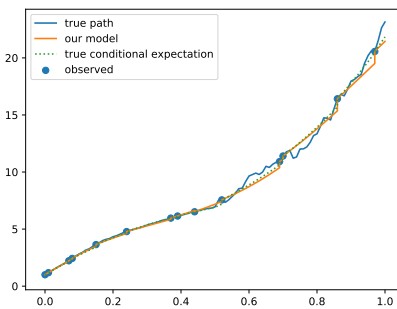 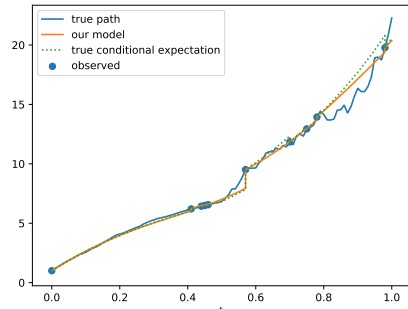

Figure 12: Our model evaluated on a stochastic dataset that follows an Ornstein-Uhlenbeck SDE on the time interval $[0, 0.5]$ and an Black-Scholes model on the time interval $(0.5, 1]$.

### F.2.3 DATASET WITH EXPLICIT TIME DEPENDENCE

**Dataset.** We use the Black-Scholes datasets of Section F.1, where we replace the constant $\mu$ by the time dependent constant $\frac{\alpha}{2}(\sin(\beta t) + 1)$, for $\alpha, \beta > 0$. The conditional expectation changes accordingly. For the data generation we use the same hyper-parameters as in Section F.1, except that we use instead of $\mu$ the parameter $\alpha = 2$ and $\beta \in \{2\pi, 4\pi\}$.

**Architecture & Training.** Same as in Section F.1, but with batch size 100. Moreover, we used 400 neurons in each hidden layer, to account for the more complicated setting, where an explicit time dependence has to be learnt.

**Results.** We show plots on test samples of the datasets in Figure 5. We see that the model learns to adapt to the time-dependent coefficients.

### F.3 DETAILS ON CONVERGENCE STUDY

**Evaluation metric.** For the sampling time grid with equidistant step size $\Delta_t := \frac{T}{\nu}$, $\nu \in \mathbb{N}$, on $[0, T]$ and the true and predicted conditional expectation for path $j \in \mathbb{N}$, $\hat{X}^j$ and $Y^j$ respectively, we define the evaluation metric as

$$\text{eval}(\hat{X}, Y) := \frac{1}{N_2} \sum_{j=1}^{N_2} \frac{1}{\nu + 1} \sum_{i=0}^{\nu} \left( \hat{X}^j_{i\Delta_t} - Y^j_{i\Delta_t} \right)^2, \tag{46}$$

where $N_2$ is the number of test samples, and $j$ accordingly iterates over the paths in the test set.

**Increasingly big training sets.** We use the following procedure to create increasingly big training sets, while keeping the *exactly same* test set for evaluation. Out of the initial $20'000$ paths, we take $N_2 := 4'000$ paths, which are fixed as the testing set. Out of the remaining $16'000$ paths, we randomly choose $N_1$ training paths for $N_1 \in \{200, 400, 800, 1'600, 3'200, 6'400, 12'800\}$.

**Increasing neural network sizes.** The increasingly big neural networks are defined as follows. For all involved networks, we use the feed-forward 2-layer architecture with $\tanh$ activations (cf. Appedix F.1), where each hidden layer has the same size $M$ for $M \in \{10, 20, 40, 80, 160, 320\}$.

**Results on Black-Scholes dataset.** In accordance with the theoretical results in Theorem E.2 and E.13, we see that the evaluation metric decreases when $N_1$ and $M$ increase (Figure 13). It is important to notice, that already a quite small number of samples can lead to a good approximation of the conditional expectation, if the network is big enough. In particular, the Monte Carlo approximation of the theoretical loss function is good already with a few samples. On the other hand, even with a large number of samples, the evaluation metric does not become so small, if the network size is not big enough. From a practitioners point of view, this is good news, since increasing the network size is often much easier than collecting more training data.

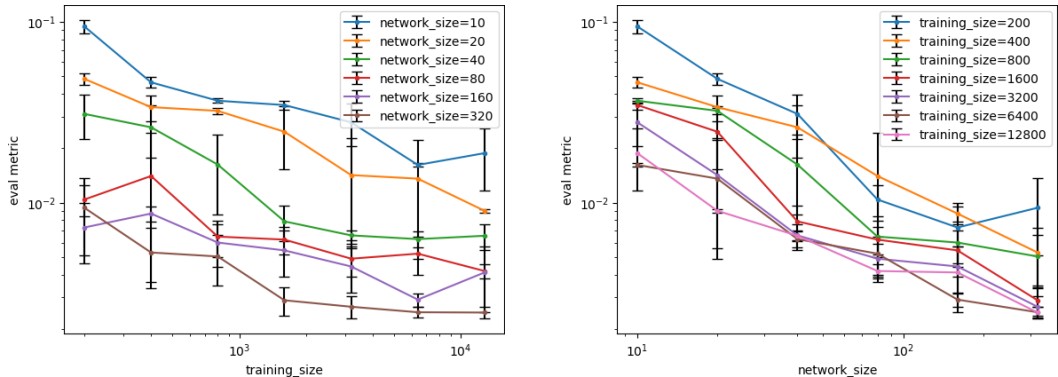

Figure 13: Black-Scholes dataset. Mean $\pm$ standard deviation (black bars) of the evaluation metric for varying $N_1$ and $M$.

**Results on Ornstein-Uhlenbeck and Heston dataset.** We get very similar results on the Ornstein Uhlenbeck (Figure 14) and Heston dataset (Figure 15). Similar as for Black Scholes, also for Ornstein-Uhlenbeck the training size $N_1$ is not as important as the network size $M$. For all network sizes, increasing the training size further than 1600 hardly changes the performance, while increasing $M$ is crucial to get better performance. In contrast to this, for the Heston dataset we see that a large

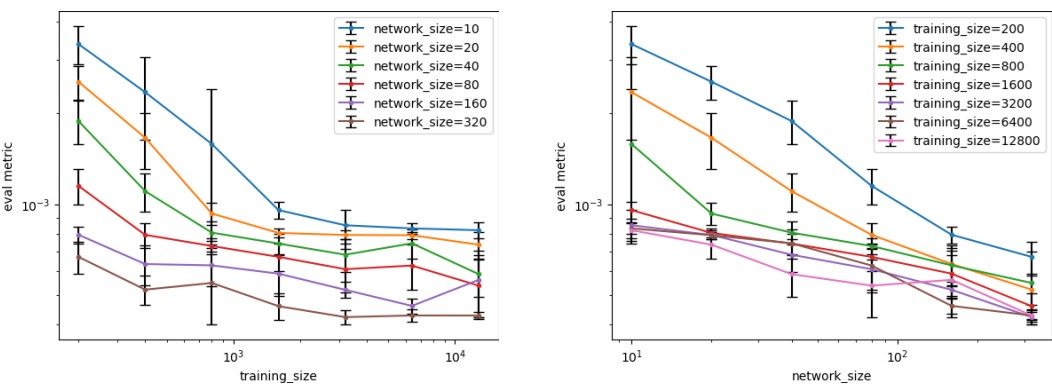

Figure 14: Ornstein-Uhlenbeck dataset. Mean $\pm$ standard deviation (black bars) of the evaluation metric for varying $N_1$ and $M$.

number of training samples is more important to get a smaller convergence metric. This reflects the fact, that the Heston dataset is more complex and more difficult to learn.

### F.4   DETAILS ON COMPARISON TO GRU-ODE-BAYES

**GRU-ODE-Bayes (Brouwer et al., 2019).** To the best of our knowledge, this is the neural network based method with the most similar task to ours. In particular, this continuous time model is trained to learn the unknown temporal parameters of a normal distribution, best describing the conditional distribution of $X$ given the previous observations. This distribution is given by the Fokker-Planck equation. Brouwer et al. (2019) outlined, that their model can exactly represent the Fokker-Planck dynamics of the Ornstein-Uhlenbeck process, since the corresponding distribution is Gaussian.

**Implementation of GRU-ODE-Bayes.** We use the code of the official implementation of (Brouwer et al., 2019)[8] and slightly adjust it for our purpose. In particular, we do not use incomplete observations, hence the input mask used for this task has always only 1-entries. Furthermore, we slightly changed the scheme how the time steps are taken, to be the same as in our implementation, so that comparisons can be made. Besides these minor changes, the original model is used and

---

[8]https://github.com/edebrouwer/gru_ode_bayes

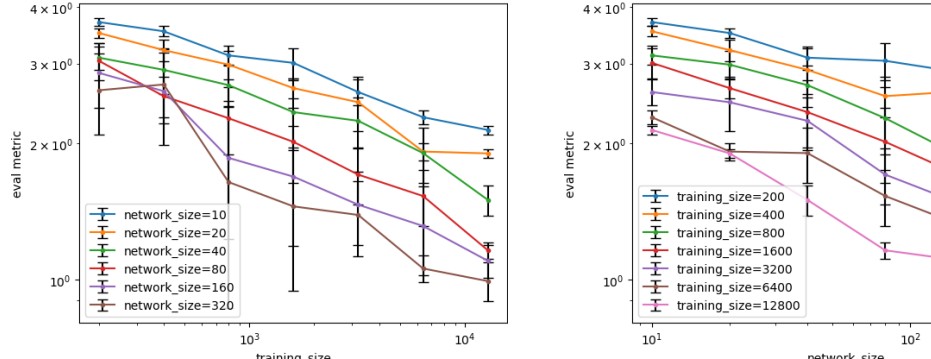

Figure 15: Heston dataset. Mean $\pm$ standard deviation (black bars) of the evaluation metric for varying $N_1$ and $M$.

trained on all 3 datasets (Black-Scholes, Ornstein-Uhlenbeck and Heston). We tried out all combinations of the following parameters and always chose the best performing one for comparison to our model: impute $\in \{\text{True}, \text{False}\}$, logvar $\in \{\text{True}, \text{False}\}$, mixing $\in \{0.0001, 0.5\}$ and hiddensize $\in \{50, 100\}$, whereby phidden and prephidden were chosen to be equal to hiddensize. The first parameter choices were the ones used in the official implementation. The model was always trained using the Euler-method for solving ODEs and with dropout $= 0.1$, to be comparable to our implementation. Furthermore, we always used the full GRU-ODE-Bayes implementation, since it should be the more powerful one. For the comparison, we only use the estimated mean of the normal distribution (the estimated variance is not used), which is precisely the estimated conditional expectation of the model.

**Implementation of NJ-ODE.** Same as described in Section F.1. For each dataset only one model was trained, since we already saw the convergence properties before and wanted to have a qualitative comparison to GRU-ODE-Bayes. In particular, we did not try to optimize our hyper-parameters for best performance (e.g. by choosing different number of layers or neurons or different activation functions) and used about $10K$ trainable parameters compared to the best performing GRU-ODE-Bayes models of our study which used $112K$ trainable parameters.

**Datasets.** Same as described in Section F.1. The train and test sets were fixed to be the same for all trained models.

**Training.** All models were trained using the Adam optimizer (Kingma & Ba, 2014) with a learning rate of 0.001 and weight decay 0.0005 for 100 epochs using a batch size of 20. A random initialization was used.

### F.4.1 FURTHER RESULTS AND DISCUSSION OF THE COMPARISON

**Evolution of losses and evaluation metric during training.** In Figure 16 and 17 we see, that already after a few epochs the NJ-ODE model finds close to optimal weights for the given network size on the Black-Scholes and Orstein-Uhlenbeck dataset and oscillates around this optimum. The Heston dataset is considerably more difficult and the model slowly converges to close-to-optimal weights for the given network size. In comparison to this, it takes the GRU-ODE-Bayes model longer to converge to its close-to-optimal weights on the Black-Scholes and Orstein-Uhlenbeck dataset. Moreover, the model does not converge to close-to-optimal weights on the Heston dataset, but rather oscillates between bad weights. Due to some very large outliers, this is not directly visible in the plot, but can be deduced from Table 6.4.

**Comparison of predicted paths.** For each dataset we show 5 paths that were predicted with NJ-ODE and with GRU-ODE-Bayes, first at the optimal epoch, i.e. where the test loss was minimal during training (Figures 18, 19, 20), and then at the last epoch (Figures 21, 22, 23). For the sake of comparison, in each row the performance on the same test sample is shown. The results are very similar for NJ-ODE and GRU-ODE-Bayes on the Black-Scholes and Ornstein-Uhlenbeck dataset, with sometimes the one and sometimes the other having a slightly better prediction. On the Heston

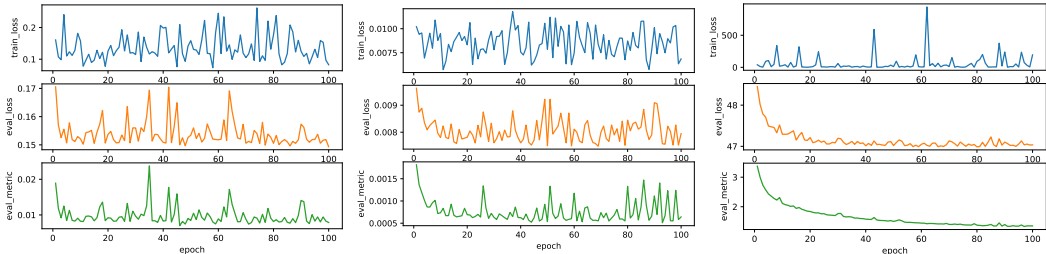

Figure 16: NJ-ODE on Black-Scholes, Ornstein-Uhlenbeck and Heston (from left to right). Blue (1st row): training loss, orange (2nd row): evaluation loss, green (3rd row): evaluation metric.

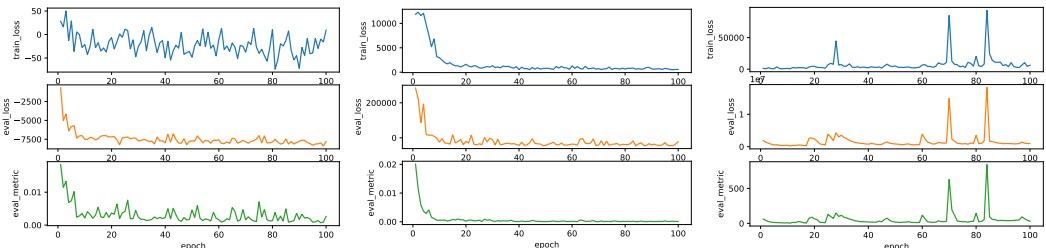

Figure 17: GRU-ODE-Bayes' best performing models on Black-Scholes, Ornstein-Uhlenbeck and Heston (from left to right). Blue (1st row): training loss, orange (2nd row): evaluation loss, green (3rd row): evaluation metric.

dataset, the results of NJ-ODE are good predictions, being correct whenever a new observation is made and not too far away even after longer periods without observations. On the other hand, GRU-ODE-Bayes is not even correct at times of new observations and learns an incorrect behaviour in between observations (e.g. making kinks where there should not be any). For the last epoch, we see that this malfunction is amplified.

### F.4.2 FURTHER EXPERIMENTS

Since the results of GRU-ODE-Bayes were unexpectedly bad on the Heston dataset, we performed additional tests. In particular, we retrained the best combinations of GRU-ODE-Bayes with the larger batch size 100. For smaller versions of the network (M), i.e. with $hiddensize = prephidden = phidden = 50$ this stabilized the training, such that the models converged. Still, even at the best epoch, the models suffered from the same difficulties as explained in Section F.4.1. Increasing the $hiddensize$ by factor 2 to 100 (L), made the training unstable again, and the models did not converge any more. This gives more empirical evidence, that GRU-ODE-Bayes can not be reliably trained on more complex datasets, where the target conditional distributions differs to much from a normal distribution. In contrast to this, retraining NJ-ODE with batch size 100, once for the smaller version described in Section F (S), once for a larger version with 100 instead of 50 neurons in all hidden layers (M) and once with 200 neurons (L), yielded the expected results of better performance with larger networks. In particular, there is no instability for larger networks. In Table F.4.2 we show results of our model and of the best GRU-ODE-model for the given size.

Table 4: The minimal, last and average value of the evaluation metric (smaller is better) on the Heston dataset throughout the 100 epochs of training, together with the number of trainable parameters.

|          | min  | last  | average | params  |
|----------|------|-------|---------|---------|
| GRU (M)  | 2.38 | 4.63  | 8.25    | 28'802  |
| GRU (L)  | 5.58 | 17.39 | 1877.26 | 112'602 |
| ours (S) | 1.71 | 1.71  | 2.05    | 10'071  |
| ours (M) | 1.21 | 1.23  | 1.60    | 35'121  |
| ours (L) | 1.07 | 1.08  | 1.36    | 130'221 |

## F.5 REAL WORLD DATASETS WITH INCOMPLETE OBSERVATIONS

### F.5.1 LOSS FUNCTION FOR INCOMPLETE OBSERVATIONS

In accordance with the self-imputation scheme, the loss function is adjusted to only use the non-imputed coordinates, i.e. if $m$ is the random process in $\{0,1\}^{d_X}$ describing which coordinates are observed we have

$$\Psi(Z) := \mathbb{E}_{\mathbb{P} \times \tilde{\mathbb{P}}} \left[ \frac{1}{n} \sum_{i=1}^{n} \left( |m_{t_i} \odot (X_{t_i} - Z_{t_i})|_2 + |m_{t_i} \odot (Z_{t_i} - Z_{t_i-})|_2 \right)^2 \right].$$

### F.5.2 CLIMATE FORECASTING DETAILS

**Dataset.** We use the publicly available United State Historical Climatology Network (USHCN) daily dataset (Menne et al., 2016) together with all pre-processing steps as they were provided by Brouwer et al. (2019). In particular, there are 5 sporadically observed (i.e. incomplete observations) climate variables (daily temperatures, precipitation, and snow) measured at $1'114$ stations scattered over the United States during an observation window of 4 years (between 1996 and 2000) where each station has an average of 346 observations over those 4 years. For 5 folds, the data is split into train (70%), validation (20%) and test (10%) sets. The task is to predict the next 3 measurements after the first 3 years of observation. The mean squared error between the prediction and the correct values is computed on the validation and test set.

**Baselines.** We use the results reported in (Brouwer et al., 2019, Table 1) as baselines for our comparison and perform the exact same 5-fold cross validation using the same folds with the same train, validation and test sets. For completeness, we give all results of the table together with our results in Table 6.5. We only show the mean squared error (MSE), since our model does not provide the negative log-likelihood.

**Implementation of NJ-ODE.** We once use the architecture described in Section F.1 (S) and once use the exact same architecture, but with hidden size $d_H = 50$ and $400$ instead of $50$ nodes in each hidden layer (L). To deal with the incomplete observations, the self-imputation scheme described in Section 6.5 is used. In particular, the data is self-imputed and passed together with the observation mask as input to the network $\tilde{\rho}_{\theta_2}$. Moreover, the network $f_{\theta_1}$ uses the NJ-ODE output at the last observation time instead of the last observation as input.

**Training.** Both versions of NJ-ODE were trained using the Adam optimizer (Kingma & Ba, 2014) with a learning rate of $0.001$ and weight decay $0.0005$ for 100 epochs using a batch size of 100. A random initialization was used. The performance on the validation set was used for early stopping after the first 100 epochs, i.e. early stopping was possible at any epoch between $\{101, \ldots, 200\}$.

**Results.** The results of our model and all models reported in (Brouwer et al., 2019, Table 1) are shown in Table 6.5.

### F.5.3 PHYSIONET PREDICTION DETAILS

**Dataset.** We use the publicly available PhysioNet Challenge 2012 dataset (Goldberger et al., 2000) together with all pre-processing steps as they were described and provided by Rubanova et al. (2019). In particular, there are 41 features of 8000 patients that are observed irregularly over a 48 hour time period. The observations are put on a time grid with step size $0.016$ hours leading to 3000 grid points. While in their paper Rubanova et al. (2019) say that they use 2880 grid points (i.e. minute wise) in their implementation they used 3000. Moreover, in contrast to what was written in the paper, the 4 constant features were not excluded in their implementation, hence we also keep them. Furthermore, we also rescale the time-grid to $[0, 1]$ and normalize each feature as Rubanova et al. (2019) did it for training the model. The dataset is split with the same fixed seed into $80\%$ training and $20\%$ test set. In particular, no cross validation but only multiple runs with new random initializations are performed to be exactly comparable to the results reported by Rubanova et al. (2019). On the test set, the observation paths are split in half. The first half (first 1500 time steps) is used as input for the model, from which the second half (second 1500 time steps) should be forecast. The masked mean squared error (MSE) with a certain balancing procedure between the predictions and the true values is reported. For comparability we use the exact same implementation of the MSE as Rubanova et al. (2019).

**Baselines.** We compare the performance of our model to latent ODE on the extrapolation task (as described above) on physionet. As baseline for our comparison we use the results reported in (Rubanova et al., 2019, Table 5). For completeness, we show all extrapolation results of the table together with our results in Table 6.5. We shortly outline the different approach of latent ODE compared to our model for the given extrapolation task. Latent ODE also splits the training samples similar to the test samples in half, using the first half as input and the second half as target. It is trained as an encoder-decoder, encoding the observations in the first half and reconstructing (decoding) the second half. This falls in the standard supervised learning framework. In particular, this approach cannot straight forward be extended for online forecasting. Moreover, this approach might learn certain path dependencies. On the other hand, our model is trained as always, online forecasting after each observation until the next observation is made. Instead of splitting the training samples we use the entire path as input for our unsupervised training framework. Our model is based on the assumption that paths are Markov, therefore it cannot learn path dependencies, i.e. dependencies on more than just the last observation. However, by training the model also on the second half of the training samples, it learns the underlying behaviour there, which should be helpful for the extrapolation task.

**Implementation of NJ-ODE.** We use the architecture described in Section F.1, but with hidden size $d_H = 41$. To deal with the incomplete observations, the self-imputation scheme described in Section 6.5 is used. In particular, the data is self-imputed and passed together with the observation mask as input to the network $\tilde{\rho}_{\theta_2}$. Moreover, the network $f_{\theta_1}$ uses the NJ-ODE output at the last observation time instead of the last observation as input.

**Training.** The NJ-ODE was trained using the Adam optimizer (Kingma & Ba, 2014) with a learning rate of 0.001 and weight decay 0.0005 for 175 epochs using a batch size of 50. 5 runs with random initialization were performed over which the mean and standard deviations were calculated. In particular, these runs always used the same training and test set, specified by the same random seed as in thee implementation of Rubanova et al. (2019).

**Results.** The results of our model and all models reported in (Rubanova et al., 2019, Table 5, extrapolation) are shown in Table 6.5. We report the minimal MSE on the test set during the 175 epochs, since it was not differently specified in (Rubanova et al., 2019). However, if the MSE of the epoch is used where the training loss is minimal, the results are nearly the same with $1.986 \pm 0.058$ ($\times 10^{-3}$). Moreover, we trained a larger model for 120 epochs, where 200 nodes were used instead of 50 ($187'323$ parameters in total), leading to slightly better results of $1.934 \pm 0.007$ ($\times 10^{-3}$) (at minimal MSE) and $1.982 \pm 0.027$ ($\times 10^{-3}$) (at minimal training loss).

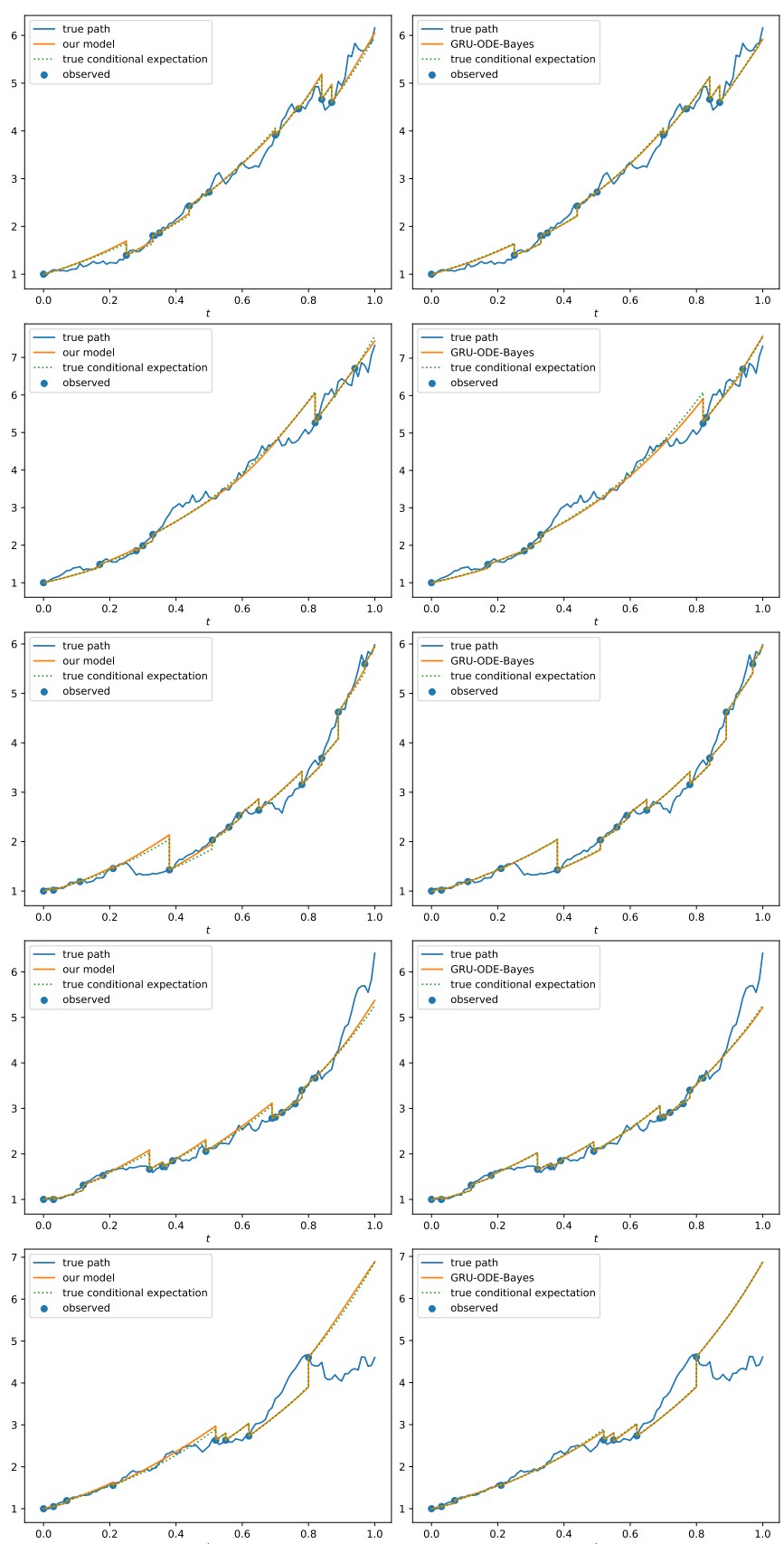

Figure 18: Comparison of predictions for Black-Scholes paths at best epoch.

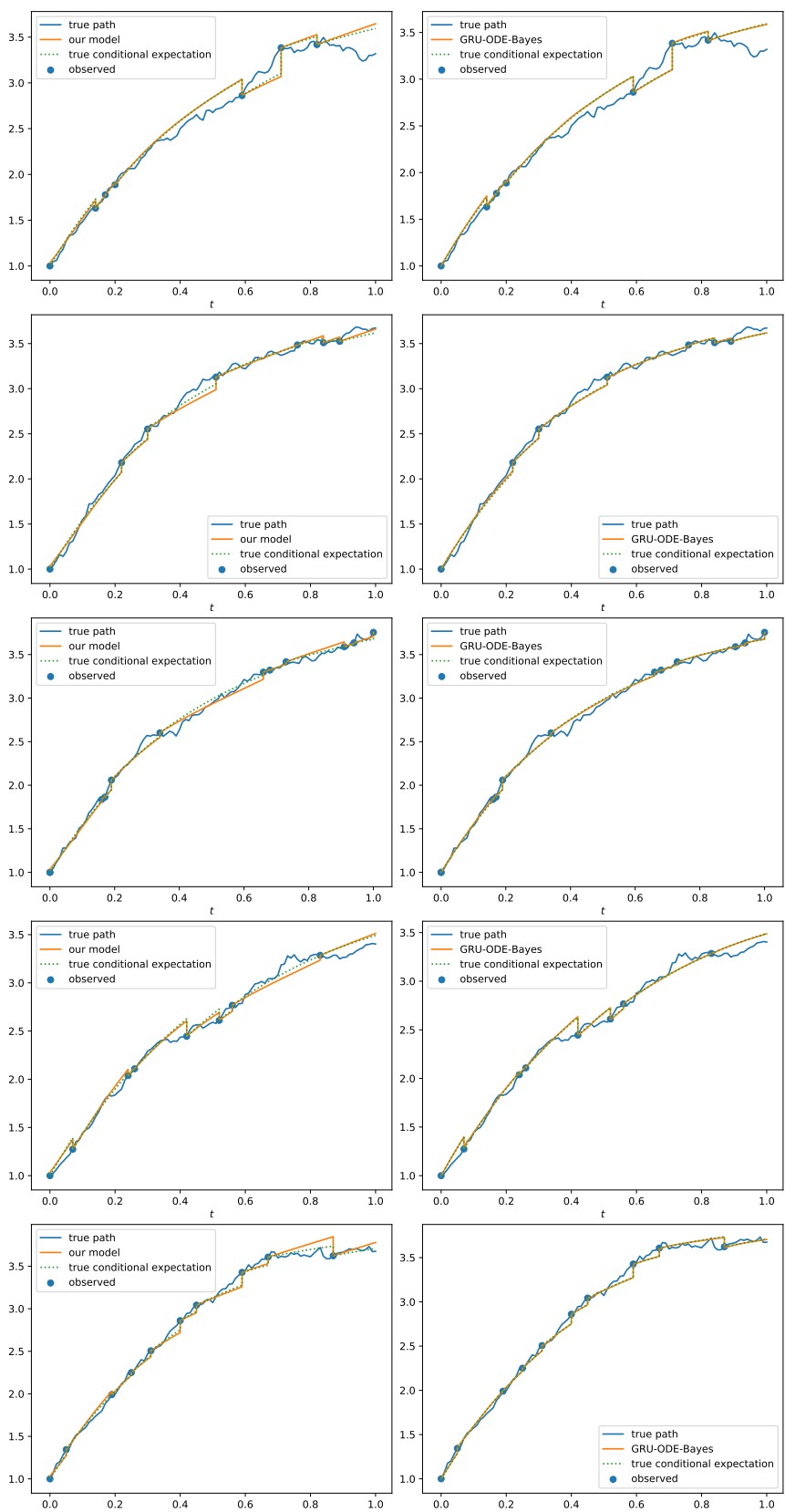

Figure 19: Comparison of predictions for Ornstein-Uhlenbeck paths at best epoch.

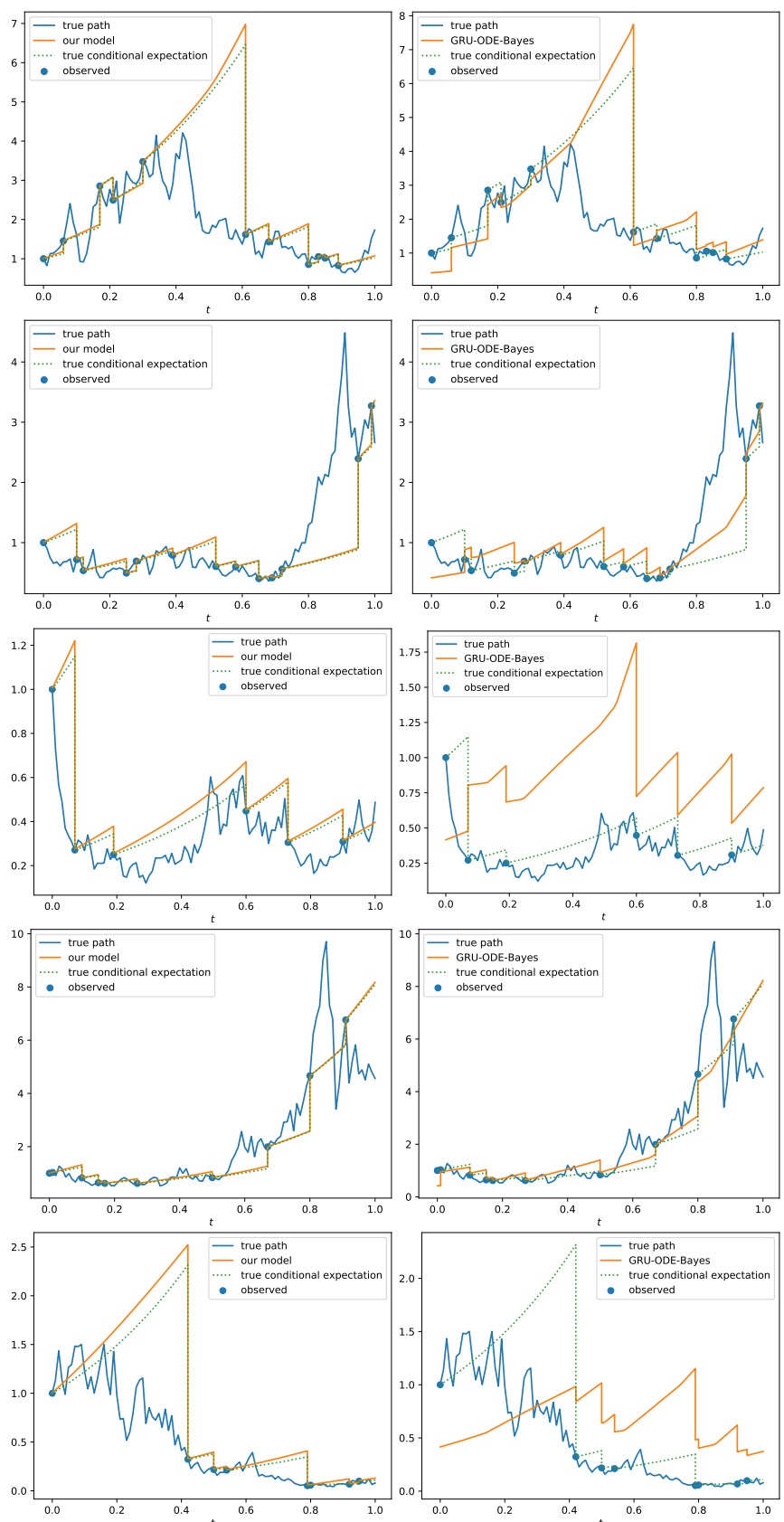

Figure 20: Comparison of predictions for Heston paths at best epoch.

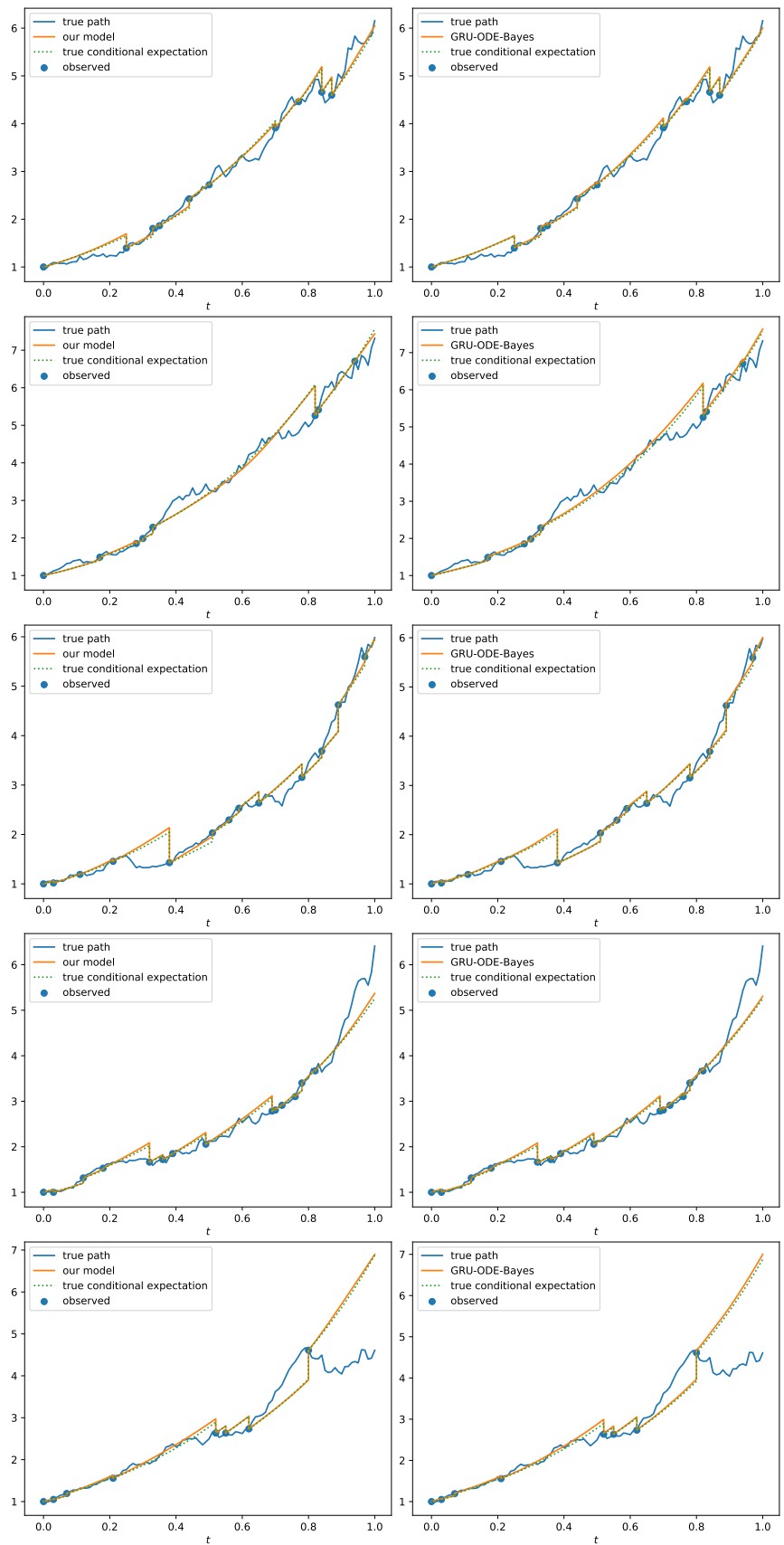

Figure 21: Comparison of predictions for Black-Scholes paths at last epoch.

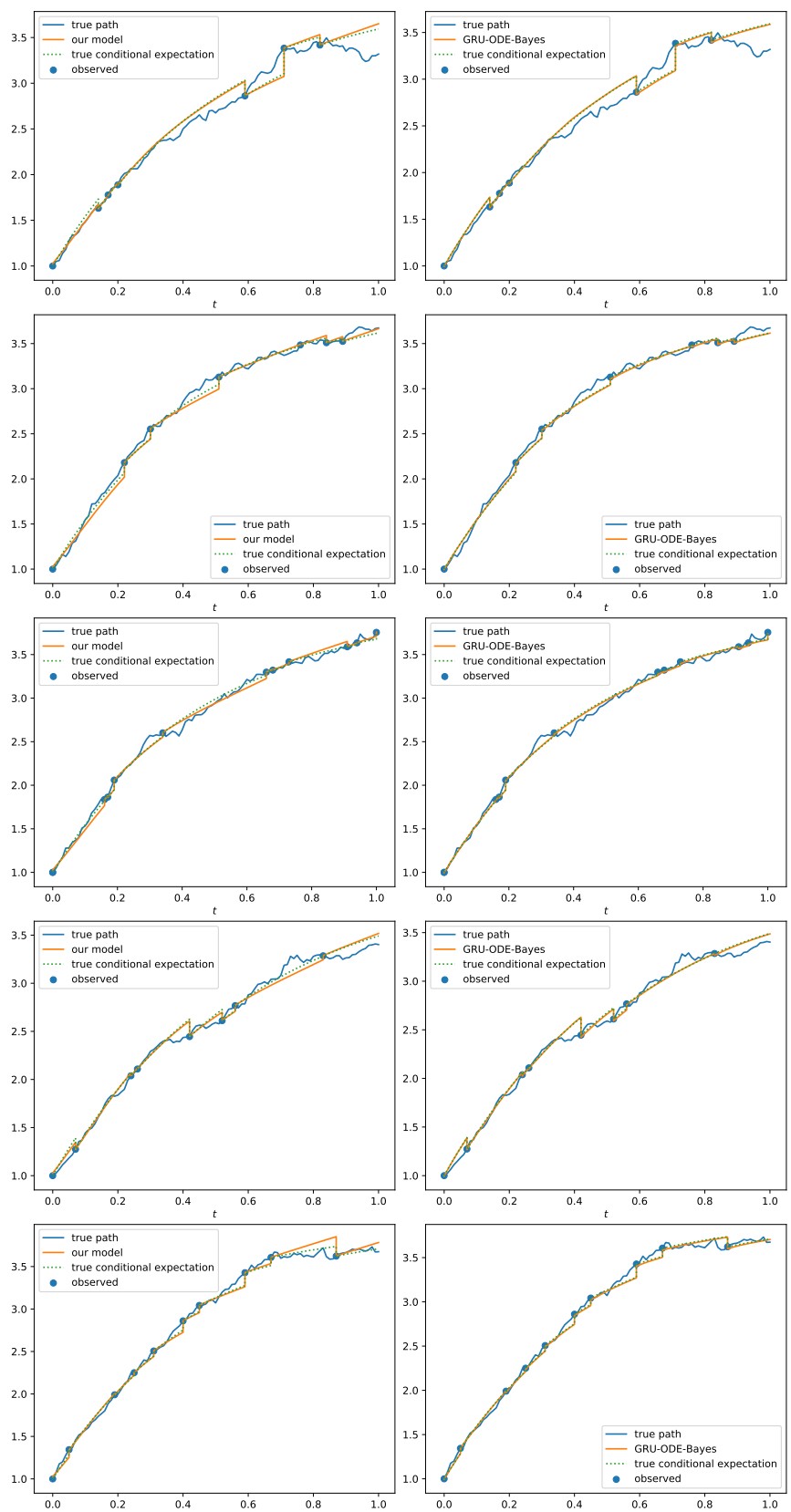

Figure 22: Comparison of predictions for Ornstein-Uhlenbeck paths at last epoch.

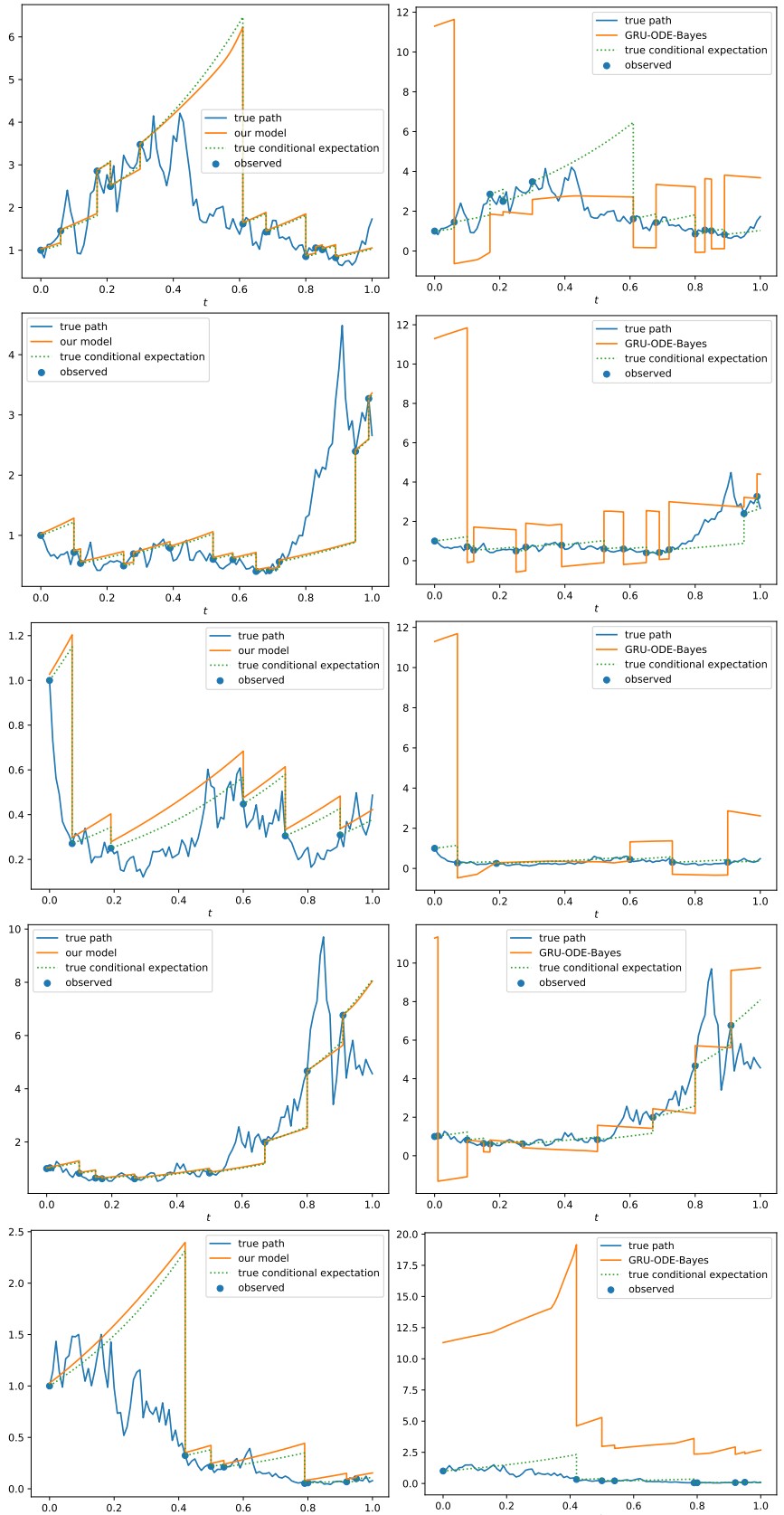

Figure 23: Comparison of predictions for Heston paths at last epoch.

