# OpenReview forum: "Neural Jump Ordinary Differential Equations: Consistent Continuous-Time Prediction and Filtering"
_ICLR.cc/2021/Conference — ICLR 2021 Poster_

### Official Review · AnonReviewer1 · 2020-10-28
**Official Blind Review #1**

**Rating:** 6
**Confidence:** 4

**Review:**

The submission studies a simplified model of ODE-RNN and GRU-ODE-Bayes theoretically, proves convergence results, and presents experimental results in companion with the theoretical results.


The paper does a good job in defining concepts precisely. Though, this has come at a cost of highly complex notation, which may hinder the average researcher in the ML+diffeq community, who may not have a strong background in probability theory, to understand the paper. I would therefore recommend the authors to simplify the notation by deferring the precise mathematical definition of concepts such as information sigma algebras to the appendix.


The section (sec. 2.4) on optimal approximation of a stochastic process in the main text is somewhat vague. Optimality certainly depends on the cost function being considered, in which case, the appendix states that the 2-norm is used here. The particular norm being used is somewhat independent of the construction of the probability space, e.g. we could consider the same prob. space and evaluate the difference between the random variable and the fixed prediction using some other function, say the metric function induced by the L1-norm). This makes terms such as “L^2(omega X omega tilde, ...)-minimizer” somewhat confusing. Note my comment here is somewhat handwavy about the precise technicalities, but it should convey the relevant idea.


My main concern regarding the paper is about novelty. It seems that the model considered in section 3 falls broadly in line with ODE-RNN and GRU-ODE-Bayes. On the other hand, the experiments section also doesn’t compare against latent ODE, which is a strong but relevant baseline.


The section (sec. 4) on theoretical convergence results mostly assume that the ERM can already be found. This rather strong assumption therefore leaves the theorems in that section not unexpected, and at the same time, less relevant for practitioners. It is also unclear whether convergence rates can be derived.

The paper does a decent job in clarifying its relationship with prior work.


Post-rebuttal:
- I thank the authors for improving the presentation of the paper and including additional experiments comparing to latent ODE.

---

> ### Author Response · Authors · 2020-11-18
> **Simplified paper, clarification of contribution and comparison to latent ODE**
>
> Thank you for your review.
>
> __Simplify the paper__
> We have simplified the main paper a lot, by moving the precise mathematical definition of concepts and the theorems to the appendix. The paper is now written in a similar way as other papers cited, in order to reach a bigger audience of the Machine Learning community.
>
> __Section on optimal approximation is vague__
> You are completely right that this section (but also the corresponding references in the abstract and introduction) was vague and that a different norm would yield different optimizers. Therefore, we put more emphasis on clarifying that we consider the minimization problem with respect to the $L^2$-norm throughout the paper.
>
> __Novelty__
> Our model is different from the previous work. In the ODE-RNN and in the GRU-ODE-Bayes, it is a recurrent neural network, where between two observations, the hidden state is modeled by a neural ODE. In our model, we don’t use any recurrent neural network. This makes the model much easier and faster to train. We instead have a neural ODE that takes three more parameters (the last observation, the current time and the duration between the current time and the last observation). Moreover, we want to emphasize that we provided a novel training framework. For the first time, we provided a mathematical formulation, a rigorously defined problem statement and based on the new objective function, the theoretical guarantees that our algorithm works, which would not have been possible with a different training framework. In the GRU-ODE-Bayes paper, the authors do not give any theoretical guarantees for the model, only an empirical study.  In contrast, our method is proven to converge to the optimal solution.
>
> __Added comparison with latent ODE__
> In our paper we consider exactly the same task as the GRU-ODE-Bayes paper. The latent ODE paper, although being very similar, does not consider the same task. In particular, the latent ODE (as it is) can not be used for online forecasting. This is reflected in the way it is applied to the extrapolation task, where the model is trained in a supervised learning setting, mapping the first half (input) of the time series to the second half (target). Compared to this, our (and GRU-ODE-Bayes) approach can be interpreted as unsupervised learning. In contrast to the latent ODE, the ODE-RNN might be used for the online forecasting task, but the authors emphasize (in their paper and their official implementation of the models) that ODE-RNN should be used for interpolation tasks only.
> This is the reason why we did not compare our method to the latent ODE in the first place.
> Although our approach is different from latent ODE’s approach, their extrapolation task is one that can be tackled by both models. We have added an experiment to the paper, where we apply our model in the exact same setting as the latent ODE for the extrapolation task on physionet. Our model achieves a performance of $1.945 \pm 0.007$ ($\times 10^{-3}$) and outperforms the latent ODE with a reported performance of $2.208 \pm 0.050$ ($\times 10^{-3}$).
>
> __Assumption that ERM can be found in convergence results__
> We deliberately constructed the loss function in such a way that convergence can be proven, therefore, it is not surprising that our results are expected. However, it is the first time that such a proof was provided for the class of neural ODE based models.
> We have added a paragraph after the (informal) theorem outlining that the assumption that the ERM can already be found is not restrictive. There exist global optimization methods, as for example simulated annealing, that provably converge to a global optimum in probability. Apart from that, several works try to show that most local optima of neural networks are nearly globally optimal. This implies that in practice, using standard stochastic gradient descent methods which converge to local minima, will supply nearly optimal weights that should be good enough for the approximation.
> Since the proofs of our theorems depend on the universal approximation theorem, which does not provide convergence rates, we also cannot derive convergence rates from our analysis.
>
> __Summary__
> We hope we have addressed every concern that the reviewer has raised. We would be very happy to have further discussion if there are any other obstacles to raising the review score.

---

> > ### Comment · AnonReviewer1 · 2020-11-25
> > **new comments**
> >
> > I genuinely thank the authors for putting in the effort to try address my concerns. The draft is now more appealing to the average machine learner to which this conference targets.
> >
> > After carefully pondering specific points, I still have many concerns, which I believe should be carefully addressed:
> >
> > 1. After reading Algorithm 1 for multiple times, I cannot see the motivation behind fixing a step size \delta t. It seems the inner loop can be rewritten as a single ODE solve, with the the output y_t being the result of applying the outputNN with the corresponding h_t. The overall model can thus be described as "the state follows an ODEs in between observations, and is applied a jump (defined by an NN at observations)". This is basically the high-level framework of both GRU-ODE-Bayes and ODE-RNN.
> >
> > Take GRU-ODE-Bayes as example, the ODE defining the state's evolution in between observations is the continuous-time version of a gated recurrent unit, while the jump update is the gated recurrent unit's update.
> >
> > If this accurate, then in my opinion the difference between the proposed NJ-ODE and GRU-ODE-Bayes is that 1) NJ-ODE uses a general NN to define the ODE between observations, as opposed to the cont. time version of GRU, and 2) uses a general NN to define jump updates at observations, as opposed to the GRU update.
> >
> > While the authors have mentioned this distinction vs ODE-RNNs, I believe the authors should also include a description comparing to GRU-ODE-Bayes.
> >
> > 2. After carefully reading eq (7), I sense a striking resemblance compared to the loss in the GRU-ODE-Bayes paper. Loosely speaking, the "jump part at observation" can be mapped to the "preLoss" in GRU-ODE-Bayes (assuming observation), with the caveat that in the GRU-ODE-Bayes paper, the predictive mean is based on the hidden state prior to jump. The "continuous part between two observations" can be mapped to the "postLoss" in GRU-ODE-Bayes, with the caveat that p_obs needs to be ignored there (so that the KL between Gaussians is the norm of mean scaled by a function of the shared variance).
> >
> > Apart from theoretical convergence reasons, what are the motivations in making these changes? How does changing specific things (e.g. use pre-jump vs post-jump update for preloss) affect results in practice? What happens if only one aspect of the overall loss is changed? Are the differences significant?
> >
> > 3. I thank the authors for attempting to address my concern regarding the theory.
> >
> > While I agree that it is possible to attain global convergence with sim. annealing, I shall emphasize that this is definitely not what we typically use in practice. While there are also special architectures/training paradigms (e.g. NTK) that guarantee global convergence, there are also ample evidence that these architectures and paradigms differ from what's happening in practice when SGD is used to train an NN. For this reason, I don't think the theory is particularly relevant for justifying the modeling part.
> >
> > The theoretical argument, while being a new statement, relies mostly on standard results to prove and does not introduce new math/stats techniques, and therefore is unlikely to benefit theoretician working in related subfields.

---

> > > ### Author Response · Authors · 2020-11-25
> > > **Answers to new comments 1/2**
> > >
> > > We thank you for taking the time to study the updated paper and the answers we provided. We also thank you for your new valuable remarks and concerns which contributed to further improvements of the paper.
> > >
> > > __1. Algorithm__
> > > You are completely right that, from a theoretical point of view, the inner loop of the algorithm could be expressed by a single ODE, where continuously-in-time the outputNN is applied to the latent variable $h_t$ producing $y_t$ for all $t_i < t < t_{i+1}$. Actually, this is what we describe in equations (30) and (31) in the appendix. However, if we want to give an implementable pseudo-algorithm, this continuous-in-time procedure needs to be discretized. Since the outputs between observation times are important, we wanted to show how this discretization can be done. We think that it gives additional insight for the practical point of view, which was not provided by GRU-ODE-Bayes and ODE-RNN.
> > >
> > > You are correct about the high-level framework, which is the same as in GRU-ODE-Bayes and ODE-RNN. To establish the connection to this, we added a description similar to the ODE-RNN description (new equation (7)), and clarified that the pseudo-algorithm is an implementable version of it. Thank you for pointing this out.
> > >
> > > Your description of the GRU-ODE-Bayes architecture and the difference to NJ-ODE is completely correct. As we described in the paper in the paragraph “GRU-ODE-Bayes” after equation (6), this architecture can be understood as a special case of the ODE-RNN architecture. In particular, a GRU is used for the RNN and a continuous version of the GRU for the neural ODE. Therefore, we think it is enough to explain the difference between our model architecture and the ODE-RNN architecture.  Thank you for pointing this out, we clarified that in the paragraph where the GRU-ODE-Bayes is presented.
> > >
> > > __2. Loss function__
> > > You are right that there exist similarities between the loss function of GRU-ODE-Bayes and NJ-ODE. This is not surprising, since both models try to control the jumps and also the behaviour between the jumps, which is done by the respective parts of the losses.
> > > To respond to your question, apart from theoretical convergence reasons, there is no other motivation for our choice of loss function. Those choices were completely guided by the convergence proof. We did not try to change specific things as you suggested (combination of different loss functions) for the following reason.
> > >
> > > To apply a negative log-likelihood loss function (pre-loss) and a loss function based on the KL-divergence (post-loss), an assumption on the conditional distribution is needed. In particular, GRU-ODE-Bayes makes the assumption that the conditional distribution is gaussian and models its time-dependent parameters. This assumption is restrictive, if the true underlying conditional distribution can not be described by a gaussian distribution (as for example in the case of the Heston dataset). We go another way, where we do not make any assumption on the underlying distribution. This implies that we cannot use a loss term where such an assumption would be needed.

---

> > > ### Author Response · Authors · 2020-11-25
> > > **Answers to new comments 2/2**
> > >
> > > __3. Theory__
> > > We completely agree with you that those methods that provably attain a global optimum are not the ones used in practice. We would like to emphasize that the convergence guarantees we provide are independent of the choice of the optimization method used to find the optimal weights. By citing those algorithms with provably convergence, we only wanted to mention that if you really wanted to have theoretical convergence from A to Z, you could get it.
> > >
> > > We are fully aware that the study of global convergence of the stochastic gradient descent based methods with non convex objective functions and more specifically neural networks is an active research topic with multiple open questions. Therefore, it is natural not to try to solve this hard and independent problem here. However, even if a proper convergence study of this part is not available, it does not mean that the rest of the convergence study is not relevant for justifying the modeling part. Let us recall our main theorem.
> > >
> > > (1) We assume that for each number of sampling and for every size of the neural networks, their weights are chosen optimally, as to minimize the loss function. (2) Then, if the number of paths and the size of the neural networks tend to infinity, the output of our model converges in mean ($L^1$-convergence) to the conditional exception of the stochastic process $X$ given the current information.
> > >
> > > This theorem proves that our algorithm with our specific loss function converges to the target process (the conditional expectation of the stochastic process). We want to emphasize that assuming to have the optimal weights (with respect to the loss function) does not immediately imply that the output of the model will give the desired result. For example, when we first started to construct our loss function, the convergence to conditional expectation was not guaranteed. We had to change the loss function in order to guarantee this convergence. For example, in GRU-ODE-Bayes, this type of convergence is not given, meaning that they cannot claim that the output of their model will always give the desired result. For instance, in an additional empirical example we provided (Heston model), their algorithm cannot manage to reach the desired result and produces worse outputs when the size of the network is increased. For this reason, we think the theory is particularly relevant for justifying the modeling part, even if we assume to have found the optimal weights.
> > >
> > > Concerning the theoretical argument, while we don’t introduce new tools but rather make use of several existing mathematical tools (Probability theory, Stochastic Calculus and the universal theorem of approximation), we provide a new proof showing that the output of the model converges to the desired result. We are confident that this proof can inspire further research to derive similar theoretical guarantees. For instance, one could be inspired by this proof to show convergence of the ODE-RNN or GRU-ODE-Bayes, or to modify them in order to guarantee convergence. This proof might be adjusted for other similar problems. Therefore, we think that the given theoretical argument can be beneficial to related subfields.
> > >
> > > We thank you for initiating this discussion about the need of theoretical guarantees.
> > >
> > > __Summary__
> > > We hope this addresses all of the reviewer's concerns. We thank you again for the interesting discussion and valuable feedback.

---

### Official Review · AnonReviewer3 · 2020-10-28
**Contribution over existing work is unclear, experimental validation is minimal.**

**Rating:** 4
**Confidence:** 3

**Review:**

The authors propose a method for learning the conditional expectation of stochastic process in an online fashion. The paper bears a considerable theoretical treatment, derived from the stochastic filtering literature, which is present both in the main body of the paper and the appendix. Besides the model, the paper also aims to provide a theoretical justification of the convergence of their method.

I find the contribution of the paper somewhat obscure, its aims to be incremental with respect to the previous literature, and the experimental validation heavily unconvincing. I support my recommendation through the following points:

- Following the well known (by now) neural ODE and neural jump SDE, the contribution of the paper seems minor. The authors state that they focus on giving theoretical guarantees, however, these are specific and loosely validated experimentally
- There is a fair amount of space dedicate to the theoretical presentation of the background, I agree with the importance of theory, but I failed to see how that theory supports the claims of the paper.
-the experiments are limited: only 3 synthetic examples and only one real-world one.
-the authors states that their method focuses on approximating (directly) the conditional expectation, this seems to be a different with the previous literature. However, if that's the case, the authors should consider more benchmarks such as linear filters (adapted to non-uniformly-spaced data), Gaussian processes, or general time series models.

This paper does have a contribution. My recommendation is that  the authors show it in a clearer (to the point) manner with an improved experimental validation.

---

> ### Author Response · Authors · 2020-11-18
> **Clarification of contribution over existing work and additional experiments**
>
> Thank you for your review.
>
> __Show in a clearer manner__
> We have rewritten the main paper and all the rigorous and precise mathematical formulations are moved in the appendix to let place for the important message we want to transmit. It is now written in a very simple way, in order to reach a bigger audience. Thank you for this input.
>
>
>
> __Contribution over existing work__
> Our model is different from the previous work. In the ODE-RNN and in the GRU-ODE-Bayes, it is a recurrent neural network, where between two observations, the hidden state is modeled by a neural ODE. In our model, we don’t use any recurrent neural network. This makes the model much easier and faster to train. We instead have a neural ODE that takes three more parameters (the last observation, the current time and the duration between the current time and the last observation). Moreover, we want to emphasize that we provided a novel training framework. For the first time, we provided a mathematical formulation, a rigorously defined problem statement and based on the new objective function, the theoretical guarantees that our algorithm works, which would not have been possible with a different training framework. In the GRU-ODE-Bayes paper, the authors do not give any theoretical guarantees for the model, only an empirical study.  In contrast, our method is proven to converge to the optimal solution.
>
>
> __Additional experimental validation__
> The main contribution of our project is the theoretical justification and the rigorously defined framework with the theoretical guarantees of convergence. This was not done by any paper prior to our work. For that reason, we think that having tested our model on those three synthetic datasets and on a real world dataset in addition to our theoretical guarantees was suffisant to prove that our algorithm works well in different scenarios.
> However, we have improved our experiment validation by adding the following experiments:
> - Heston model without the Feller condition.
> - Switching regime. In the first half of the path, the stochastic process is following a model M1 and in the second half of the path a model M2.
> - Model with explicit time dependence, i.e. where the drift of the SDE depends on t.
> - Convergence study also on Ornstein-Uhlenbeck and Heston dataset.
> - Experiments on Physionet in the same setting as the extrapolation experiment of the latent ODE (ODE-RNN), adding another comparison to a baseline model.
>
> __Different task than in previous literature__
> Actually, the task of predicting the conditional expectation is not new. It is different from the tasks considered in the latent ODE, but very similar to what the GRU-ODE-Bayes does. They also estimate the conditional expectation together with the standard deviation under normality assumption. More precisely, they try to predict the conditional distribution, under the assumption that it is given by a normal distribution. The predicted mean parameter of the normal distribution therefore is exactly the conditional expectation, which is the main interest in all real world forecasting applications. This is the reason why we mainly compared our method to GRU-ODE-Bayes.
>
> __Summary__
> We hope we have addressed every concern that the reviewer has raised. We would be very happy to have further discussion if there are any other obstacles to raising the review score.

---

### Official Review · AnonReviewer2 · 2020-10-30
**Review of neural jump ODEs**

**Rating:** 7
**Confidence:** 5

**Review:**

Summary: This paper introduces Neural Jump Ordinary Differential Equations as a method for learning models of continuous-time stochastic processes sampled at random time epochs. Specifically, the paper studies the problem of estimating the marginal conditional expectation (i.e., the L2 optimal approximation conditional on the available information) by estimating an auxiliary stochastic differential equation, parameterized by  neural networks, that approximates the conditional expectation of the process of interest at each point in time. The neural networks are trained by using a “randomized” mean squared-loss objective. The main theoretical results in the paper include asymptotic consistency of the optimal objective value in the limit of a large neural network, as well as consistency of a Monte Carlo sample average estimator of the value. The paper also establishes the L2 convergence of the estimated auxiliary solution to the marginal conditional expectation.

The technical details in the paper are mostly sound, and I believe it should be of interest to a wide community. The question of estimating stochastic models sampled at regular or irregular intervals is of broad utility. There are some technical issues however, but these can be resolved I believe. In particular, in the conclusions of Theorem 4.1 and 4.2, it seems as though the authors claim almost sure convergence, unless I am misunderstanding their statement.  What the authors establish is convergence in L2, but why does is imply almost sure convergence? Wouldn’t one require uniform integrability to conclude more? Furthermore, this is not a process level convergence result, and therefore I do not believe that they can conclude (as in Remark G.3) that the limit holds almost surely. (Also, the authors seem to suggest tin Remark G.2 that they’re not establishing L2 convergence, but this could be a problem with the writing).

Coming to the writing, I note that I did find the paper somewhat sloppy in its use of terminology and notation. For instance on p.1 the authors state “...while stochastic processes are continuous in time...” This is not quite true, since one can define discrete-time stochastic processes. I also found the discussion around  justifying “irregular” sampling of the stochastic process to be poorly written. In particular, it is stated that “...dividing the time-line into equally-sized intervals...is again making assumptions and information is lost...” well, any sampling will involve a loss of information, and the randomized sampling process described in this paper also involves assumptions. I don’t think this comment is appropriate. Furthermore, the authors do not make a clear case for why their irregular sampling procedure is appropriate. I’m quite certain that the sampling process introduces bias into the estimation; for instance, Theorem 1 of ref. [1] below provides sufficient conditions under which an “irregularly” sampled estimator of a functional of an SDE is unbiased. The authors must do a better job of justifying their method. I would also urge them to add an example of a randomized sampling process; for instance, a Poisson process sampler would satisfy their definition, in which case the sampling time epochs form an ordered statistic.

Coming to the development of the stochastic model, it is unclear to me as to why all of the random “objects” cannot be defined on the same sample space. Essentially, couldn’t one view the sampling process as a point process on the same sample space supporting the SDE?

Next, in Prop. 2.1, the authors state that the optimal adapted process approximating process is \hat{X}_t — but \hat{X_t} is only defined pointwise (i.e., at each time ‘t’) and it is not defined as a stochastic process. Indeed, for that the authors must describe the finite dimensional distributions for all finite sets of time epochs to define the stochastic process. I believe it is inappropriate to call this a stochastic process. This doesn’t affect the main results, since the authors only establish convergence in an L2 sense, where the full distribution is not necessary.

Some further minor comments:
1. Change the term “observation dates” to “observation epochs”.
2. Change “amount of observations” to “number of observations” (or samples).
3. On P.3 in the definition of \lambda_k, the set \mathcal{B}([0,T]) is undefined.
4. The notation defining the function \tilde{\mu} is very confusing, please change.
5. P.4 “...since the variation of u...” should be “...since the total variation of u...”
6. What do you mean by “ergodic” approximation of the objective? Isn’t it simply a sample average approximation? Which ergodic theorem is playing a role here?
7. I would also urge you clearly define what you mean by \mathbb{L}-convergence, for completion.

[1] Unbiased Estimation with Square Root Convergence for SDE Models, Chang-Han Rhee and Peter W. Glynn.

---

> ### Author Response · Authors · 2020-11-18
> **Clarification of theorems, terminology and other remarks**
>
> Thank you for your review.
>
> __Some unclarities in conclusion of Theorems__
> There was a misunderstanding, we thank you for pointing this out. We try to clarify the points. The conditional expectation is the $L^2$-minimizer for the considered forecasting task. However, we can show convergence of the output of our model to the conditional expectation only with respect to the $L^1$-norm (additional assumptions would be needed for convergence in $L^2$-norm). It is important to differentiate here between
> the 2-norm that is used to make the d_X-dimensional random variables 1-dimensional inside the expectations and
> the $L^1$-norm used to show $L^1$-convergence to 0 for this 1-dimensional random variable.
> Since the 2-norm is equivalent to any other norm on $\mathbb{R}^{d_X}$, this choice does not influence the result in any way.
> As correctly remarked by the reviewer, convergence in $L^1$ does not imply almost sure convergence. However, we did not claim almost sure convergence, but only that the limits are equal almost surely, which is a direct consequence of $L^1$-convergence. To make our claim clearer, we added Lemma E.6 stating this consequence and reference to it.
>
> __Sloppy use of notation and terminology__
> We agree that the terminology wasn’t used appropriately at the outlined points and thank him for bringing this to our attention. We changed the passages to be more precise and appropriate.
>
> __Irregular sampling procedure & sampling process as point process__
> We are not sure to correctly understand the remark. The irregular observation dates are needed to describe data that is observed at irregular times. In particular, we do not take the point of view that we have a model of which we can sample as often as we want. Instead, we try to give a mathematical description for data that is irregularly observed at random time points. We changed the terminology in the paper the better explain our point of view. We tried to keep the definition of the irregular observation times very general, under the  assumption that the observation times are independent of the stochastic process.
> The reviewer is right that point processes on the same probability space are a way to define observation dates that might be correlated with the stochastic process. We had the  impression that the given way of defining the observation dates make them easier to understand, even though the product probability space consequently has to be considered. As suggested, examples for randomized sampling processes were added to the paper, including the suggested Poisson point process.
> We do not analyse the bias of our algorithm, but only show that it is consistent. It is clear that the sampling procedure can introduce a bias to an estimator. An extreme case would be, that the time interval is divided in half and observations are only made on the first half, for which any estimator could hardly learn anything about the second half. However, this is already incorporated in our convergence results through the dependence on the probability measures $\lambda_k$.
>
> __Is__ $\hat{X}_t$ __a stochastic process?__
> Yes, in contrast to a Gaussian process, in its basic definition, a stochastic process is just a collection of random variables indexed by some index set (Wikipedia). Often this set is the time interval, as for example in the definition in [1, page 3 after Theorem 1]. In particular, a stochastic process is defined pointwise at each $t$. Therefore, this definition applies to $\hat{X}_t$, which is defined pointwise.
> [1] P. Protter. Stochastic integration and differential equations. 2005.
>
> __About further minor comments__
> 1. We do not understand why the term “observation epochs” would be suited better. In particular, observations are always made only at discrete time points, rather than on time intervals (which would correspond to “epochs” from our point of view). Maybe we misinterpreted this comment?
> 2. We changed it, thanks for pointing this out.
> 3. The definition of $\mathcal{B}([0,T])$ was added, thanks.
> 4. We made a small change to the notation. We are not sure which notation would be better or less confusing for $\tilde{\mu}$. Would you have any suggestions?
> 5. Correct, we changed it, thanks for pointing out.
> 6. What we mean here is that we take an average base only on one realization of the path, by averaging over the different observations in time rather than by averaging over multiple realizations of the paths. Such a time-average only equals the sample average under ergodicity assumptions, which we assume to be satisfied here. We do not explicitly use an ergodic theorem, but suppose that the claim of such a theorem is satisfied in the stated way.
> 7. The definition was added to the paper (now in appendix).
>
> __Summary__
> We hope this addresses all of the reviewer's concerns. If the reviewer has any further questions by which our paper and their score may be improved, then we would be happy to address these as well.

---

### Official Review · AnonReviewer4 · 2020-11-01
**Not very novel method, difficult read.**

**Rating:** 7
**Confidence:** 2

**Review:**

##########################
The paper proposes an algorithm and an analysis of its convergence.
The algorithm propose to learn a model of temporal data y_1 , ... , y_T given input x_1, ..., x_T
The observations are assumed to arise from a deterministic latent h
governed by a piecewise continuous ode (in between consecutive times t_i, t_i+1)
with additional deterministic jumps at transitions.

In the ODE-RNN paper, the latent h can be expressed as a single ODE for the whole time horizon (rewriting the
jump with a skip transition).

This paper appears to me as taking this expression and choosing a particular bounded form for the
dynamics, jump and readout functions
A statistical asymptotic analysis of the convergence of the algorithms, for random times and inputs is given.

##########################

Methodology:
I find the paper quite difficult to read, I blame both its structure and my lack of ease with the mathematics used here.
However from what I have understood of the algorithm proposed, I find the methodological contribution very limited.

Clarity:
I come from the machine learning community and read with no difficulty
papers cited in the related work section.
In comparison, I find this paper extremely difficult to read and parse despite containing the same kind of information.

Asymptotic analysis.
I leave to other reviewers the evaluation of the convergence analysis.
My evaluation being partial, my confidence rating is set accordingly.

* For a machine learning paper presenting in the end a 3 line simple algorithm, the paper contains
a lot of superfluous mathematical notation that crowds the paper and make the reading very tedious.
Many of the papers cited Brouwer 2019, Rubanova 2019, Li 2020, offer a much smoother read in that respect.
As is, this paper feels better suited to a more specialist statistics venue.

* For example, many elements are introduced in the main text and are not really necessary to understand what the paper does
The detailed section on random inputs is used only in a theorem coming later, why have it in the main text in so much details.
On the other end, a description of the method this paper builds on is left into appendices.

#########################

Additional comments:
* the formatting of the references is very inconsistent, please update

---

> ### Author Response · Authors · 2020-11-18
> **Simplified paper and clarification of our contribution**
>
> Thank you for the review.
>
> __Difficult read__
> We simplified the paper a lot. Everything is now described with simple words and there are no more mathematical technicalities in the main paper. The rigorous problem statement, the mathematical description and all the theoretical guarantees are moved into the appendix. To clarify even more the paper, we describe the previous methods and we explain how we built our model in the main paper as you suggested.
>
> We believe that it is important to have a solid and rigorous mathematical explanation of those recent techniques and we think that this is a significant contribution for the machine learning community. However, we completely agree with you that it can be explained in a better and simpler way, such that a bigger audience can understand and take advantage of our contribution. This is what we tried to do and we hope that you will appreciate the current version. Thank you for taking the time to go through it. We also changed the formatting of the references as you suggested.
>
> __On choice of venue__
> We have submitted to ICLR because our paper is about improving an already-existing machine learning technique.
>
> __Novelty__
> Our model is different from the previous work. In the ODE-RNN and in the GRU-ODE-Bayes, it is a recurrent neural network, where between two observations, the hidden state is modeled by a neural ODE. In our model, we don’t use any recurrent neural network. This makes the model much easier and faster to train. We instead have a neural ODE that takes three more parameters (the last observation, the current time and the duration between the current time and the last observation). Moreover, we want to emphasize that we provided a novel training framework. For the first time, we provided a mathematical formulation, a rigorously defined problem statement and based on the new objective function, the theoretical guarantees that our algorithm works, which would not have been possible with a different training framework. In the GRU-ODE-Bayes paper, the authors do not give any theoretical guarantees for the model, only an empirical study.  In contrast, our method is proven to converge to the optimal solution.
>
> __Summary__
> We hope we have addressed every concern that the reviewer has raised. We would be very happy to have further discussion if there are any other obstacles to raising the review score.

---

> > ### Comment · AnonReviewer4 · 2020-11-23
> > **post rebuttal evaluation**
> >
> > Following https://openreview.net/forum?id=JFKR3WqwyXR&noteId=akFNuozOZ1p
> > I have updated my rating.
> >
> > Concerns on clarity and motivation have been addressed properly

---

> > > ### Author Response · Authors · 2020-11-24
> > > **Thank you for the revaluation**
> > >
> > > We thank you for having studied our updated paper and for updating your score. We are glad that you are satisfied with the new version.

---

### Decision · Program_Chairs · 2021-01-07
**Final Decision**

**Decision:**

Accept (Poster)

**Comment:**

This paper proposes a refinement, and analysis of, continuous-time inference schemes.

This paper got in-depth criticism from some very thoughtful and expert reviewers, and the authors seem to have taken it to heart.  I'm still worried about the similarity to GRU-ODE-Bayes, but I feel that the clarifications to the general theory of continuous-time belief updates is a worthy contribution, and the method proposed is a practical one.  One reviewer didn't update their score, but the other reviewers put a lot of thought into the discussion and also raised their scores.

I do think the title and name of the method is a bit misleading - I would call it something like "Consistent continuous-time filtering", because the jump ODE is really describing beliefs about an SDE.

---

> ### Author Response · Authors · 2021-03-16
> **Thank you**
>
> We thank you and all the reviewers for all the valuable feedback that contributed to considerably improving the paper.
> We changed the title of the paper based on your suggestion, thank you.